# A 1 km soil organic carbon density dataset with depth of 20cm and 100cm from 1985 to 2020 in China

Yi Dong[&], Xinting Wang[&], Wei Su[*]

College of Land Science and Technology, China Agricultural University, Beijing 100083, China

5 *Correspondence to*: Wei Su (suwei@cau.edu.cn)

&These authors contributed equally to this work and should be considered co-first authors.

**Abstract:** Soil organic carbon (SOC) is an important component of the global carbon cycle and a vital indicator of ecosystem health, playing key roles in agricultural productivity and climate change mitigation. To trace the spatiotemporal dynamics of SOC in China, a high-resolution (1 km) Soil Organic Carbon Density (SOCD) dataset for the 0–20 cm and 0–100 cm depths spanning the period from 1985 to 2020 is produced in this study. By integrating Landsat archives, topographic and meteorological data, and 11,743 soil profile measurements, we produced the SOCD dataset from 1985 to 2020 in China using the Random Forest ensemble learning approach. Specially, a climate zoning strategy was developed to account for the significant environmental heterogeneity across China. The validation of our SOCD estimated results with 0-20 cm depth with independent testing samples showed strong agreement with $R^2$=0.63 and RMSE=2.03 (kg C/m²) for 0-20 cm SOCD estimation and $R^2$=0.62 and RMSE=6.16 (kg C/m²) for 0-100 cm. Moreover, our SOCD estimated results with 0-20 cm depth are aligned well with independent samples ($R^2$=0.76, RMSE=1.75 kg C/m²) and Xu's dataset ($R^2$=0.68, RMSE=1.70 kg C/m²). Furthermore, the validation of our SOCD estimated results with 0-100 cm depth with independent measurements from Dong et al. (2024) showed strong agreement ($R^2$=0.50, RMSE=4.93 kg C/m²). Furthermore, our SOCD product exhibits high consistency with existing global datasets (HWSD, SoilGrids250m, and GSOCmap), showing the best fit with SoilGrids250m ($R^2$=0.74, RMSE=1.03 kg C/m²). Comparisons of model predictions to independent datasets from the 1980s, 2000s, and 2010s in China reveal substantial connections and demonstrate strong performance over time. The estimated SOCD products, along with the compiled raw soil profile observations for both 0–20 cm and 0–100 cm depths, are openly available via Figshare (https://doi.org/10.6084/m9.figshare.27290310.v2) (Dong et al., 2024).

# 1 Introduction

Soil organic carbon (SOC) plays a fundamental role in the earth system by mediating the fluxes of carbon, energy, and water (Chaney et al., 2019; Crow et al., 2012). The foundation of soil fertility lies in soil carbon, a significant component of terrestrial carbon storage. SOC accounts for more than half of total soil carbon and is an essential component of the soil carbon cycle, which has a major impact on soil fertility and agricultural productivity (Baldock, 2007; Chen et al., 2022). A combination of natural and human forces is placing significant strain on the global SOC reservoir. SOC content estimation has become a hot spot in global climate change due to its close relationship with climate change. The sustainability of agricultural production is threatened worldwide by soil degradation and the loss of intimate relationships. The sustainability of agricultural production is threatened worldwide by soil degradation and the loss of intimate relationships (Xu et al., 2018). This lack of high-resolution, long-term observational data impedes precise assessment of soil degradation and carbon sequestration potential. Therefore, developing a robust, spatiotemporally continuous SOC density (SOCD) dataset for China is urgent.

In recent years, increasing attention has been paid to estimating SOC across global, national, and regional scales (Padarian et al., 2022). In-depth studies to estimate subsurface SOC content estimation, particularly at a regional scale, remain challenging due to the difficulty of data collection, the lack of long-term observations, and the depth dependency of soil carbon sequestration (Padarian et al., 2022). The advancement of digital soil mapping technology opens up new paths for estimating SOC content in large-scale and long-term series (Li et al., 2024). The use of machine learning techniques for digital soil modeling is a common concept in DSM. Compared to traditional mapping methods such as geo-statistics, expert knowledge, and individual representation, machine learning techniques provide a new paradigm for estimating SOC content in large-scale and long-term series. To produce continental-scale SOC-weighted mean maps, Odgers et al. (2012) used an equal-area spline function for soil databases, while Mulder et al. (2016) used a machine learning model with a three-dimensional distribution to estimate SOC content in eastern France. These studies provide evidence for a comprehensive and accurate understanding of soil properties and their spatial variation. Despite these advances, most digital soil mapping studies have focused on a specific period and the long-term dynamics of SOCD mapping have not yet been developed. Emadi et al. (2020) predicted the SOCD in northern Iran using a sample of 1879 measurements, and Nabiollahi et al. (2019)

used a random forest (RF) model to predict the SOCD at 137 sites in Marivan, Kurdistan Province, Iran. However, these studies only focus on local zones. In China, researchers have paid considerable attention to the sequestration potential of SOC storage, but most studies have focused on specific experimental areas or ecosystem types. Fang et al. (2007) estimated the carbon sink of terrestrial vegetation in China. Furthermore, these studies often lack attention to long-term trends and dynamics, resulting in insufficient data sets to fully understand climate change and the impact of human activities on SOCD. At the national level, there is relatively little study on the potential for organic carbon storage across different ecosystem types (O'Rourke et al., 2015). The scarcity and unevenness of SOC data in China, as well as the lack of effective estimation methods, all contribute to the uncertainty of SOC prediction. In addition, the diverse and complex topography in China, as well as the lack of measured SOCD data, have increased the difficulty of SOC content estimation. Previous studies often used the data from inventories of relevant resources to make rough calculations of carbon sinks (Pan et al., 2004). Unfortunately, the spatial continuity and variability of SOC, the spatial differentiation of organic carbon sequestration potential, and the influence of environmental factors have not been considered in previous studies. Especially in western China, there is almost no measured SOC data (Liu et al., 2022), which poses a challenge for understanding terrestrial ecosystems and soil carbon sinks in China. Given these challenges, it is urgent to carry out SOCD mapping and analyze the temporal and spatial changes of SOCD in China.

To produce robust and accurate long-term SOCD products in China, we explore the RF models with climate zoning to predict SOCD in China from 1985 to 2020 and improve the study of SOCD maps for the 0-20 cm and 0-100 cm soil layers in China. The Landsat TM/ETM+/OLI images, topography, meteorology, and soil properties data are used for SOCD mapping in this study. The main contributions of this study can be summarized as follows.

(1) A nationwide, long-term soil organic carbon density dataset from 1985 to 2020 with depths of 20cm and 100cm in China is provided in this study.

(2) The machine learning RF models zoned by climate zones in China are developed for SOCD estimation, and the spatial-temporal variability of soil carbon is considered in our SOCD estimation.

(3) The proposed framework provides a comprehensive understanding of SOCD estimation including spectral indices of satellite remote sensing images, digital elevation model (DEM) and its topographic derivatives, meteorological features, and soil properties. The technique offers the potential for SOCD mapping with sufficiently measured SOC content data.

## 2 Study area and data sources

### 2.1 Study area

The study area, which extends throughout China, is characterized by complex and diverse terrains including mountains, plateaus, basins, plains, and deserts (Yuan et al., 2023). In addition, China has a large latitude difference from 4°N to 53°N and a large longitude difference from 73°E to 135°E. Therefore, there are obvious differences in precipitation and

temperature in the study area, which bring significantly different accumulation processes and spatial patterns of soil carbon (Zheng et al., 2023) . In addition, there are various soil types, including red soil, brown soil, black soil, and chestnut calcium soil, which have obvious spatial characteristics in the study area (Shangguan et al., 2014) . For these reasons, we developed four different RF models for SOCD estimation for four temperature zones from south to north in China including humid area, semi-humid area, semi-arid area and arid area.

### 2.2 Data sources

(1) SOC content data

After removing duplicates and incomplete records, we compiled a comprehensive database of 11743 soil profiles containing measured SOC content and bulk density across China, spanning three distinct periods including the 1980s, 2000s, and 2010s.The SOC content and soil mass weight data of the 1980s were collected from the profile database of the Second

National Soil Survey (1980-1996) (http://www.geodata.cn). The SOCD data of the 2000s was collected from the China Terrestrial Ecosystem Carbon Density Dataset (2000-2014) (http://www.cnern.org.cn/). The SOC content data of the 2010s was collected from the Soil Attribute Data of the China Soil System Record (2010s) (https://www.resdc.cn/), which was measured in the China Soil System Survey Collection and China Soil System Journal Compilation Project. To enhance spatiotemporal coverage, particularly for data-scarce regions, we incorporated additional SOC data from two recent national

data products: the national soil organic carbon density dataset for 2010–2024 in China (Chen et al., 2025) and the updated

China dataset of soil properties for land surface modelling (Shi et al., 2025). We harmonized the point-level information from these datasets (profile ID, latitude and longitude, upper and lower depth, SOC content, sampling year, and land-use type) to match the structure of our database. Then a detailed overlap analysis between these profiles and our original compilation was done. Because many profiles in Chen et al. (2025) and Shi et al. (2024) originated from the same legacy

sources as our database, we applied a strict de-duplication procedure based on geographic coordinates, sampling year, and depth structure to identify duplicated entries. Profiles that matched existing profiles within a small spatial tolerance and with similar temporal and depth characteristics were treated as duplicates and excluded. Only those profiles that could be clearly identified as non-overlapping were retained and merged into our database.

To evaluate the generalization capability of our developed model rigorous, we employed three independent datasets that

were not involved in the training process. These datasets cover different spatial scales and ecosystem types, ensuring a robust assessment of our SOCD products, including the measured SOC content data in the Heihe River basin (Song et al., 2016), the measured SOCD data from Xu et al. (2018), and the soil inorganic carbon (SIC) and SOC density dataset from Dong et al. (2024). The SOC content data of the Heihe River basin were collected from the spatio-temporal Tripolar Environmental Big Data Platform (https://poles.tpdc.ac.cn/zh-hans/). The measured SOCD data from Xu et al. (2018) focuses on SOC densities

and soil carbon storage with a depth of 0–20 cm in various terrestrial ecosystems in China. The data was measured in field campaigns between 2004 and 2014, as well as some unpublished field measurements. The dataset from Dong et al. (2024) provides comprehensive measurements of SOC and inorganic carbon densities across 0-100 cm profiles in Chinese grassland and desert ecosystems, along with key environmental drivers such as climate variables, soil properties (texture, pH, conductivity), nitrogen deposition, and root biomass. This multi-source validation enhances the robustness of our SOCD

assessments across different ecosystems and soil depths.

(2) Landsat archives

The time-series archived Landsat 4, 5, 7, and 8 TM/ETM+/OLI images spanning from 1985 to 2020 (Yu et al., 2023) are used for SOCD estimation, which are retrieved from the GEE cloud computing platform (Liu et al., 2024). Preprocessing of Landsat images, including radiometric calibration, atmospheric correction, geometric correction, cloud identification, and

spectral index calculating are carried out on the GEE cloud computing platform. Random sampling and statistical regression

analysis are performed to determine the calibration coefficients for each band spectral reflectance. Principal major axis regression models are used to normalize the reflectance data for different sensors. Radiometric correction coefficients of different Landsat sensors are calculated (Fig. 1). The spatially overlapping images are combined into one image using the aggregation function, and the combined image dataset is subjected to stitching operations to produce spatially coherent

images. A variety of spectral indices were calculated using Landsat images after processing. Spectral indices Normalized Difference Vegetation Index (NDVI), Bare Soil Index (BSI), Enhanced Vegetation Index (EVI), Land Surface Water Index (LSWI), and Soil-Adjusted Vegetation Index (SAVI) were calculated using Landsat images. The formulae for these spectral indices are as follows:

$$NDVI = \frac{\rho_{NIR} - \rho_{Red}}{\rho_{NIR} + \rho_{Red}} \tag{1}$$

$$BSI = \frac{(\rho_{SWIR} + \rho_{Red}) - (\rho_{NIR} + \rho_{blue})}{(\rho_{SWIR} + \rho_{Red}) + (\rho_{NIR} + \rho_{blue})} \tag{2}$$

$$EVI = 2.5 \times \frac{\rho_{NIR} - \rho_{Red}}{\rho_{NIR} + 6 \times \rho_{Red} - 7.5 \times \rho_{blue} + 1} \tag{3}$$

$$SAVI = \frac{\rho_{NIR} - \rho_{Red}}{(\rho_{NIR} + \rho_{Red} + 0.5) \times 1.5} \tag{4}$$

$$LSWI = \frac{\rho_{NIR} - \rho_{SWIR1}}{\rho_{NIR} + \rho_{SWIR1}} \tag{5}$$

Where, $\rho_{NIR}$ is the reference of near-infrared band, $\rho_{Red}$ is the reference of red band, $\rho_{blue}$ is the reference of blue band,

$\rho_{SWIR}$ is the reference of short-wave infrared band and $\rho_{SWIR1}$ is the reference of short-wave infrared band 1.

The land cover dataset newly released by Wuhan University (Yang and Huang, 2021) is used in this study. This is the first China Land Cover Annual Data Set (CLCD) derived from Landsat on the GEE platform.

(3) DEM and its topographic derivatives

Terrain is an important factor affecting the formation of soil organic matter. The DEM data is used for SOCD estimation,

which is downloaded from the Resource and Environment Science Data Platform of the Chinese Academy of Sciences (https://www.resdc.cn) with a spatial resolution of 500 m. Topographic data and its topographic derivatives are extracted from the DEM data. There are four terrain derivatives, including Slope, Aspect, Elevation, and Topographic Wetness Index (TWI), which are calculated using SAGA GIS version 8.0.1 (https://saga-gis.org/) (Zhang et al., 2023). The spatial resolution

of all raster data was uniformly adjusted to 1000m using resampling techniques to achieve spatial consistency between
different datasets.

(4) Meteorological data

The meteorological features including Temperature (Tem), Precipitation (Pre) and Solar Radiation (SR), measured in 2,400 Chinese meteorological stations are used to quantify the effects of meteorological fluctuations. All meteorological data are downloaded from the China Meteorological Data Network (http://data.cma.cn/). For spatial consistency, the meteorological

data is defined and projected into WGS 84 coordinates. All meteorological point data are interpolated into grid data with 1000 m spatial resolution using the ANUSPLIN program (Padarian et al., 2022). Crucially, to account for the lapse rate effect in complex terrain, DEM data was used as a covariate during the spline interpolation process.

(5) Published soil database

There are four published soil databases used to validate the SOCD estimation results in this study. One is the Harmonized

World Soil Database (HWSD v2.0), produced by the International Institute for Applied Systems in Vienna and the Food and Agriculture Organization of the United Nations. There are two soil properties including soil bulk weight and organic carbon content are used for SOCD estimation at depths of 0-20 cm, 20-40 cm, 40-60 cm, 60-80 cm, and 80-100 cm. The SoilGrids250m v2.0 dataset including the soil silt content, sand content, clay content, and organic carbon content data with the spatial resolution of 250 m are downloaded from FAQ SoilGrids (https://soilgrids.org/) for validation. For spatial

consistency, this soil attribute datum is resampled to 1000 m. This soil product with five depth intervals (5 cm, 15 cm, 30 cm, 60 cm, and 100 cm) is used to calculate the soil silt content (Silt), sand content (Sand), clay content (Clay), and organic carbon at 0-20 cm and 0-100 cm (Zhang et al., 2023). Taking the clay content data as an example, the clay content with depths of 0-20 cm and 0-100 cm is calculated as follows:

$$CLY_{020} = \frac{CLY_{05}}{4} + \frac{CLY_{515}}{2} + \frac{CLY_{1530}}{4} \qquad (6)$$

$$CLY_{0100} = \frac{CLY_{05}}{20} + \frac{CLY_{515}}{10} + \frac{3}{20} \times CLY_{1530} + \frac{3}{10} \times CLY_{3060} + \frac{2}{5} \times CLY_{60100} \qquad (7)$$

Where, $CLY_{05}$, $CLY_{515}$, $CLY_{1530}$, $CLY_{3060}$, and $CLY_{60100}$ are the clay content (g/kg) at depths of 0-5, 5-15, 15-30, 30-60, and

60-100 cm respectively.

The GSOCmap dataset (https://www.fao.org/), which is the first global SOC product led by FAO, is used for validation. GSOCmap is a 1-kilometer soil grid that covers depths ranging from 0 to 30 centimeters. The SOC Dynamics ML dataset in China is now available on the Dryad platform (https://datadryad.org/). Using machine learning, the dataset aims to capture the dynamics of SOC and its drivers in different soil horizons in China between the 1980s and 2010s (Li et al., 2022). The dataset contains valuable information such as SOC stocks, carbon fixation rates, and SOC content. While these existing datasets offer broad insights into SOC, our study specifically focuses on refining the estimation of SOCD for precise national-level carbon accounting across multiple historical periods. The organic carbon density with the depth of 20 cm and 100 cm in the 1980s, 2000s, and 2010s in China is used. This study focuses on SOCD, which is different from SOC content. The conversion from SOC content to SOCD is presented in Section 3.1.

## 3 Methodology

### 3.1 Converting SOC to SOCD with normalized soil depth

The dataset from the 2000s provided pre-calculated SOCD values (derived from SOC, bulk density, and coarse fragments by the original data source), while the data reported in 1980s and 2010s are SOC content. For data consistency, we converted all SOC content data to SOCD using Equation 8. The SOC content data of 1980s were from the Second National Soil Survey, and the SOC content data of 2010s were from the Soil Attribute Data of China Soil System Record, which had several different soil depths. For the consistency of the measured data, we convert the soil data with different depths into the SOCD with the depth of 0-20 cm and 0-100 cm using the package "mpspline2" v.0.1.3 (Bishop et al., 1999). The observed horizons, defined by their upper and lower depths, were input to `mpspline2`, which fits a mass-preserving spline to the vertical SOC profile and integrates this spline over the target depth intervals. We used the default value of 0.1 for the spline smoothing parameter lambda. We do not extrapolate beyond the observed soil depth when calculating SOCD. Profiles shallower than 100 cm are used to compute SOCD only for depth intervals that are fully covered by observations, but they are excluded from 0–100 cm SOCD statistics. When we report and analyze SOCD for the full 0–100 cm interval, we therefore restrict the calculations to profiles with an observed depth of at least 100 cm after quality control. For all datasets, SOCD (kg C/m$^2$) is

calculated using bulk density (kg/m³), and coarse fractions percentage (%) provided by the National Soil Information Grids of China (Liu et al., 2022).

$$SOCD = \frac{SOC \times BD \times SD}{100} \times \left(1 - \frac{CF}{100}\right) \tag{8}$$

Where, SOC is the soil organic carbon content (%), SOCD is the soil organic carbon density, BD is the soil bulk density, SD is the soil depth (cm), and CF is the coarse fractions in a specific soil layer.

## 3.2 Feature selection for RF modelling

To achieve optimal prediction accuracy for SOCD and to elucidate its underlying mechanisms using RF models, comprehensive feature selection for the numerous potential environmental driving factors is a critical prerequisite (Jiang et al., 2024). This process is instrumental in mitigating model complexity, enhancing computational efficiency, improving model interpretability, and eliminating data redundancy that could adversely affect model performance. In this study, the initial feature set comprised diverse categories of crucial environmental drivers, including remote sensing indices (e.g., NDVI, BSI, etc., derived from Landsat satellite imagery), topographic factors (e.g., elevation, slope, aspect, etc., generated from DEM), climatic factors (e.g., mean annual temperature, mean annual precipitation, etc.), as well as auxiliary soil attributes (e.g., soil type) and other relevant indicators (Fig. 2).

Our methodology commenced with a combined approach of correlation analysis, random forest importance ranking, and combinatorial optimization. First, a Pearson correlation matrix was constructed for the initial candidate features, and those exhibiting high correlation (specifically, where the absolute value of the Pearson correlation coefficient exceeded 0.95) were removed to reduce redundancy. The remaining features, representing a refined set, then underwent an importance assessment and ranking utilizing the RF algorithm. A preliminary RF model was constructed with these features as inputs and SOCD as the target variable, and each feature's importance in predicting SOCD was quantified using Gini importance scores, thereby enabling the preliminary identification of core factors possessing substantial explanatory power for SOCD variation. This iterative procedure ensured the high independence of the selected feature set, preventing information overlap from impairing model performance and interpretability. Finally, to identify the optimal feature combination capable of maximizing model prediction accuracy, an exhaustive combinatorial search was conducted on the 10 most informative features remaining after

the initial screening steps. Through a comprehensive evaluation of all possible feature subsets' performance, aiming to maximize the coefficient of determination ($R^2$), seven key environmental driving factors were ultimately identified as collectively providing the best predictive performance for SOCD: Temperature, Elevation, NDVI, Clay, SR, BSI, and Slope. This rigorous selection process ensures that the chosen feature set effectively characterizes SOCD dynamics while optimizing the model's predictive capability.

The selected features represent fundamental controls on SOCD through their influence on microbial activity (temperature), carbon input (vegetation indices), physical protection (clay content), and soil redistribution processes (slope). This multi-stage selection approach effectively balanced model complexity with predictive power while maintaining the ecological interpretability of the final feature set. The robustness of the selected features was further confirmed through cross-validation, demonstrating consistent performance across different validation datasets.

### 3.3 Climate zoning in China

Climate zoning is carried out to quantify the differences in temperature and precipitation in China and improve the accuracy of SOCD estimation. China spans a vast geographical area, crossing multiple major climate zones from the eastern coast to the western interior and from the subtropical monsoon climate in the southeast to the temperate continental climate in the northwest. This extensive climatic complexity leads to pronounced regional heterogeneity in soil formation and carbon cycling, which necessitates a zoned approach for accurate SOCD estimation. According to Tang et al. (2018), there are obvious differences in SOCD observed in different climate zones of China for the diverse and complex environmental factors under warm-temperate climate conditions with a mean precipitation (MAP) threshold of 400 mm and a mean annual temperature (MAT) threshold of 10 °C. To mitigate the interannual variability, the multi-annual average temperature and precipitation are used to classify the climatic differences in China into four subzones including humid areas (MAP $\geqslant$ 400 mm and MAT $\geqslant$ 10°C), semi-humid area (MAP $\geqslant$ 400 mm and MAT $\leqslant$ 10°C), semi-arid area (MAP $\leqslant$ 400 mm and MAT $\leqslant$ 10°C) and arid area (MAP $\leqslant$ 400 mm and MAT $\geqslant$ 10°C) (Fig. 3). Soil data and environmental variables are grouped in each subzone, and zonal SOCD estimation models are developed for each subzone with depths of 0-20 cm and 0-100 cm (Fig. 3).

## 3.4 SOCD estimation using zoned RF models

For the estimation of SOCD, RF models were developed independently across four distinct climate subzones (arid, semi-arid, humid, and semi-humid) and for two soil depths (0-20 cm and 0-100 cm).Within each subzone, the RF model aggregates predictions from numerous decision trees, enhancing forecast stability and accuracy (Wu et al., 2021). This ensemble approach inherently mitigates overfitting, as individual trees are constructed from random subsets of data and features (Sun et al., 2024), thereby significantly improving the generalization of models. Especially, our RF model is conceptualized as a single, unified space-time model, meticulously trained on a comprehensive pooled dataset spanning distinct historical decades (1980s, 2000s, and 2010s). This unified framework, a key novelty of our approach, facilitates consistent SOCD prediction across multiple historical intervals (1985-2020 in five-year increments) for the vast and diverse Chinese region. The methodology effectively leverages the 'space-for-time' principle (Heuvelink et al., 2021) by integrating soil samples collected across these decades into a single training process. This enables the RF model to learn intricate relationships between environmental covariates and SOCD under varying historical conditions, inferring temporal SOCD evolution driven by dynamic factors based on observed spatial patterns.

The RF model inputs, established within the Scikit-Learn framework, comprised both static and dynamic predictors. Dynamic covariates, such as temperature, NDVI, SR, and BSI, were precisely matched to their corresponding five-year mapping periods by utilizing their average values for those intervals (e.g., 1985-1990). Model parameters, including the number of trees, the percentage of randomly selected features, and maximum tree depth, were tuned using a param_dist dictionary to optimize performance during cross-validation. The model's robustness and spatiotemporal capabilities are underscored by a sophisticated stratified spatiotemporal K-fold cross-validation strategy. This involved spatially stratifying the study area into K independent sub-regions to address autocorrelation and assess generalization to new locations. Critically, temporal stratification ensured proportional representation of samples from all three decades within each spatial fold's training and validation sets, allowing the model to learn complex SOCD change patterns over time. The optimized RF model was subsequently employed to predict SOCD across the entire study area, utilizing measured SOCD values alongside spectral indices from soil properties, Landsat archives, topographic derivatives, and meteorological elements. Model performance and generalization ability were rigorously validated using the coefficient of determination ($R^2$) and root mean

square error (RMSE). The trained model was saved using the joblib library, and the resulting estimations were combined with a geographic coordinate system to generate digital SOCD maps, facilitating the exploration of relationships between SOCD and optimized environmental variables.

## 4 Results and conclusions

### 4.1 Statistical analysis of sampling points

The statistics of the measured SOCD values are presented in Fig. 4. For the 0–20 cm soil layer, the mean value of SOCD in the 1980s was 4.16 kg C/m², and the data showed a positively skewed distribution. In the 2000s, the mean value of SOCD slightly decreased to 4.02 kg C/m², accompanied by increased variability resulting from a larger number of sampling sites. Sampling density was highest in the 2010s, during which the mean value of SOCD rose modestly to 4.14 kg C/m² and the distribution became more strongly skewed. For the 0–100 cm soil layer, the mean value of SOCD in the 1980s was 10.92 kg C/m². In the 2000s, the mean value decreased notably to 8.81 kg C/m², accompanied by reduced data variability. In the 2010s, the mean value of SOCD increased again to 11.65 kg C/m², and the distribution displayed a more pronounced skew and thicker tails, reflecting greater heterogeneity in deep-soil carbon stocks across regions (Fig. 4).

Fig. 5 shows the geographical arrangement of SOCD data based on Whittaker biomes with depths of 20 cm and 100 cm in the 1980s, 2000s, and 2010s in China. The distribution of samples shows significant regional concentration and geographical variation, with most points concentrated in the northeastern plain, southwestern plateau, hilly zones, and southeastern coastal zones. There are fewer SOCD samples in northwestern China due to difficult human accessibility, lower vegetation cover, less human activity, and a dry environment. In terms of timing, there are fewer SOCD sample sites in the 1980s. The number of sampling sites increased in the 2010s, particularly in agriculturally developed and densely populated areas.

### 4.2 Model performance of SOCD estimation

To evaluate the model performance of SOCD estimation at depths of 0-20 cm and 0-100 cm, two key indicators were utilized, including the coefficient of determination ($R^2$) and the Root Mean Square Error (RMSE). $R^2$ quantifies the proportion of variance in the dependent variable explained by the model, while RMSE assesses the discrepancy between

model predictions and estimated results. The precision of RMSE values is further characterized by their 95% confidence intervals (CI), providing insight into the robustness and statistical significance of observed performance differences.

As shown in Fig. 6, climatic zoning improved model substantially performance for both soil depths relative to the global (non-zoned) approach. For the SOCD prediction in the depth of 0-20 cm, the global model achieved an accuracy of $R^2=0.46$ and RMSE = 2.38 kg C/m² (95% CI of [2.22, 2.55]). After incorporating climatic zoning, the accuracy of SOCD estimation was significantly improved with $R^2=0.63$ and RMSE = 2.03 kg C/m² (95% CI of [1.95, 2.13]), demonstrating an $R^2$ increase of 0.17 and an RMSE decrease of 0.35 kg C/m². The non-overlapping confidence intervals for the global and zoned models (95% CI of [2.22, 2.55] vs. [1.95, 2.13]) clearly indicate a statistically significant improvement in RMSE due to climatic zoning. Similarly, for the SOCD prediction in the depth of 0-100 cm, the global model yielded an $R^2=0.43$ and RMSE = 8.06 kg C/m² (95% CI of [6.76, 9.49]). With climatic zoning, the performance enhanced to $R^2=0.62$ and RMSE = 6.16 kg C/m² (95% CI of [5.65, 6.89]), reflecting an $R^2$ increase of 0.19 and an RMSE decrease of 1.9 kg C/m². Here again, the distinct confidence intervals ([6.76, 9.49] vs. [5.65, 6.89]) confirm the statistical significance of performance enhancement from zoning. Overall, the SOCD estimation model for the 0-20 cm depth generally exhibited a higher $R^2$ compared to the 0-100 cm depth model, indicating greater complexity in modeling deeper SOC dynamics with available covariates.

Further analysis of model performance across different climate zones revealed distinct patterns at the 0–20 cm depth (Fig. 7). The model achieved the highest accuracy in the semi-arid zone ($R^2=0.70$, RMSE =1.95 kg C/m², 95% CI of [1.80, 2.11]), followed closely by the semi-humid zone ($R^2=0.67$, RMSE =2.20 kg C/m², 95% CI of [2.07, 2.32]). In contrast, the arid zone exhibited moderate performance ($R^2=0.65$, RMSE =1.86 kg C/m², 95% CI of [1.15, 2.68]), while the humid zone showed the lowest correlation ($R^2=0.50$, RMSE =1.90 kg C/m², 95% CI of [1.79, 2.01]). The superior model fit in the semi-humid and semi-arid regions is primarily characterized by consistently higher $R^2$ values. Although RMSE values vary, these transitional zones avoid the extreme uncertainty observed in the arid zone, suggesting that the model captures SOCD spatial variability more effectively in these areas. This performance is likely attributable to a more balanced distribution of SOCD sampling points, which mitigates the bias caused by extreme values. Furthermore, the environmental covariates selected for the model appear to exhibit a stronger and more consistent correlation with SOCD dynamics in semi-arid and semi-humid climates compared to other zones, thereby contributing to the improved estimation accuracy.

For the 0-100 cm depth, the semi-humid zone demonstrated the highest $R^2=0.68$ with RMSE = 6.29 kg C/m² (95% CI of [5.77, 6.76]). The semi-arid zone had an $R^2=0.64$ and RMSE = 5.92 kg C/m² (95% CI of [5.22, 6.58]). The arid zone showed an $R^2=0.61$ and RMSE = 5.71 kg C/m² (95% CI of [5.12, 6.32]). The humid zone had an $R^2=0.54$ and RMSE = 6.37 kg C/m² (95% CI of [5.13, 8.18]). Notably, although the semi-humid zone retained the highest $R^2$, the arid zone exhibited the lowest prediction error (RMSE) and a relatively narrow confidence interval. This suggests that while the model explains a moderate proportion of the total variance in arid regions, its predictive errors are tightly constrained. This phenomenon may be linked to the distinct environmental controls in drylands, where soil moisture is the dominant limiting factor. The high sensitivity of vegetation and soil microorganisms to water availability in arid zones creates strong, clear predictive signals that the model can easily capture. In contrast, in humid regions, SOC dynamics are governed by complex interactions among multiple factors—such as precipitation, temperature, and dense vegetation cover. These confounding influences can obscure direct relationships, thereby reducing the model's predictive power, particularly for deeper soil layers where long-term accumulation processes dominate. Our results confirm that the proposed modeling framework effectively captures these divergent mechanisms of SOC accumulation across different climatic regimes.

To elucidate the drivers of SOCD estimation, we analyzed the hierarchical importance of optimized environmental features (Fig. 8). The results reveal distinct driver mechanisms across soil depths. For the 0–20 cm depth (Fig. 8a), Temperature emerged as the dominant predictor, accounting for 35.91% of the model's contribution. It was followed by solar radiation (SR) (17.57%), Elevation (13.21%), NDVI (11.37%), Clay (8.99%), BSI (6.69%), Slope (3.70%), and CLCD (2.56%). The overarching influence of temperature on topsoil SOCD operates primarily through the regulation of microbial kinetics. While elevated temperatures accelerate the heterotrophic respiration and decomposition of SOC, they may simultaneously enhance plant productivity and residue turnover, thereby increasing carbon inputs. SR, as the second most important factor, acts in concert with temperature to drive the surface energy balance and potential evapotranspiration. These hydrothermal dynamics directly regulate soil water status: optimal moisture levels favor SOC accumulation by supporting vegetation growth, whereas moisture deficits induced by high radiation and evaporation can limit inputs or accelerate oxidative loss. The high ranking of NDVI further underscores the critical role of vegetation vitality, as it determines the quantity of organic litterfall and root exudates returned to the soil.

For the 0–100 cm depth (Fig. 8b), while Temperature remained the leading factor (25.62%), its relative dominance decreased compared to the surface layer. Notably, the importance of topographic and edaphic factors increased, with the hierarchy shifting to: Elevation (16.45%), SR (16.43%), NDVI (14.59%), Clay (12.44%), BSI (9.49%), Slope (3.32%), and CLCD (1.67%). The increased prominence of Elevation reflects its control over the vertical zonation of hydrothermal conditions (including radiation receipts), which fundamentally shapes soil formation processes and the depth-wise distribution of organic carbon.

The shift in feature importance between the two models highlights the complexity of SOC dynamics and the necessity of depth-specific modeling. In the 0–20 cm model, bioclimatic factors (Temperature, SR, and NDVI) exert a more pronounced influence, reflecting the direct sensitivity of topsoil to atmospheric energy inputs and immediate vegetation exchange. In contrast, the 0–100 cm model shows a marked increase in the contribution of soil physicochemical properties (e.g., Clay) and stable topographic features (e.g., Elevation, Slope). This suggests that SOCD in deeper layers is increasingly governed by long-term pedogenic processes, geological context, and depositional dynamics rather than immediate surface accumulation. Incorporating these depth-dependent determinants—spanning climatic, topographic, biological, and edaphic variables—is essential for accurately capturing the spatial and temporal heterogeneity of soil organic carbon stocks.

### 4.3 Validation with independent sample points

The SOCD estimation result is validated with independent published SOCD data in the Heihe River basin by Li et al. (2022). The Heihe River basin is a major ecological and agricultural zone in northwest China. There are special geographical and climatic characteristics for the soil carbon accumulation in the Heihe River basin, which are important for exploring soil quality in arid and semi-arid zones. Validation is carried out by comparing measured data in the Heihe River basin with the estimated SOCD in this study. The comparison results show that our estimated SOCD is highly consistent with the measured SOC data from the Heihe River basin (Fig. 9). The estimated SOCD and the measured SOCD have a significant correlation, which is shown by the $R^2$ value of 0.76, and the RMSE value of 1.75 (kg C/m²) (95% CI of [1.25, 2.23]) for the estimated result with the depth of 0-20 cm. Additionally, the proposed model demonstrates superior accuracy compared to Li's dataset, which reported an $R^2$ of 0.60 and an RMSE of 2.27 (kg C/m²) (95% CI of [1.59, 2.98]).

To further assess model robustness across a wider range of environmental conditions, our estimated SOCD results are compared with the data published data by Xu et al. (2018), which is the data on carbon storage of terrestrial ecosystems in
China with a depth of 0-20 cm These samples are widely distributed across the southern Tibet Autonomous Region, Qinghai Province, and eastern Inner Mongolia Autonomous Region. This is very good evidence for validating the robustness, reliability, and generalizability of the SOCD estimation model in this study. The estimated SOCD results are compared with the measured SOC data in the field campaign of Xu et al. (2018). In addition, the field data were compared with 0-20 cm organic carbon density maps generated by a machine learning analysis dataset of SOC dynamics and their drivers in China
during 2000-2014. The results of the comparative analysis are encouraging and show high agreement between the estimated SOCD using our developed model and the measured SOC data. Specifically, the $R^2$ value is 0.68 and the RMSE value is 1.70 (kg C/m²) [95%CI: 1.51, 1.90], which further confirms the accuracy of our SOCD estimation model (Fig. 9). Furthermore, the model outperforms Li's dataset, which yielded an $R^2$ of 0.54 and an RMSE of 2.04 (kg C/m²) [95%CI: 1.73, 2.34], underscoring the enhanced predictive accuracy of our approach.

Our estimated SOCD results with the depth of 0-100 cm were validated furtherly with independent measurements from Inorganic carbon pools and their drivers in grassland and desert soils (Dong et al., 2024) and compared with the machine learning-derived SOCD simulations by Li et al. (2022). The SOCD dataset of Dong et al. (2024), covering grassland and desert ecosystems across China, provides robust in-situ measurements of SOCD (0-100 cm) alongside critical environmental drivers (e.g., climate, soil properties), offering an ideal benchmark for evaluating model generalizability in arid and semi-
arid regions. Comparative analysis revealed that our SOCD estimates achieved significantly better agreement with the independent validation data ($R^2 = 0.50$, RMSE = 4.93 kg C/m²) [95%CI: 1.65, 7.47] than Li et al.'s simulations ($R^2 = 0.31$, RMSE = 5.80 kg C/m²) [95%CI: 4.16, 7.48] (Fig. 9). This demonstrates the superior accuracy of our approach in capturing deep soil carbon dynamics (0-100 cm), particularly in heterogeneous grassland and desert environments. The higher $R^2$ and lower RMSE values underscore the improved capability of our model to resolve spatial patterns of SOC storage compared to
earlier machine learning-based efforts.

## 4.4 Comparison with published SOCD products

The 1-km-resolution SOCD dataset of China is produced in this study, which is compared with the published SOCD products including HWSD v2.0, SoilGrids 250m, and GSOCmap datasets to validate and confirm its accuracy and reliability. As shown in Fig. 10, our estimated SOCD dataset exhibits strong consistency with published products, with the highest agreement observed with SoilGrids 250 m ($R^2$ = 0.74). This performance is substantially better than the correlations with GSOCmap ($R^2$ = 0.64) and HWSD ($R^2$ = 0.71). The HWSD v2.0 dataset is jointly published by the Food and Agriculture Organization of the United Nations (FAO) and the International Institute for Applied Systems (IIAS) in Vienna, which provides soil data on a global scale. The correlation of our SOCD dataset with HWSD is reported with the $R^2$ value of 0.71 and the RMSE value of 11.52 (kg C/m$^2$). The GSOCmap dataset is led by the FAO and is intended to cover various ecosystems around the world. This is the first global SOC map. The correlation of our SOCD dataset with GSOCmap is reported with the $R^2$ of 0.64 and the RMSE of 1.25 (kg C/m$^2$). The SoilGrids250m dataset is created using ISRIC's digital soil mapping technology, which is a global soil dataset. The correlation of our SOCD dataset with SoilGrids250m is reported with the $R^2$ of 0.74 and the RMSE of 1.03 (kg C/m$^2$). Models are more accurate and applicable than global soil databases in capturing SOCD changes in China. This study highlights the need to create and implement region-specific models that utilize current geographic and environmental data to provide a more precise tool for accurately estimating soil carbon reserves.

To evaluate the temporal accuracy of our product, we compared the SOCD estimates for the 1980s, 2000s, and 2010s against the corresponding SOC Dynamics ML dataset (Fig. 11). The results demonstrate a strong and consistent agreement between the estimated and measured values across all periods. Specifically, the model achieved $R^2$ values of 0.78, 0.76, and 0.73, with corresponding RMSEs of 0.90, 1.02, and 1.07 kg C/m$^2$ for the 1980s, 2000s, and 2010s, respectively. While the $R^2$ shows slight fluctuations, the consistently low RMSE values indicate that the model remains robust over time. Overall, these comparisons validate the reliability of the SOCD estimation framework developed in this study. The sustained accuracy over three decades highlights the model's capability to provide precise long-term SOCD estimates, underscoring the importance of integrating multi-temporal field data with advanced analytical methods.

## 4.5 Spatiotemporal changing of SOCD in China

The SOCD changes over time from the 1980s to the 2010s are validated in Fig. 12 compared with the published investigations. Fig. 12 reveals that our estimated SOCD results with depths of 0-20 cm (a) and 0-100 cm (b) are falling in the value range of the previous investigations of Xie et al. ( 2007), Wu et al. (2003), Wang et al. (2004), Xu et al. (2018), Wang et al. (2021), Li et al. (2022), Zhang et al. (2023). A slight increasing trend in SOCD was observed in the 0–20 cm topsoil from the 1980s to the 2010s (Fig. 12a). This resulted from that the topsoil is more susceptible to the direct effects of soil

management practices and environmental changes (Oechaiyaphum et al., 2020). In contrast, the estimated SOCD in the 0–100 cm profile remained relatively stable throughout the study period (Fig. 12b). Fig. 13 and Fig. 14 show the spatiotemporal distributions of the estimated SOCD at the 5-year interval from the 1980s to the 2010s. And the regions with high SOCD value in depth of 0-20 cm are in northeast and southwest China with red color in Fig. 13. Comparably speaking, there are the largest area of high SOCD value labeled dark red color bar in period of 2010-2015 (Fig. 13f). From the

perspective of longitude, the SOCD distribution shows different pattern, and it is homogeneous in high and low longitudes where the land cover is forest mostly. Conversely, the variance of SOCD is higher in mid-longitude regions where is with distinct land cover types. Similarly, the regions with high SOCD value in depth of 0-100 cm are in northeast and southwest China with green color in Fig. 14. And there are smaller variance of SOCD in high and low longitudes, and there are higher variance of SOCD in mid-longitude regions. There are many driving factors for the changing of SOCD in China. Targeted

monitoring and management practices should be implemented for SOCD trends at different soil depths to maximize soil carbon sequestration and continuously improve soil quality.

## 5 Data availability

The 1-km soil organic carbon density dataset for China at depths of 0–20 cm and 0–100 cm from 1985 to 2020 is freely avail able at https://doi.org/10.6084/m9.figshare.27290310.v2(Dong et al., 2024). The dataset can be imported into remote sensing

processing software (e.g., ENVI), standard geographical information system software (e.g., ArcGIS). In addition, the original CSV-format field measurement dataset used in this study is provided to enhance transparency, reproducibility, and facilitate further applications of the SOCD dataset.

## 6 Conclusions

In this study, a SOCD dataset with a resolution of 1-km resolution and soil depths of 0-20 cm and 0-100 cm is created from 1985 to 2020 in China. The accuracy and validity of this dataset are validated by three independent metrics and data and four types of published global products. The conclusions are as follows.

(1) The delineation of climatic zones for SOCD estimation modeling has been proven useful for enhancing the precision of the models and effectively addressing the uneven distribution of measured SOC.

(2) Independent validations confirmed the robustness of the estimated SOCD. For the 0-20 cm depth, our estimates showed strong agreement with measured data from the Heihe River basin ($R^2$=0.76, RMSE=1.75 kg C/m²) and the Xu dataset ($R^2$=0.68, RMSE=1.70 kg C/m²). Furthermore, for the 0-100 cm depth, validation against independent measurements from Dong et al. (2024) also indicated strong agreement ($R^2$=0.50 RMSE=4.93 kg C/m²).

(3) Compared to published global products including HWSD, SoilGrids250m, and GSOCmap, the estimated SOCD in this study was consistent and accurate. Comparison with the SoilGrids250m dataset shows the superiority of zoning RF models in capturing variations in SOCD in China with $R^2$=0.74 and RMSE=1.03 (kg C/m²).

(4) Temporal evaluations showed good agreement between our SOCD estimates and independent measurements from the 1980s, 2000s, and 2010s. The time-series analysis revealed clear SOCD variations across China and across soil depths, reflecting the influence of agricultural management, land-use changes, and climate variability.

While this dataset represents a significant advancement in national-scale SOC accounting, the continuous integration of new soil profile data remains essential for further model refinement. Future research should prioritize quantifying the impacts of specific land management strategies on SOC dynamics. Furthermore, given the persistent uncertainties in large-scale soil carbon estimates, we advocate for standardized sampling protocols, broader data sharing, and strengthened global collaboration to improve the accuracy of future soil carbon inventories.

**Author contributions**

Yi Dong and Xinting Wang collected the SOC data, did the field campaign and SOCD estimation, performed the analysis, and wrote the paper; Wei Su designed the research and revised the manuscript.

**Competing interests**

The authors declare that they have no conflict of interest.

**Acknowledgments**

This study was supported by the National Natural Science Foundation of China under the project (No. 42471402), the Intergovernmental International Scientific and Technological Innovation Cooperation Project of National Key R&D Program (No. 2025YFE0102000), and Beijing Natural Science Foundation (L251053).

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

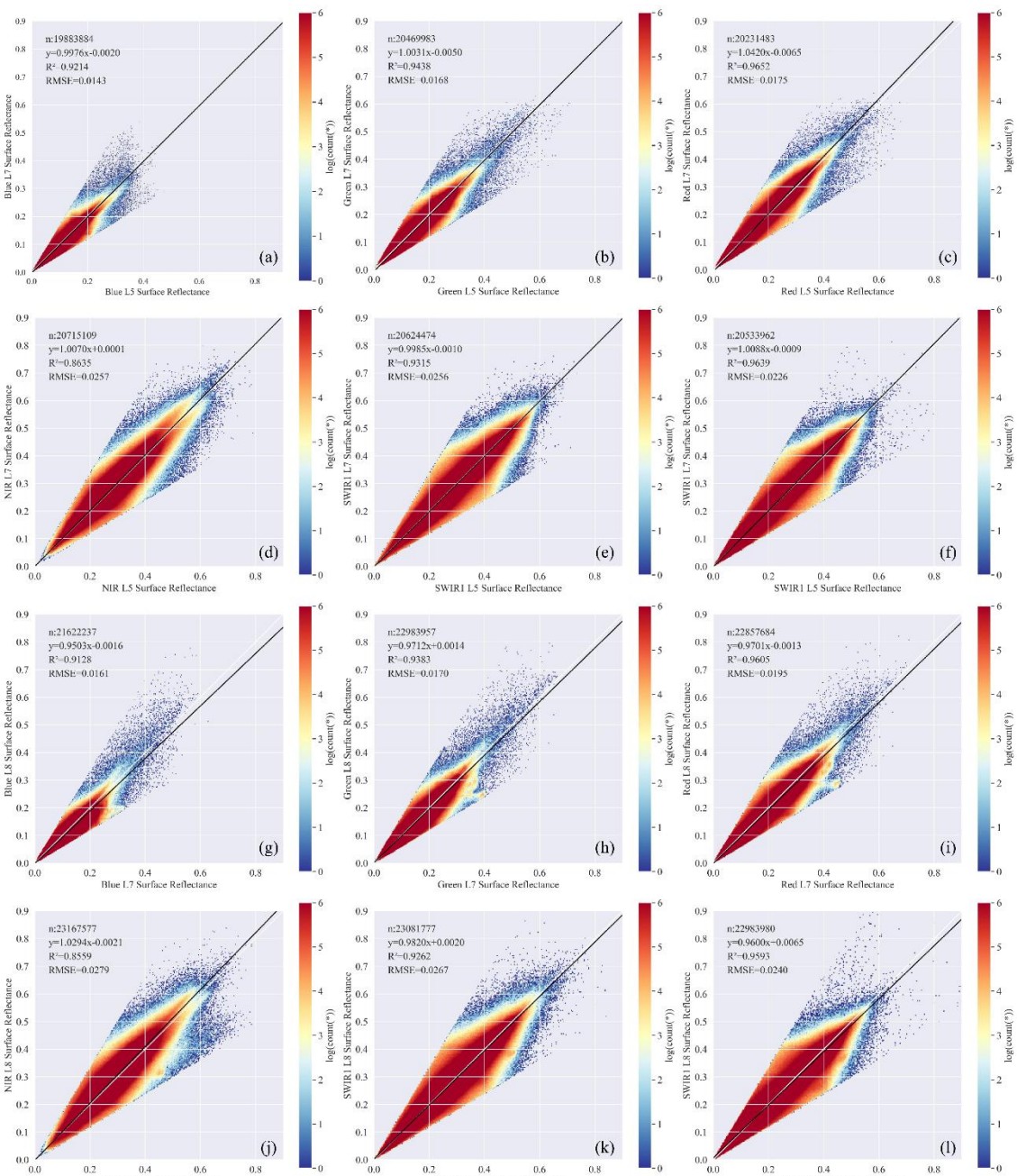

**Figure 1.** Radiometric normalization coefficients between Landsat 5 TM、Landsat 7 ETM+ (a-f) and Landsat 7 ETM+、Landsat 8 OLI (e-j) sensors for different bands including blue, green, red, NIR, SWIR1, and SWIR2. The radiometric normalization coefficients for each sensor are represented by fitted lines and correlation coefficients, indicating the correlation between the reference of different sensors, and characterizing the spectral response of the sensors in the different wavelength bands.

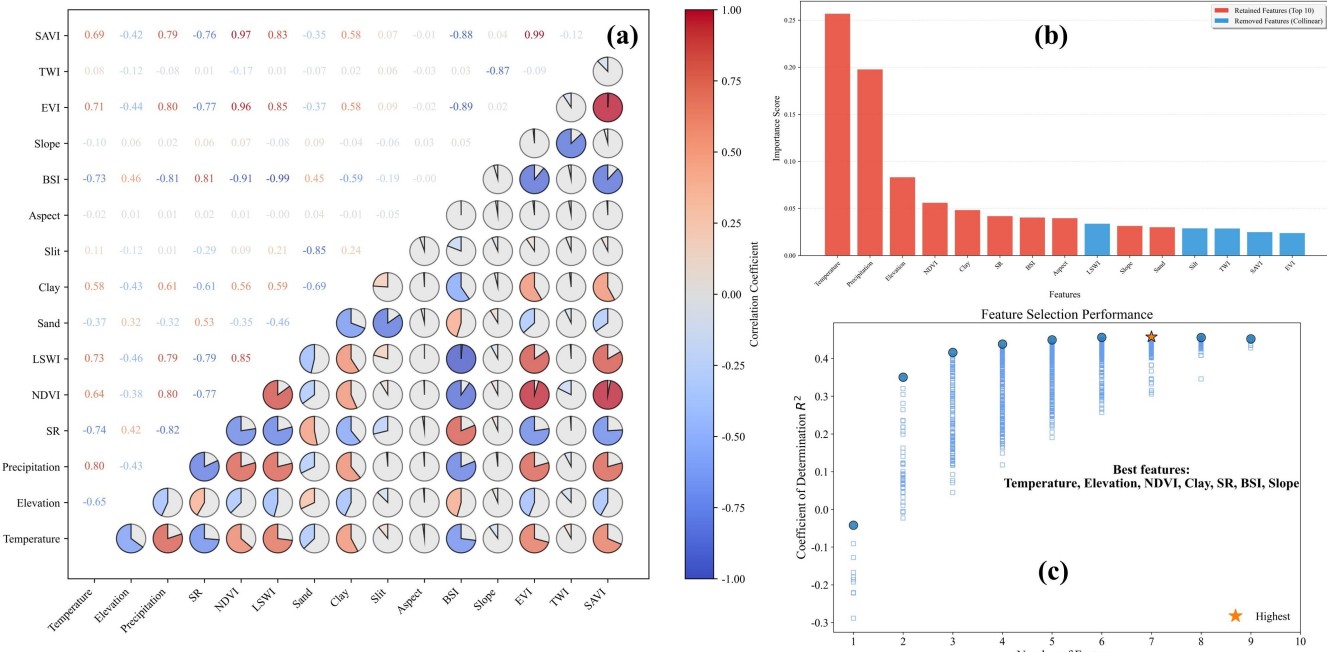


**Figure 2.** Feature selection process for SOCD estimation. (a) Pearson correlation matrix of top environmental covariates (upper triangle shows correlation coefficients; red=positive, blue=negative), with boxed features indicating the final selected variables. (b) Hierarchical feature importance evaluation combining correlation filtering (removing $|r| > 0.95$), random forest-based ranking (Gini importance), and combinatorial optimization. (c)The optimal feature set (highlighted in bold)

comprised seven variables: mean annual temperature, elevation, NDVI, clay content (Clay), Solar Radiation (SR), bare soil index (BS1), and slope, which collectively maximize prediction accuracy ($R^2$) while maintaining ecological interpretability.

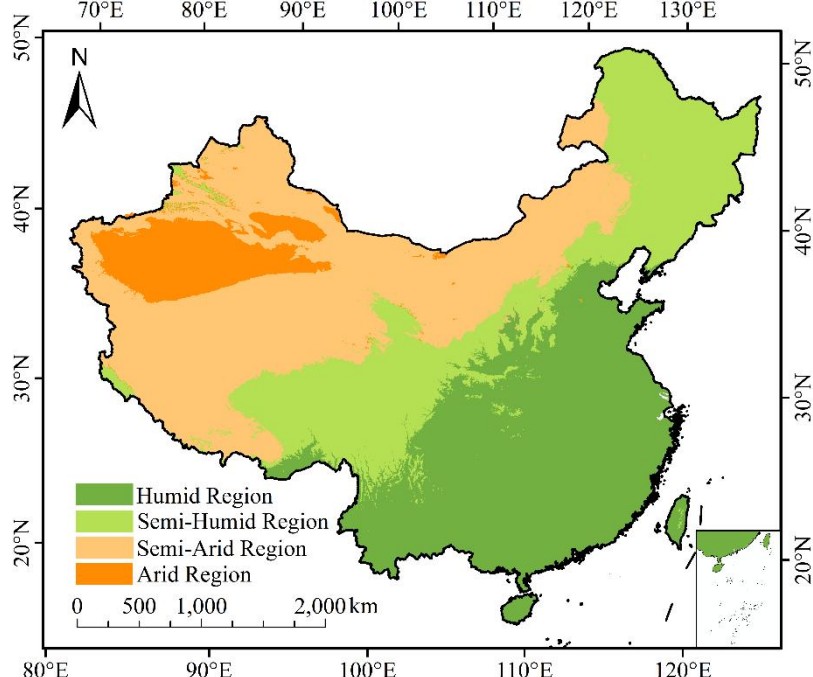

**Figure 3.** Climatic zones for SOCD estimation modeling. Climate zoning comes from the time-series climate data including temperature

and precipitation. According to the difference in climate zones, it can be divided into humid, semi-humid, arid, and semi-arid zones.


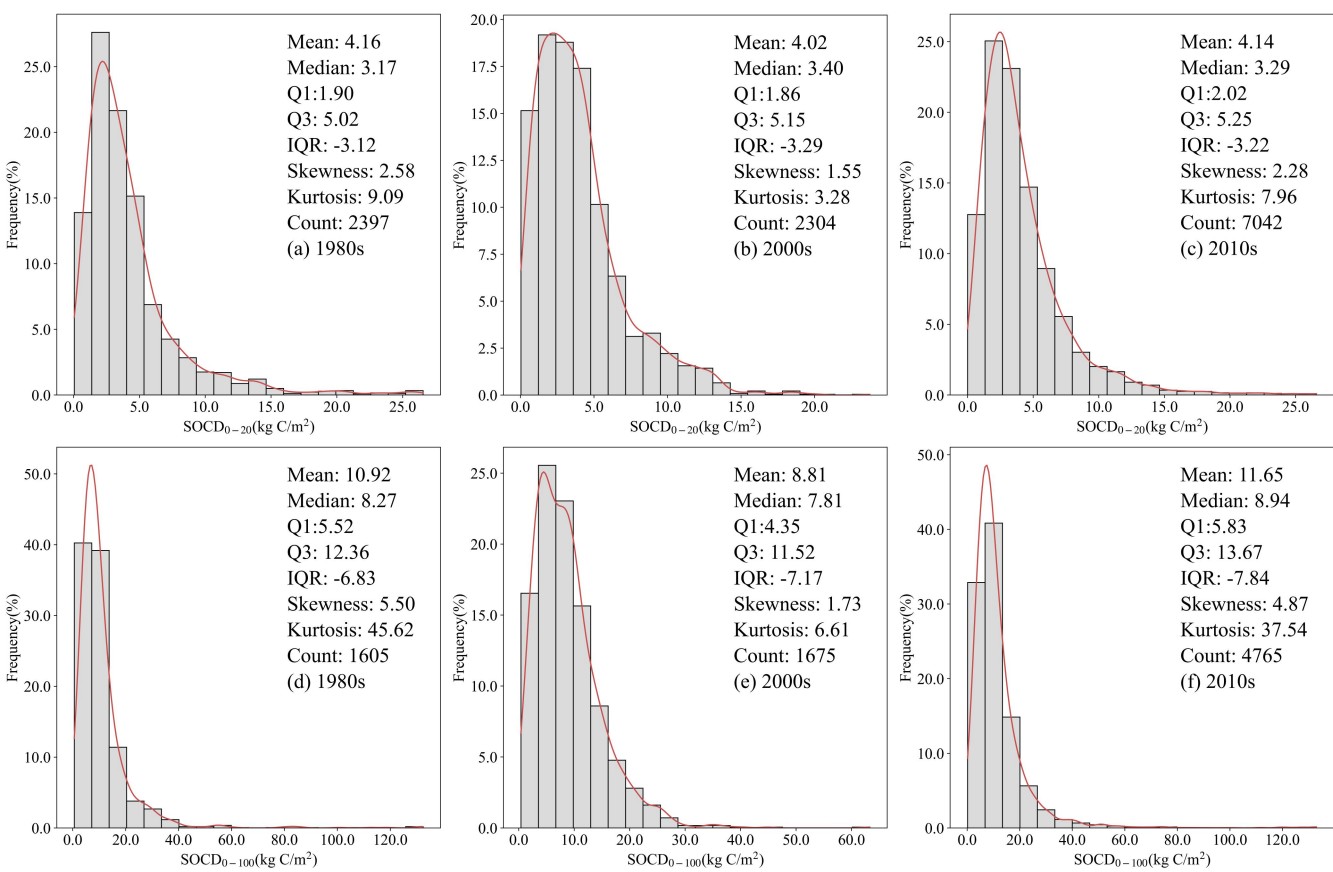

**Figure 4.** Statistical characteristics of soil sample points in different periods. Frequency distribution of SOCD data with the soil depth of

0-20 cm (a-c) and 0-100 cm (d-f) during the 1980s, 2000s, and 2010s.

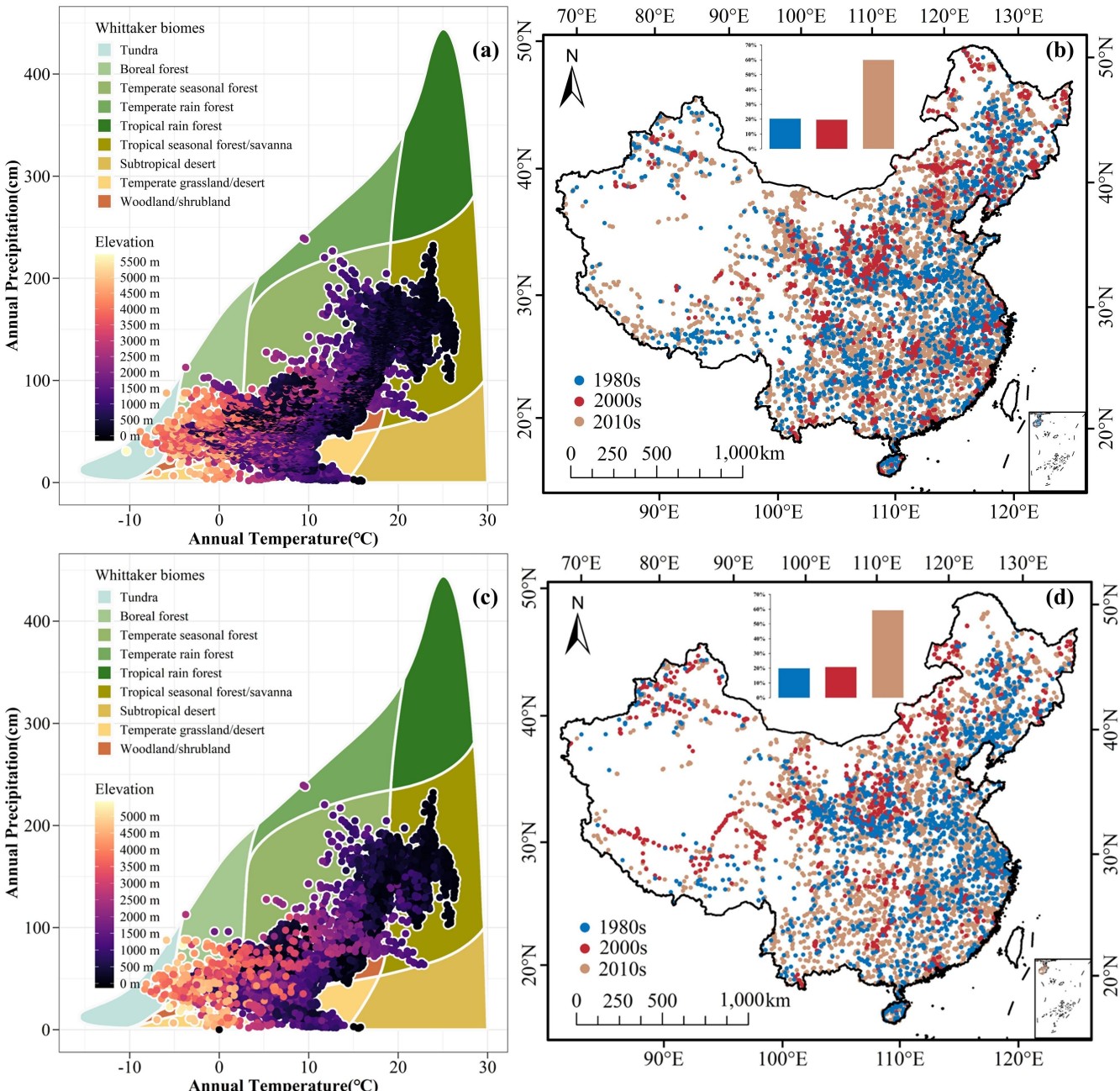

**Figure 5.** Spatial distribution of SOC soil sample points with depth of 0-20 cm (b) and 0-100 cm (d). And the Whittaker biomes of soil sample points with depth of 0-20 cm and 0-100 cm are shown in (a) and (c).

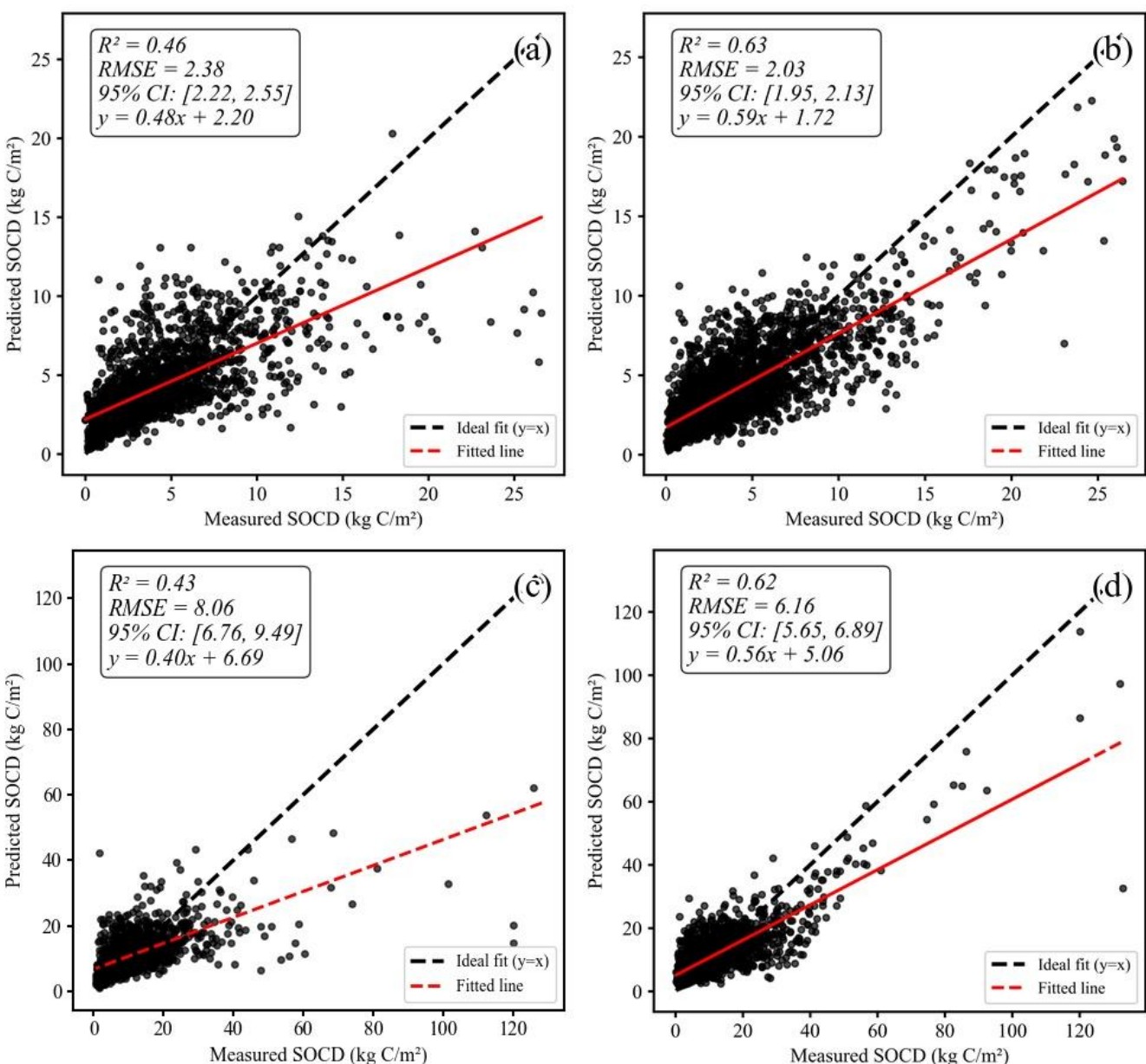

**Figure 6.** The model performance of global and zoning models with the depth of 0-20 cm and 0-100 cm. The SOCD prediction model of 0-20 cm and 0-100 cm soil depth is evaluated strictly by using a variety of statistical indicators, corresponding to four evaluation results, 0-20 cm global model (a), 0-20 cm regional model (b), 0-100 cm global model (c), and 0-100 cm regional model (d).

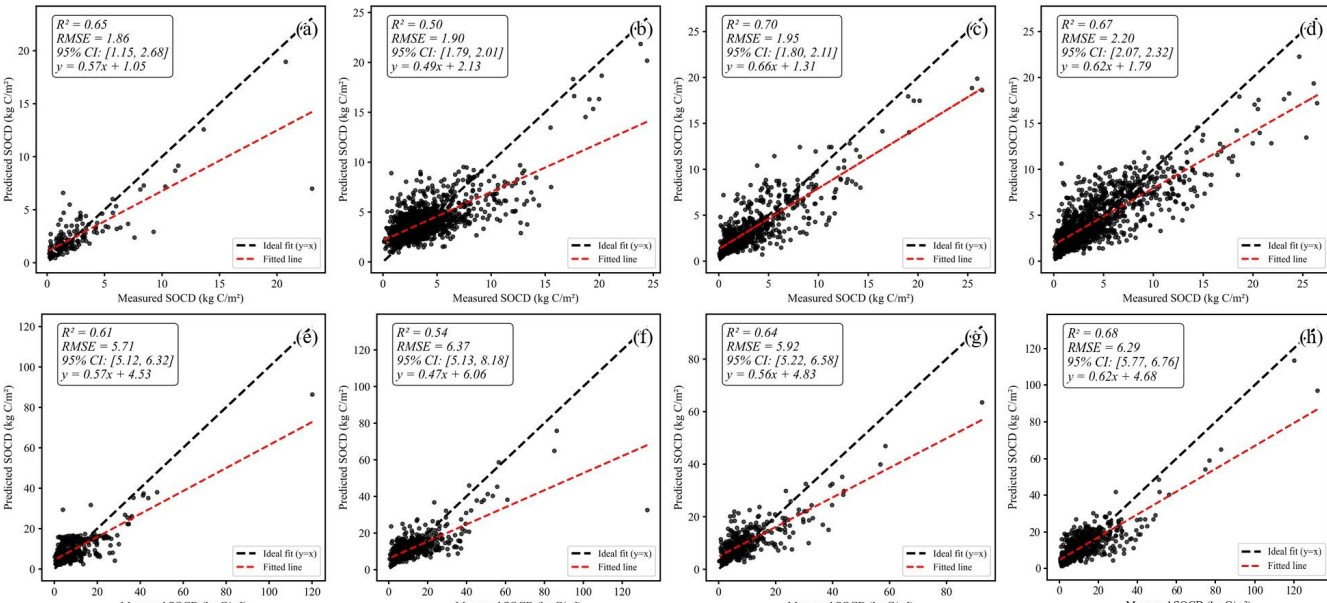

**Figure 7.** The model performance of different zoning models with the depth of 0-20 cm and 0-100 cm. Panels (a) and (e) depict the model performance for arid regions. Panels (b) and (f) illustrate results for humid regions. Panels (c) and (g) showcase semi-arid regions. Finally, panels (d) and (h) display model accuracy in semi-humid regions.

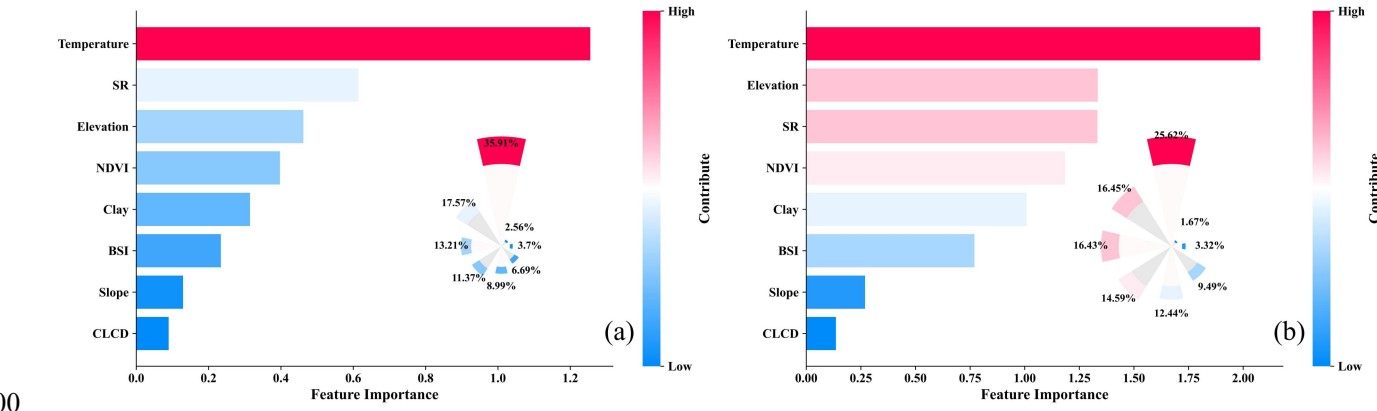

**Figure 8.** Importance ranking of features for SOCD estimation with the depth of 0-20 cm and 0-100 cm. It reports the contribution of different environmental variables to the SOCD estimation with different soil depths, including feature importance ranking for 0-20 cm depth (a) and feature importance ranking for 0-100 cm depth (b).


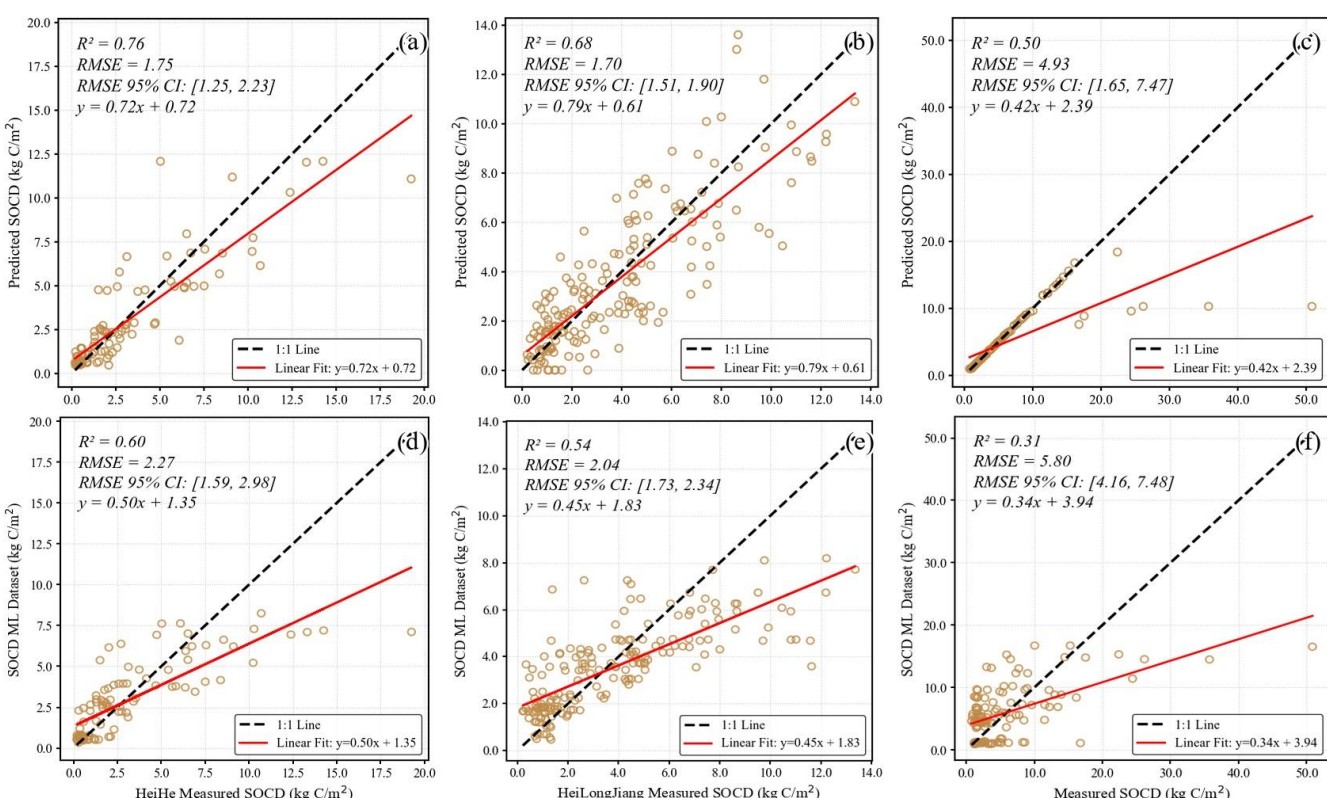

**Figure 9.** Comparison of predicted and machine learning (ML) derived SOCD with independent measurements at various depths. Panels (a), (b), and (c) display correlations of this study's predicted SOCD, while (d), (e), and (f) show correlations of SOC Dynamics ML dataset. Specifically, (a) and (d) are for 0-20 cm SOCD against Heihe River basin measurements. (b) and (e) compare 0-20 cm SOCD with Xu's published data. (c) and (f) present 0-100 cm SOCD correlations with measurements from Dong et al. (2024) and simulations from Li et al. (2022).

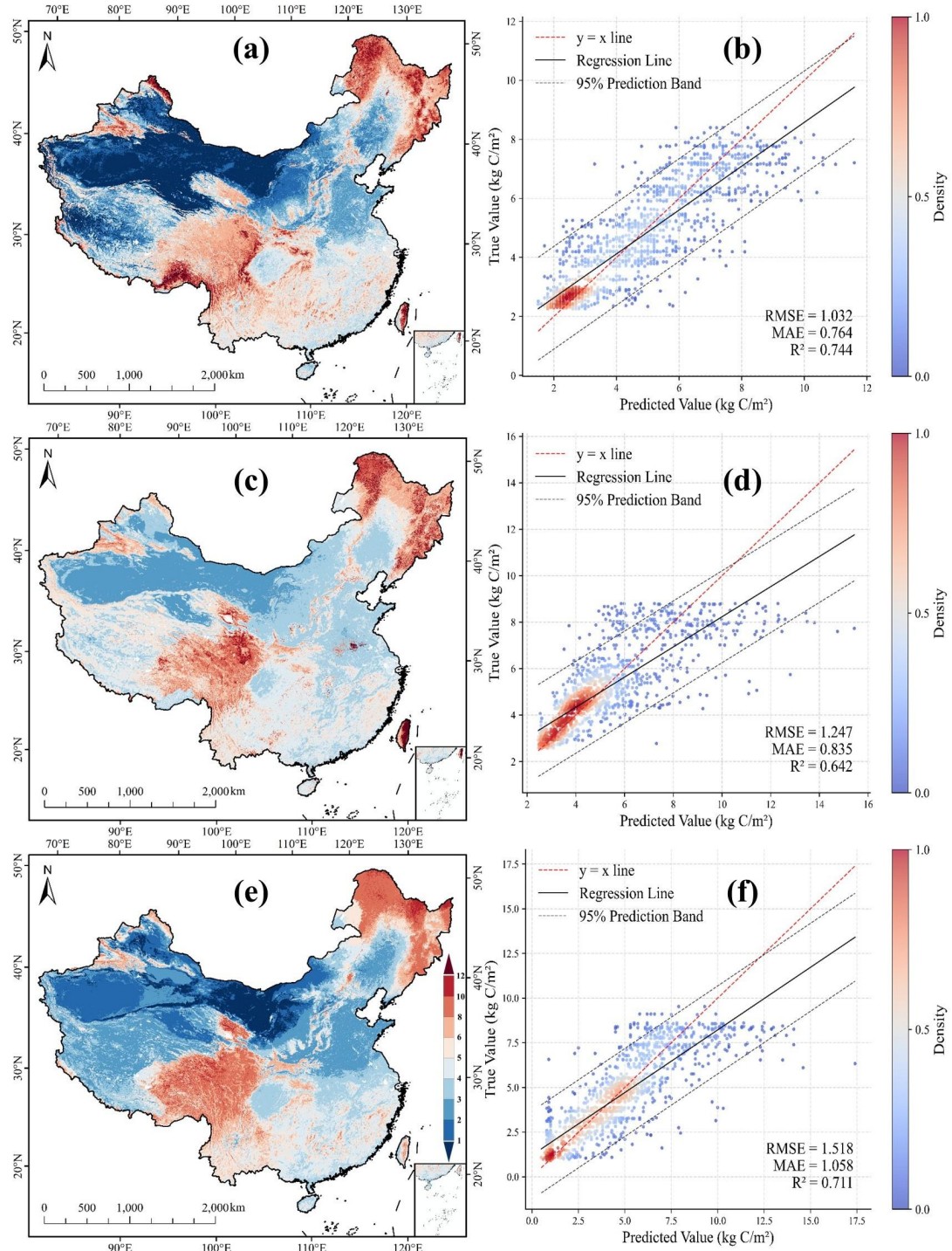

**Figure 10.** Comparison with three published global products. Our estimated SOCD is compared with the SoilGrids250m (a & b), GSOCmap (c & d), and HWSD v2.0 (e & f) datasets.

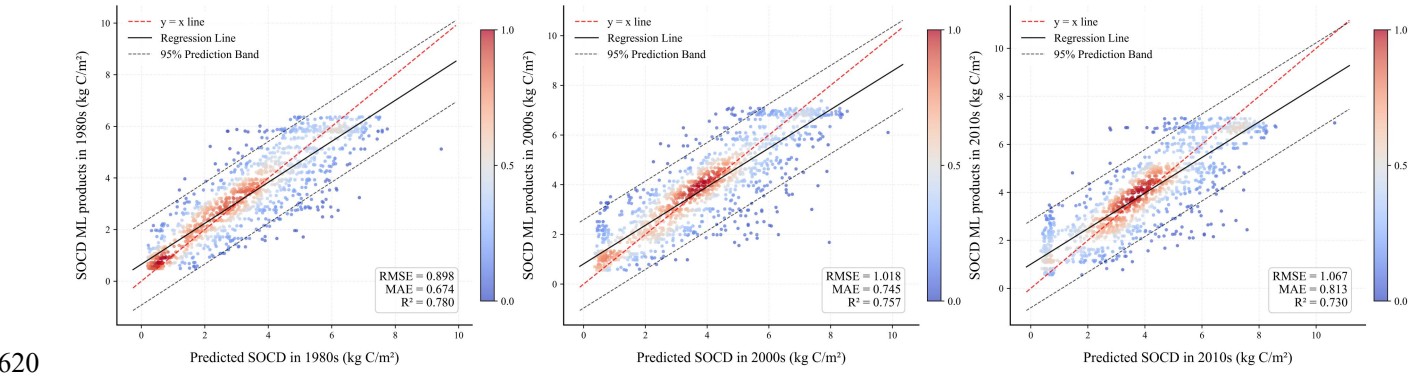


**Figure 11.** Comparison with the SOC Dynamics ML dataset with a depth of 0-20 cm in China in the 1980s (a), 2000s (b), and 2010s (c).

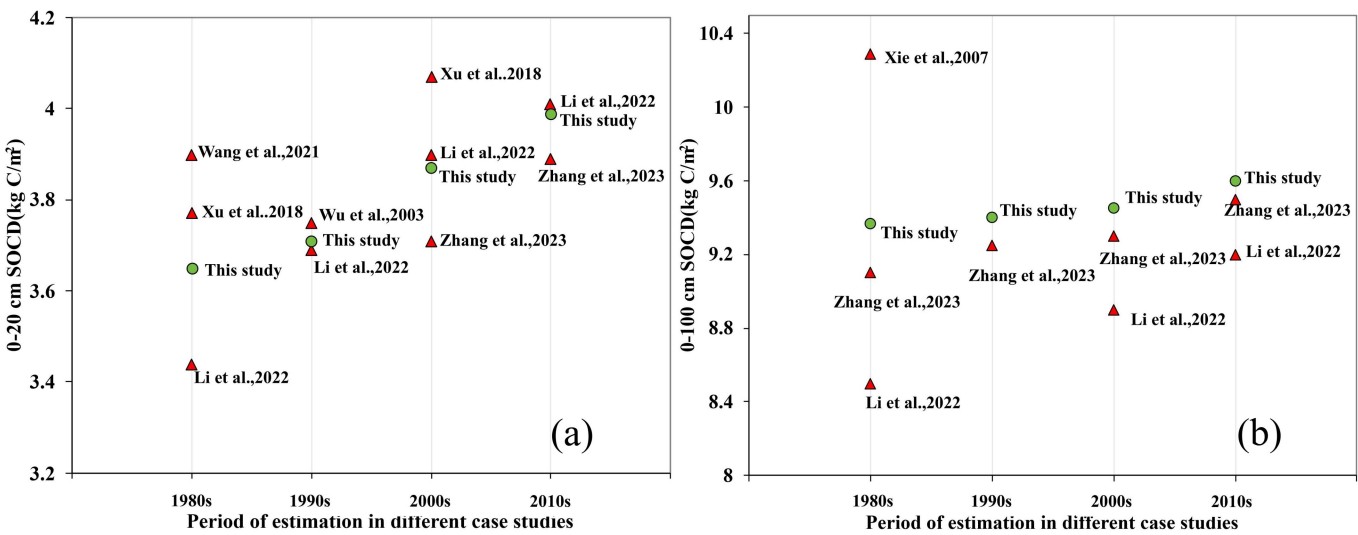

**Figure 12.** Aggregated results of estimated SOCD with the depth of 0-20 cm (a) and 0-100 cm (b) in China from this study and previous investigations.

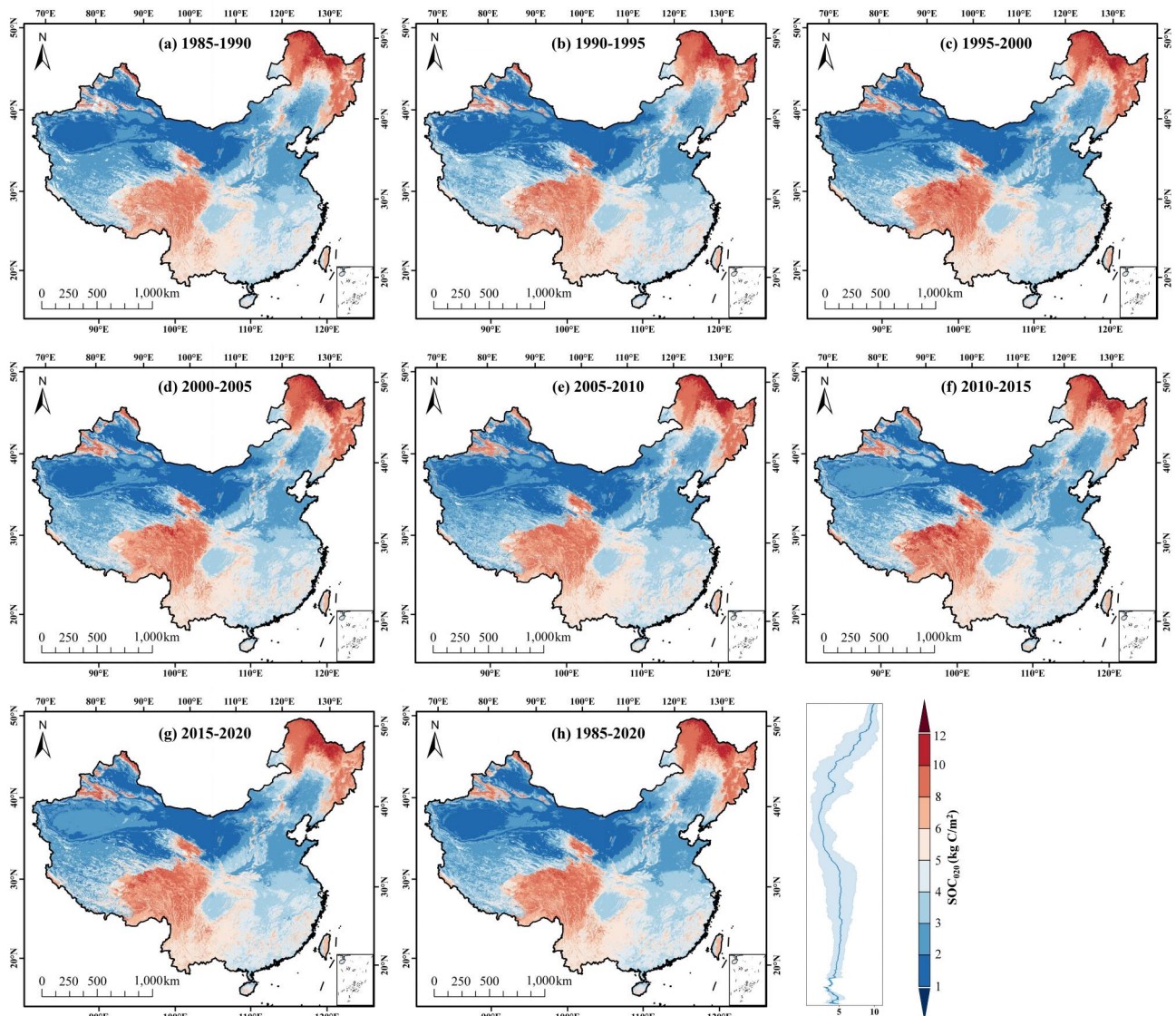

**Figure 13.** Spatial distribution of estimated SOCD at a depth of 0-20 cm in 1985-1990 (a), 1990-1995 (b), 1995-2000(c), 2000-2005 (d), 2005-2010 (e), 2010-2015 (f), 2015-2020 (g) and average from 1985 to 2020 (h). The lower left histograms in each panel show the area ratios for different SOCD levels.

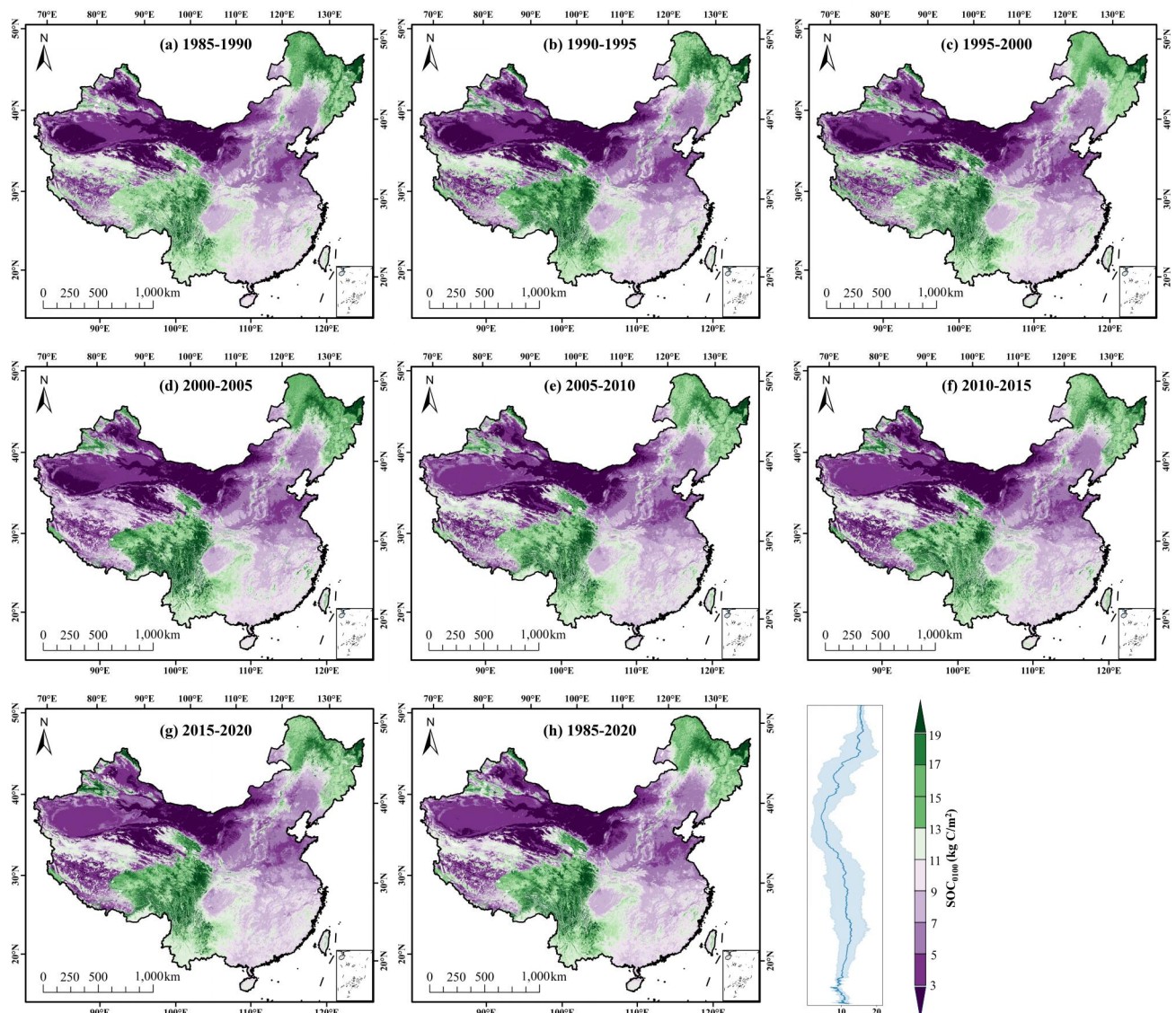

**Figure 14.** Spatial distribution of estimated SOCD at a depth of 0-100 cm in 1985-1990 (a), 1990-1995 (b), 1995-2000(c), 2000-2005 (d), 2005-2010 (e), 2010-2015 (f), and 2015-2020 (g) and average from 1985 to 2020 (h). The lower left histograms in each panel show the area ratios for different SOCD levels.