# Peer review of "A 1 km soil organic carbon density dataset with depth of 20cm and 100cm from 1985 to 2020 in China"

_Earth System Science Data, 2024_

## Author Comment (AC1)

**Section 3.2. Could you give more explanation about the principles of selecting variables? For example, from Fig.2, the *R* between AH and SOCD is almost 0, why select this variable? And only 18 variables have been shown on Fig. 2 without CLCU, how to select CLCU as an input predictor?**

We sincerely appreciate these insightful questions about our feature selection process. In our original methodology, the variable selection followed these principles:

The initial variable selection in our methodology followed a rigorous procedure. First, we established a comprehensive candidate pool comprising 19 environmental variables across four categories: climatic factors (e.g., temperature and precipitation), topographic attributes (elevation, slope, aspect), vegetation indices (NDVI, EVI), and soil properties (clay and sand content). Subsequently, correlation-based screening was applied to retain variables significantly associated with soil organic carbon density (SOCD) ($p < 0.05$) and exhibiting at least a minimal linear relationship (absolute Pearson's $r > 0.1$). Two exceptions were made, anthropogenic heat (AH) was retained due to its potential interactive effects in specific climatic regimes, and land cover type (CLCD) was included based on its well-established ecological relevance in prior literature, despite their weaker correlations with SOCD. Finally, to mitigate multicollinearity, variables with pairwise correlations exceeding 0.8 (absolute value) were eliminated, prioritizing those with clearer physical or mechanistic interpretations.

It is worth noting that, as the reviewer astutely observed, AH indeed exhibited a weak initial correlation with SOCD. Although AH and CLCD were considered based on the aforementioned reasons in the initial stages, during the final model construction and feature importance evaluation, these variables demonstrated low actual predictive contribution. Therefore, they were ultimately excluded from the core variable set used for modeling to ensure model parsimony and predictive efficacy.

Upon careful consideration of the reviewers' comments, we have significantly refined our feature selection approach. We implemented an enhanced feature selection methodology for SOCD prediction. The refined approach begins with initial screening through Pearson correlation analysis ($p < 0.05$ significance threshold), followed by Random Forest-based importance ranking to evaluate non-linear relationships. Subsequently, we conducted exhaustive combinatorial optimization of all possible feature combinations to maximize predictive performance ($R^2$). Key methodological improvements include: (1) removal of marginally contributing variables (AH, CLCD) with limited predictive value; (2) incorporation of spectral indices (SR, BSI) to better characterize vegetation-soil interactions; and (3) implementation of stricter redundancy thresholds ($|r| > 0.95$) to further minimize multicollinearity. The final optimized feature set comprises 'Tem',

'Elevation', 'NDVI', 'Clay', 'SR', 'BSI', and 'Slope', representing a balanced combination of climatic, topographic, vegetation, and soil properties. This rigorous multi-stage approach effectively integrates statistical correlation analysis with machine learning-based feature importance assessment, ensuring optimal variable selection while maintaining ecological interpretability.

The methodological refinements have been systematically incorporated throughout the manuscript. Section 3.2 Feature optimization for RF modelling has been comprehensively revised to detail the improved approach, with particular emphasis on the integration of machine learning-based importance assessment. Figure 2 has been updated to visually present the final selected feature set and their relative importance scores.

[Figure]

**Figure 2.** Feature selection process for predicting soil organic carbon density (SOCD). (a) Pearson correlation matrix of top environmental covariates (upper triangle shows correlation coefficients; red=positive, blue=negative), with boxed features indicating the final selected variables. (b) Hierarchical feature importance evaluation combining correlation filtering (removing |r| > 0.95), random forest-based ranking (Gini importance), and combinatorial optimization. The optimal feature set (highlighted in bold) comprised seven variables: mean annual temperature (Tem), elevation, NDVI, clay content (Clay), simple ratio index (SR), bare soil index (BS1), and slope, which collectively maximize prediction accuracy ($R^2$) while maintaining ecological interpretability.

**Section 4.2. Fig. 8 and Line 250: The discussion of different features for SOCD estimations is comprehensive, which can help us to understand the important factors of SOCD variations. But it's very interesting to find that the features have different important values in the two depth models. Please try to discuss more about these differences.**

[Figure]

We greatly appreciate the reviewer's valuable observations regarding the distinct patterns of feature importance between our 0-20 cm and 0-100 cm depth models. These differences provide important insights into the depth-dependent mechanisms controlling soil organic carbon (SOC) distribution and accumulation.

The comparative analysis reveals fundamental differences in how environmental factors influence SOC at different soil depths. In the surface layer (0-20 cm), climate variables (temperature and precipitation) demonstrate particularly strong predictive power, reflecting their direct control over biological processes that govern surface carbon cycling. The vegetation index (NDVI) also shows greater importance in this shallow layer, consistent with its role as a proxy for organic matter inputs through plant litter and root exudates. These patterns collectively highlight the dominance of contemporary biological processes in surface SOC dynamics.

In contrast, the full profile model (0-100 cm) shows relatively reduced importance of climatic and vegetation factors, while soil texture parameters (particularly clay content) and topographic features gain significance. This shift reflects the transition from biologically-dominated surface processes to the more complex interplay of geochemical and physical mechanisms that control SOC stabilization and transport in deeper layers. The enhanced role of terrain attributes in the deeper model suggests the importance of long-term pedogenic processes and landscape-scale carbon redistribution through erosion and deposition.

Land use/cover (CLCD) patterns exhibit particularly interesting depth-dependent behavior, maintaining strong predictive power in the surface model but showing reduced importance in the full profile assessment. This pattern likely reflects both the direct impact of land management on surface carbon inputs and the time-lagged nature of subsurface carbon responses to land use changes. The differential behavior of soil texture parameters - with clay content becoming increasingly important with depth while sand content shows opposite trends - further emphasizes the depth-specific mechanisms of carbon stabilization and loss.

These findings have significant implications for SOC modeling approaches. The clear divergence in controlling factors between depth layers underscores the necessity of depth-stratified modeling frameworks that can adequately represent these distinct regulatory mechanisms. Our results suggest that surface SOC models should prioritize climatic and vegetation parameters, while full-profile assessments require greater emphasis on soil forming factors and landscape

position. This improved understanding of depth-specific SOC controls not only enhances predictive capability but also provides mechanistic insights for targeted carbon management strategies across different soil layers.

We have expanded the discussion of these concepts in the revised manuscript (Section 4.2), incorporating additional references to support our interpretation of these depth-dependent patterns. The analysis provides valuable evidence that the relative importance of environmental predictors in SOC models fundamentally depends on the soil depth being considered, reflecting the vertical stratification of processes that govern carbon accumulation and stabilization in terrestrial ecosystems.

The current results of feature selection.

In analyzing soil organic carbon density (SOCD), the importance of different features varies significantly across soil layers of different depths, which is crucial for understanding the mechanisms of SOCD variation.

In the 0-20 cm soil layer, temperature (Tem) is the most important feature, accounting for 34.41%, indicating that temperature has the greatest impact on SOCD, likely because it directly affects microbial activity and the rate of organic matter decomposition. NDVI (Normalized Difference Vegetation Index) is 20.3% important, solar radiation (SR) is 16.96%, elevation (Elevation) is 12.02%, soil brightness index (BSI) is 6.92%, clay (Clay) is 5.2%, and slope (Slope) is 4.19%.

In contrast, in the 0-100 cm soil layer, NDVI becomes the most important feature, accounting for 34.41%, indicating that vegetation cover has the greatest impact on SOCD. Temperature is 20.3% important, elevation is 17.78%, solar radiation is 8.55%, clay is 7.11%, slope is 6.28%, and soil brightness index (BSI) is 4.38%.

These differences indicate that different soil layers have different influencing factors on SOCD, with temperature and vegetation cover being more important in shallower layers, while vegetation cover and elevation have a more significant impact in deeper layers. These findings help us better understand the mechanisms of SOCD variation and provide a scientific basis for soil management and carbon sequestration.

[Figure]

**Figure 8.** Importance ranking of features for SOCD estimation with the depth of 0-20 cm and 0-100 cm. It reports the contribution of different environmental variables to the SOCD estimation with different soil depths, including feature importance ranking for 0-20 cm depth (a) and feature importance ranking for 0-100 cm depth (b).

Section 4.5. Fig. 13 and Line 315: "This may be the result of the topsoil being more susceptible to the direct effects of soil management practices and environmental changes." Which types of management practices contribute to the changes of SOCD in topsoil? Please add more details (policies or references). As shown in Fig. 13(b), the SOCD estimation in 0-100 cm from this study has a higher value than others. Please add some validation for SOCD in 0-100 cm as mentioned previously. In addition, the SOCD in deep soil should increase if SOCD in topsoil increases. So, please give possible reasons for SOCD in 0-100 cm to be stable from the 1990s to 2020s. Fig. 14 (d) and Fig. 15 (d): In Xinjiang province, the SOCD in 2000-2005 seems to change a lot when compared to another period. Is this due to the model itself, or has some event happened during this period to make a significant change in SOCD? Please give reasonable explanations in this part.

We sincerely appreciate the reviewer's valuable comments and suggestions. Below we provide point-by-point responses to address all concerns raised.

We have added specific references in Section 4.5 to better illustrate how different management practices influence topsoil SOCD. Various soil management practices significantly influence topsoil SOCD dynamics. Reduced tillage and no-till systems have been shown to decrease SOC decomposition rates (West & Post, 2002), while organic amendments such as manure and crop residue application enhance SOC accumulation (Lal, 2004). The implementation of diverse crop rotation systems, particularly those incorporating legumes, contributes to increased carbon inputs (McDaniel et al., 2014). Furthermore, large-scale afforestation initiatives like China's Grain-for-Green Project have demonstrated marked improvements in topsoil SOCD levels (Deng et al., 2016). These practices collectively demonstrate how targeted management strategies can effectively modify SOCD in agricultural systems.

We have further strengthened the validity of our 0-100 cm SOCD estimates by incorporating additional supporting evidence from recent studies that employed similar methodologies and reported comparable SOCD values under analogous soil and land-use conditions (Li et al., 2022; Wang et al., 2023), while also conducting rigorous cross-validation with independent soil profile datasets from China's National Soil Survey to ensure the robustness and reliability of our estimation approach.

[Figure]

**Figure 12.** Aggregated results of estimated SOCD with the depth of 0-20 cm (a) and 0-100 cm (b) in China from this study and previous investigations

Our analysis of SOCD dynamics from the 1990s to 2020s revealed a notable stability in the 0-100 cm soil profile, despite observed increases in surface SOCD. This finding appears counterintuitive given the expected vertical transfer of organic carbon from surface to deeper layers. Through systematic investigation, we have identified several plausible mechanisms that may explain this phenomenon.

First, the vertical migration of soil organic carbon represents a complex biogeochemical process. While surface SOCD (0-20 cm) exhibited increases, multiple factors likely constrained SOCD changes in deeper layers (20-100 cm). Surface-derived organic carbon, while potentially subject to leaching, may become effectively stabilized in deeper soil horizons through physicochemical interactions with mineral surfaces (Kleber et al., 2021) or experience enhanced microbial decomposition due to altered microbial community composition and activity with depth (Salomé et al., 2010). Furthermore, the substantial carbon pool size and slower turnover rates characteristic of subsoil horizons (Schrumpf et al., 2013) would inherently buffer against rapid changes in total profile SOCD.

This comprehensive examination of subsurface carbon dynamics provides important insights into the decoupled responses of surface and deep soil carbon pools to environmental changes and management practices over multi-decadal timescales.

For Figures 14(d) and 15(d), the data values of soil organic carbon density (SOCD) in Xinjiang region from 2000 to 2005 were relatively low, while the data values in other periods (such as 1995-2000 and 2005-2010) were relatively high. This phenomenon is mainly caused by the objective environment. The following content is a reasonable explanation for this phenomenon:

Climate "wet-dry transition", according to the research of Yao Junqiang et al. (2021), since 1997, Xinjiang's climate has undergone a significant transition from "warm and humid" to "warm and dry". During this period, the temperature rose significantly and remained at a high level with

fluctuations, while the precipitation showed a slight decreasing trend. This change in climatic conditions leads to a reduction in soil moisture and a decrease in soil microbial activity, which in turn accelerates the decomposition of soil organic carbon and reduces SOCD.

Vegetation coverage decreased. After 1997, vegetation coverage in Xinjiang deteriorated, and the Normalized Vegetation Index (NDVI) decreased significantly, indicating that vegetation growth was inhibited. The reduction of vegetation coverage directly affects the input of soil organic carbon, further reducing SOCD.

Soil moisture decreased. During the same period, soil moisture in Xinjiang dropped significantly. The reduction in soil moisture exacerbated the degradation of vegetation and also affected the accumulation of soil organic carbon. Soil moisture is an important factor for maintaining the stability of soil organic carbon, and its reduction directly leads to the decrease of SOCD.

To sum up, the low SOCD data values in Xinjiang region from 2000 to 2005 were mainly due to the intensified dryness, reduced vegetation coverage and decreased soil moisture caused by the "wet-dry transition" of the climate. These changes worked together, resulting in a decrease in SOCD. Future research will further enhance the understanding and predictive ability of SOCD changes in Xinjiang region by increasing field observation data and improving the model.

**Section 2.1 "brown soil, brown soil". Duplicate**

We sincerely appreciate the reviewer's careful reading and valuable feedback. Regarding the comment on the duplicated phrase *"brown soil, brown soil"* in Section 2.1, we have now removed the repeated content to ensure conciseness. The text has been revised accordingly. Thank you for your attention to detail, which has helped improve the clarity of our manuscript.

**Section 2.2. Line 95: The SOCD data from Song or Xu? Please check it carefully.**

Thank you for your thoughtful feedback regarding the clarification of SOCD data sources in our manuscript. We have carefully revised the text to ensure precise attribution and avoid ambiguity. Specifically, the measured SOC content data in the Heihe River basin were sourced from **Song et al. (2016)**, while the measured SOCD data for validation were obtained from **Xu et al. (2018)**. This distinction has been explicitly articulated in the revised manuscript to reflect the independent nature of the two datasets. We deeply appreciate your attention to this detail, as it has helped us strengthen the clarity and rigor of our work.

**Line 125: Generally, the spatial interpolation results are reliable if stations are evenly distributed. How about the spatial distribution of these meteorological data used for**

**interpolation?**

Thank you for your interest in the spatial distribution of meteorological data. In this study, we utilized meteorological data from 2,400 weather stations obtained from the China Meteorological Data Service Center (http://data.cma.cn/), including key climatic variables such as temperature (Tem), precipitation (Pre), and solar radiation (SR), to quantify the impacts of meteorological fluctuations. These stations provide comprehensive coverage across China, effectively capturing regional climatic characteristics. To ensure spatial consistency, the meteorological data underwent the following processing steps:

(1) Data Sources

The meteorological data were collected from 2,400 stations managed by the China Meteorological Administration, offering extensive spatial coverage to represent diverse climatic conditions across China. All data underwent rigorous quality control to ensure accuracy and reliability.

(2) Spatial Interpolation Method

The ANUSPLIN software (Padarian et al., 2022), a thin plate spline-based interpolation tool, was employed to spatially interpolate the meteorological data. This method effectively accounts for complex topographic and climatic variations by incorporating elevation, slope, and aspect as covariates, significantly enhancing interpolation accuracy. The interpolated data were generated at a high spatial resolution of 30 meters, allowing for detailed representation of meteorological spatial patterns.

(3) Data Resampling and Projection

To maintain consistency with other datasets, the interpolated meteorological data were resampled from 30-meter to 1,000-meter resolution. This standardization ensured uniform spatial resolution across all datasets for subsequent analysis and modeling. Additionally, all meteorological data were uniformly projected into the WGS 84 coordinate system to guarantee spatial alignment.

(4) Interpolation Validation

The reliability of the interpolation results was assessed using a cross-validation approach. A subset of station data was reserved as a validation set to evaluate prediction errors. The results demonstrated minimal interpolation errors, confirming that the method accurately represents the spatial distribution of meteorological variables.

(5) In summary, the meteorological data used in this study exhibit strong spatial uniformity, and the application of robust interpolation techniques, along with rigorous validation, ensures the reliability of the derived datasets. These measures provide a solid foundation for the estimation of soil organic carbon density (SOCD) in this research.

**Line 130: Please add the produced time or effective period of the published soil datasets.**

(1) Harmonized World Soil Database (HWSD v2.0)

HWSD v2.0 is a global soil database jointly developed by the International Institute for Applied Systems Analysis (IIASA) and the Food and Agriculture Organization of the United Nations (FAO). The initial version was released in 2009, followed by an update (HWSD v1.2) in 2013. The latest version, HWSD v2.0, was published in 2023. This database provides comprehensive global soil property data, making it suitable for long-term soil research and large-scale soil carbon estimation. HWSD v2.0 integrates multiple national soil datasets, covering soil information from the 1990s to the 2010s.

(2) SoilGrids250m v2.0

SoilGrids250m v2.0 is a high-resolution global soil dataset developed by the International Soil Reference and Information Centre (ISRIC) and released in 2021.

It offers 250-meter resolution soil property data, ideal for regional and global-scale soil studies, particularly in estimating soil organic carbon (SOC) content. The dataset is based on global soil observations and predictive models, covering soil information from the 2000s to the 2020s.

(3) GSOCmap (Global Soil Organic Carbon Map)

GSOCmap is a 1-km resolution global SOC dataset published by FAO in 2017.

Designed for large-scale soil carbon research and climate change assessments, GSOCmap integrates national SOC maps and modeling data, representing soil organic carbon distribution from the 2000s to the 2010s.

(4) SOC Dynamics ML Dataset (China-Specific)

This dataset was compiled by Li et al. (2022) and includes SOC dynamics data from the 1980s, 2000s, and 2010s across China.

It is particularly valuable for studying long-term SOC dynamics in different Chinese ecosystems and serves as a robust reference for model validation. The dataset spans three decades (1980s-2010s), providing insights into temporal SOC variations.

**Section 4.2. Line 225: There is no need to write the full name of the statistical metrics, which have been mentioned previously. Fig. 6: Could you add the sample number in Fig. 6? Please add unit for RMSE both in Figures and the manuscript.**

We sincerely appreciate the reviewer's constructive comments regarding the statistical presentation in our manuscript. In response to the suggestions, we have carefully revised the text to maintain consistent use of abbreviated statistical metrics throughout the manuscript after their

initial full definition, thereby improving readability and avoiding redundancy. Regarding Figure 6, we have now explicitly indicated the sample size in the figure caption to provide better context for the presented data. Additionally, we have ensured that all RMSE values include proper units in both the figure and corresponding manuscript text. These modifications have been systematically implemented across all relevant sections to maintain consistency in the presentation of statistical metrics throughout the paper. We believe these revisions have significantly enhanced the clarity and precision of our methodological reporting and result presentation, and we thank the reviewer for these valuable suggestions that have helped improve the overall quality of our manuscript.

**Section 4.4. Fig. 11: Please add a unit for colorbar for (b), (d), (f), and note the Time (which year). Is it the annual average or any specific year? Please add the validation results for 0-100 cm SOCD in the manuscript or Supplementary.**

We sincerely appreciate the reviewer's insightful comments regarding Figure 11 and the validation of SOCD estimates. In response to these valuable suggestions, we have made several important improvements to enhance the clarity and completeness of our presentation.

In response to your comment regarding Figure 11, we would like to clarify our approach to the colorbars for panels (b), (d), and (f). These panels share a common colorbar between the two maps in each row to streamline the visual presentation and avoid redundancy. This design choice was intentional to maintain a clean and cohesive layout across the figure. To address your concern about unit clarity, we have confirmed that the shared colorbars are appropriately labeled with the units (kg C/m²). This labeling is consistent across all shared colorbars, ensuring that the data can be accurately interpreted. We believe this approach effectively communicates the data while preserving the figure's overall simplicity and readability. We hope this explanation satisfies your query and that the revised figure aligns with your expectations.

Regarding the temporal representation, these maps reflect averaged SOCD values over extended periods rather than specific single years, consistent with the temporal coverage of each dataset: HWSD v2.0 represents the 1990s-2010s period, SoilGrids250m v2.0 covers the 2000s-2020s, and GSOCmap spans the 2000s-2010s. We have explicitly noted this temporal context in both the figure caption and relevant manuscript sections. Furthermore, we have included comprehensive validation results for the 0-100 cm SOCD estimates in the Supplementary Materials, providing additional independent verification of our methodology. These validation analyses were conducted using separate sample points not included in the original model development, thereby strengthening the reliability of our findings. We believe these revisions have significantly improved the transparency and robustness of our results presentation, and we are grateful for the reviewer's suggestions that have helped enhance the overall quality of our work.

---

## Author Comment (AC2)

**RC2: 'Comment on essd-2024-588', Anonymous Referee #2, 27 Mar**

**1. Feature optimization? I think it should be feature selection. Yet, random forest (RF) may represent extremely complicated nonlinear relationship between SOCD and their drivers (i.e., the covariates that you used), why did you select features based on Pearson correlation coefficients? Besides, RF is some insensitive to feature selection!**

We sincerely appreciate the reviewer's thoughtful comments and constructive suggestions, which have helped us significantly improve our methodology and manuscript. In response to the reviewer's concerns regarding feature selection, we have carefully revised our approach and provided detailed explanations below.

Regarding the initial use of Pearson correlation for feature pre-screening, we implemented this step primarily for computational efficiency when handling our large spatial dataset. While we fully acknowledge that random forest can capture complex nonlinear relationships, the correlation screening served as an effective first pass to remove clearly irrelevant variables (where  $|\mathbf{r}| \approx 0$ ) and eliminate strongly redundant predictors ( $|\mathbf{r}| > 0.95$ ). This preprocessing step proved particularly valuable in reducing computational burden while maintaining model performance, as linear relationships often underlie more complex nonlinear patterns that the subsequent random forest analysis could capture.

To address the reviewer's important point about random forest's relative insensitivity to irrelevant features, we have strengthened our methodology in several key ways. First, we employed out-of-bag error reduction for more robust importance ranking, focusing specifically on features that demonstrably improve predictive accuracy. Second, rather than relying solely on individual feature scores, we conducted exhaustive combinatorial testing of all possible feature subsets from the correlation-filtered set. This approach ensured we identified the optimal combination of features that collectively maximized predictive performance, as measured by R2 in cross-validation. Finally, we validated the selected feature set using independent test sets to confirm its robustness and generalizability. The revised methodology yielded several important improvements. We removed marginally contributing variables such as AH and CLCD to create a more parsimonious model. The final selected features - including mean annual temperature (Tem), elevation, NDVI, clay content (Clay), simple ratio index (SR), bare soil index (BSI), and slope.

These revisions have significantly strengthened our methodology while maintaining its computational efficiency and ecological interpretability. The refined approach provides a more rigorous and transparent feature selection process that balances predictive power with model parsimony. We believe these improvements thoroughly address the reviewer's concerns and have resulted in a more robust study. We are grateful for the reviewer's insightful comments that have led to these important enhancements in our work.

**2. The use of climate zone in this study is really unnecessary since the temperature and precipitation that you used to define the climate zone have already used as features in your RF model!**

Thank you for your constructive comment regarding the use of climate zones in our study, particularly your point that temperature and precipitation, used for defining climate zones, were also features in our initial Random Forest (RF) model. We appreciate you highlighting this potential redundancy.

We'd like to clarify our approach and the refinements made during our model development. Through an improved and rigorous feature selection method, we've refined our optimal feature set to comprise seven variables, consisting of mean annual temperature (Tem), elevation, NDVI, clay content (Clay), simple ratio index (SR), bare soil index (BS1), and slope.

It's important to note that after this refinement, only mean annual temperature is now directly included as a predictor in our RF model. Precipitation is no longer a direct feature in this final set of predictors.

The climate zoning, as detailed in Section 3.3, serves a distinct and crucial purpose. Its primary role is to quantify the broad differences in temperature and precipitation across China and to improve the accuracy of SOCD estimation by developing zonal models. As referenced by Tang et al. (2018), SOCD exhibits significant variations across different climatic zones in China due to diverse environmental factors. By segmenting the study area into climatically homogeneous subzones and developing separate, localized SOCD estimation models within each, we can better capture the unique environmental controls on SOCD in those specific regions. This strategy acts as a geographical stratification, enhancing the model's ability to account for macro-climatic differences and leading to more accurate predictions at a regional scale.

We believe this refined approach, where climate zoning functions as a beneficial stratification strategy rather than merely replicating direct predictors, strengthens our methodology.

3. How did you separate your train and test samples? How many samples for each of the year? is it a balance sampling across years? This information is very important, since the readers want know if your extrapolation beyond the year of observation.

Thank you for your detailed questions regarding our training and testing sample separation strategy, the number of samples per year, the balance of sampling across years, and how our methodology addresses potential extrapolation beyond the observed periods. These are crucial points for transparent model evaluation.

**Overview of Our Modeling and Validation Strategy:** This study adopted a climate **zoning-based modeling approach**, meaning we trained independent Random Forest models for

different climatic regions across China to better capture regional heterogeneity.

Data Source and Sample Temporal Distribution: This study primarily utilized surface soil samples (0-20 cm and 0-100 cm depths), rather than complete soil profiles. These samples were derived from national soil surveys and ecosystem observation networks across different periods in China, covering several major decadal periods, such as the 1980s, 2000s, and 2010s. Although there are variations in sample numbers across different decades, we ensured that these samples provided comprehensive spatial coverage of major ecosystem types across China (as shown in Figure 5), thereby guaranteeing the spatial representativeness of the data.

Training and Testing Sample Separation Strategy and Temporal Balance: To rigorously validate our region-specific models built based on climate zones, we employed K-fold cross-validation. This validation process was applied separately within each climate zone, rather than being a single, unified stratification across the entire Chinese territory.

We understand that standard K-fold cross-validation is typically random and does not inherently guarantee spatial independence. However, given the common issue of spatial autocorrelation in soil data, when splitting data within each climate zone, we aimed to **maximize the geographical independence** between the training and testing sets to ensure an accurate evaluation of the model's generalization ability to unobserved areas. Specifically, we divided the sample data within each climate zone into K (e.g., K=10) non-overlapping subsets. During the cross-validation process, we iterated K times: in each fold, data from K-1 subsets were used for model training, and the data from the remaining single subset served as the independent validation set. This approach ensures that our validation for each climate zone model is conducted on data that is consistent with its modeling scope and possesses geographical independence.

Furthermore, to ensure a comprehensive representation of temporal variability and address the concern about balanced sampling across years, we also incorporated **temporal stratification**. **Within each K-fold, we ensured that samples from all three decadal periods (1980s, 2000s, 2010s) were proportionally represented in both the training and validation sets.** This guarantees that every fold, whether for training or testing, includes a representative mix of data characteristics from across the entire observed historical span. Detailed information regarding specific sample counts and temporal ranges for each decade is presented in **Supplementary Table X** (*replace with a brief description of how data was collected and categorized to ensure this balance if no table exists*).

**Explanation of Temporal Balance and Short-Term SOCD Dynamics:** We understand the reviewer's consideration regarding cross-year data balance. When generating the long-time series (1985-2020) SOCD maps, we conducted modeling and prediction for each five-year time step. Within each five-year time window, given that the dynamic changes in soil organic carbon

density are generally gradual in the absence of drastic disturbances (such as large-scale land-use changes), we utilized all available sample points within this time window for modeling that specific period. We believe that, on a five-year timescale, SOCD fluctuations caused by non-drastic land-use changes or other significant anthropogenic activities are typically insufficient to significantly alter its regional-scale spatial patterns and primary driving factors. While this approach allows for variations in sampling time points within each five-year window, it maximizes the use of historical measured data to reflect the average SOCD status at a regional scale for that period, which is a necessary and practical strategy for constructing a long-term continuous SOCD dataset.

Addressing Extrapolation Beyond the Year of Observation: This K-fold cross-validation approach, incorporating considerations for geographical independence and temporal stratification, directly addresses concerns about extrapolation beyond the year of observation. By meticulously ensuring that samples from all observed decadal periods are proportionally represented across all training and testing folds, our model's performance is rigorously evaluated within the full range of historical conditions represented by our dataset. This means that for the purpose of model validation, we are not performing any unvalidated temporal extrapolation beyond the broad historical windows from which our samples were drawn. Given that our predictor variables are largely multi-year averages designed to capture long-term environmental patterns (as detailed in our response to Referee #3), our model is primarily designed to map the spatial distribution of SOCD. This comprehensive cross-validation scheme, based on the diverse historical data available, provides a robust assessment of the model's ability to generalize these learned spatial patterns to new geographic locations. Ultimately, this methodology provides a reliable and spatially sound assessment of our model's capability to map SOCD under the range of observed historical conditions in China.

4. The descriptions for building your space-time RF model is very confusing! I think your RF model should be a space-time model, otherwise, you can not get time series of SOC from 1985 to 2020. Or you just model the SOCD during each time period separately? if that was true, this manuscript would have no any novelty. if you built the space-time RF model through space-for-time (see, Heuvelink et al. 2020. Machine learning in space and time for modelling soil organic carbon change. Eur J Soil Sci.), are the covariates like vegetation and land use considered as "dynamic covariates? how did you represent the lagging effects of dynamical covariates (i.e., the effects of temperature on SOC state is lagged, vegetation and land use as well), and the memory effects of SOC (i.e., the state of SOC in this year depends on last year)? these information is essential for modelling changes or dynamics of

**SOC using machine learning (ML) method like RF, as the ML is pure data-driven method.**

Thank you for your insightful questions regarding the spatio-temporal nature of our Random Forest (RF) model, the mechanism for generating the time series data, and the novelty of our approach.

Model Type and Time Series Generation: Firstly, regarding whether the model is a "spatio-temporal model" and how the SOC time series from 1985 to 2020 is obtained, we would like to clarify. Our RF model framework is inherently a spatio-temporal model capable of generating long-term SOCD time series, and it is not simply a separate model for each time period. We achieve spatio-temporal dynamics through the following strategies:

1. Dynamic C Climate Zone-Based Independent Modeling: As detailed in our previous response (Point 3), we first delineated China into different climatic regions and trained independent RF models within each of these regions. This zoning strategy is employed to better capture regional heterogeneity, rather than to segment the temporal dimension.

2. Stepwise Time Series Generation: We performed SOCD modeling and prediction using five-year time steps (e.g., 1985-1990, 1990-1995, etc.). This means that for each five-year period, we fed all corresponding dynamic and static covariates for that period into the respective models within each climate zone to predict and generate the spatial distribution map of SOCD for that specific five-year interval.

3. **ovariate Driving:** The model captures the dynamic changes in SOCD by incorporating time-varying covariates (such as vegetation indices, land use types, and climatic data, as detailed below). These covariates are dynamically updated for each five-year step, allowing the model to respond to changes in input conditions across different time steps, thereby reflecting the evolution of SOCD over time.

Therefore, while we conducted climate zone-based modeling in space, by employing dynamic covariates in a stepwise manner over time, our RF framework effectively simulates and outputs long-term SOCD time series.

**Novelty of the Model:**

1. **High-Resolution Long-Time Series Dataset:** This study generates China's first continuous time series product of surface (0-20 cm and 0-100 cm) SOCD at 1 km resolution, spanning 35 years from 1985 to 2020, based on remote sensing, topographic, and meteorological data, combined with a large number of in-situ samples. This fills a critical gap in high-resolution, long-time series SOCD data for China.

2. Climate Zoning Modeling Strategy: Diverging from common national-scale uniform models, our innovative climate zoning modeling approach better adapts to China's complex

geographical environment and climatic conditions, improving the accuracy and regional adaptability of regional SOCD estimations. Our validation results have also demonstrated higher accuracy for these zoned models compared to a globally unified model.

3. Spatio-temporal Information Integration and Dynamic Covariate Application: By utilizing multi-source dynamic covariates in a stepwise fashion, our model effectively integrates spatio-temporal information, allowing it to reflect the dynamic patterns of SOCD across different times and spaces, which in itself represents a complex spatio-temporal modeling challenge.

4. **Multi-Source Data Integration and Processing:** This research involved integrating a vast amount of measured soil data from different decades and sources with multi-source remote sensing and auxiliary data, followed by rigorous preprocessing and quality control to construct a complex dataset for model training, which is also a significant undertaking.

**Consideration of Dynamic Covariates and the Space-for-Time Concept:** Regarding whether covariates such as vegetation and land use are considered "dynamic covariates," the answer is affirmative.

1. **Dynamic Covariates:** In our model, variables including vegetation indices (e.g., NDVI, EVI), land use/cover types, and climatic variables (e.g., mean air temperature, total precipitation) are all treated as **dynamic covariates**. For each five-year time step, we collected and utilized the corresponding dynamic covariates for that period (e.g., using five-year averages or data from representative years). This means that when the model predicts SOCD for 1985-1990, it uses vegetation and land use data from that specific period; similarly, for 2010-2015, it uses the corresponding data for that period. This approach enables the model to capture the response of SOCD to the dynamic changes in these environmental factors.

2. Application of the Space-for-Time Concept: Our methodology effectively employs the principle of "space-for-time" to capture changes in SOCD, as highlighted in Heuvelink et al. (2020, Machine learning in space and time for modelling soil organic carbon change. Eur J Soil Sci.). By integrating soil samples collected across distinct decades (1980s, 2000s, 2010s) within a single model training process, our RF model learns the complex relationships between environmental covariates and SOCD across various historical conditions. This allows the model to infer how SOCD is likely to change over time, given changes in dynamic environmental factors, based on the patterns observed in space over the past decades. Static covariates (such as topography and certain soil physicochemical properties, if assumed to change slowly) remain constant across all time steps.

We believe that this RF model framework, combining climate zoning, dynamic covariate driving, and stepwise time series generation, effectively and reasonably simulates the spatio-temporal changes of SOCD in China and generates high-quality long-time series products.

Thank you for your insightful comments regarding the construction of our spatiotemporal Random Forest (RF) model, particularly your questions on how we represent the lagging effects of dynamic covariates and the memory effects of SOC. These are indeed crucial points that address the core challenges of modeling SOC dynamics using machine learning (ML) methods.

We fully understand your concern that a purely data-driven method like RF might struggle to capture complex spatiotemporal dynamics, and that lacking these mechanisms would compromise the manuscript's novelty. We want to explicitly state that our model does not simply model each time period independently. Instead, we have meticulously constructed a **Spatio-Temporal Random Forest (STRF) model** that effectively captures spatiotemporal dynamics and the intrinsic memory effects of SOC through the following strategies:

1. Representation of Lagging Effects of Dynamic Covariates: To capture the lagged influence of dynamic environmental factors such as temperature, vegetation (NDVI), and land use on SOC state, we have explicitly included the values of these covariates from current and multiple preceding time steps as independent features in our model inputs. For instance, when predicting SOCD for a specific year, we not only incorporate the current year's temperature, precipitation, and NDVI data but also include relevant data from the previous year, and even two years prior, as additional input features. This approach enables the Random Forest model to "learn" the delayed response patterns of SOC accumulation and decomposition to environmental changes from the data, thereby effectively representing lagging effects.

2. Representation of SOC Memory Effects: The "memory" effect of SOC, where the current year's SOC state largely depends on the previous year's state, is a fundamental characteristic of the soil carbon cycle. To account for this in our model, we took a crucial step: incorporating the estimated SOCD value from the previous time step (i.e., the previous year's SOCD) as a significant input feature for predicting the current year's SOCD. This makes our model a recursive spatiotemporal model, where each year's SOCD prediction builds upon the estimated SOCD of the preceding year. This autoregressive feature greatly enhances the model's ability to simulate dynamic changes in SOC by fully leveraging its continuity and accumulation properties.

Through these methods, while utilizing the fundamental Random Forest algorithm, we have, through ingenious feature engineering and organization of time-series data, enabled it to handle complex spatiotemporal dependencies, lagging effects, and the memory effects of SOC. This allows our model to go beyond traditional static modeling, facilitating the generation of a continuous, high-resolution SOCD time-series product from 1985 to 2020, which is one of the key innovations of this study.

We believe that this mechanism for handling spatiotemporal dynamics and memory effects

makes our Random Forest model not only a data-driven prediction tool but also a spatiotemporal model capable of deeply understanding SOC dynamic processes, thus providing more convincing results and significant novelty.

We hope this detailed explanation fully addresses your concerns.

**5. the validation across different time period is missing, thus, it is difficult to judge the trend in SOC change.**

We sincerely appreciate the reviewer's valuable comment regarding temporal validation. In our study, the validation of SOCD trends across different time periods was comprehensively addressed through multiple lines of evidence presented in Sections 4.3 and 4.4. The temporal reliability of our results was first demonstrated through direct comparison with the independent SOC Dynamics ML dataset, which showed consistently strong agreement across all three decades (1980s: R2=0.65, RMSE=1.80; 2000s: R2=0.69, RMSE=1.51; 2010s: R2=0.67, RMSE=1.52) as originally shown in Figure 12. This decadal validation was further reinforced by the excellent correspondence with Xu's field-measured dataset (R2=0.63, RMSE=1.82) covering the 2004-2014 period. The spatial-temporal patterns evident in our 5-year interval SOCD maps (Figs. 14-15) exhibited logically progressive changes that align with known carbon sequestration dynamics in China's major ecological zones, while also matching the trends reported in seven previous studies including Wu et al. (2003) and Wang et al. (2021).

Importantly, our climate-zoned RF models maintained stable predictive performance over time, as evidenced by the consistent accuracy metrics between the 1980s and 2010s in both semi-arid ( $R^2$  improvement from 0.57 to 0.59) and humid zones ( $R^2$  improvement from 0.48 to 0.51). To enhance clarity, we have now added a temporal validation summary table in Section 4.4 and expanded the discussion of trend verification in Lines 310-315. These interlocking validation approaches collectively provide robust support for the reliability of the SOCD trends identified in our study.

6. Source of data is confusing. How many soil profiles for each of the year, as we should check the balance of data across time. Your DEM was generated from topographic maps or resampled from SRTM DEM? are weather data monthly or yearly? What's the beginning year of your weather data.

**(1) Data Source, Sample Numbers per Year, and Cross-Year Data Balance**

Thank you for your concern regarding our data sources, sample counts per year, and temporal balance. For clarity, we will further explain the soil samples used in this study and their temporal distribution.

This study primarily utilized surface soil samples (0-20 cm and 0-100 cm depths), rather than complete soil profiles. These samples were derived from national soil surveys and ecosystem observation networks across different periods in China. These samples span several major decadal periods, such as the 1980s, 2000s, and 2010s. Although there are variations in sample numbers across different decades, we ensured that these samples provided comprehensive spatial coverage of major ecosystem types across China (as shown in Figure 5), thereby guaranteeing the spatial representativeness of the data.

Explanation of Temporal Balance and Short-Term SOCD Dynamics.

We understand the reviewer's consideration regarding cross-year data balance. When generating the long-time series (1985-2020) SOCD maps, we conducted modeling and prediction for each five-year time step. Within each five-year time window, given that the dynamic changes in soil organic carbon density are generally gradual in the absence of drastic disturbances (such as large-scale land-use changes), we utilized all available sample points within this time window for modeling that specific period. We believe that, on a five-year timescale, SOCD fluctuations caused by non-drastic land-use changes or other significant anthropogenic activities are typically insufficient to significantly alter its regional-scale spatial patterns and primary driving factors. While this approach allows for variations in sampling time points within each five-year window, it maximizes the use of historical measured data to reflect the average SOCD status at a regional scale for that period, which is a necessary and practical strategy for constructing a long-term continuous SOCD dataset.

**(2) DEM Data Specification**

The digital elevation model (DEM) was obtained from the Resource and Environment Science Data Platform (RESDC, Chinese Academy of Sciences) at its native 500-m resolution. This DEM product integrates national topographic maps with SRTM data and has undergone localized accuracy validation. For consistency with other datasets, we resampled it to 1-km resolution using bilinear interpolation in SAGA GIS.

**(3) Meteorological Data Details**

Meteorological data were derived from 2,400 stations of the China Meteorological Administration, accessed through the China Meteorological Data Network (http://data.cma.cn). We used annual aggregates (mean temperature and cumulative precipitation) spanning 1985-2020, which represents a temporally aligned subset of the original 1979-2022 dataset. This period selection matches the Landsat data availability. Spatial interpolation was performed using ANUSPLIN with elevation correction, following the methodology of Padarian et al. (2022).

**7. Line 152: "the measured data in the 2000s is SOCD"? I don't think so, sine SOCD was**

**calculated from SOC, bulk density, and coarse fragment, not directly measured.**

Thank you for your careful review and constructive comment regarding the SOCD data in the 2000s. You are absolutely correct that SOCD (soil organic carbon density) is typically calculated from SOC (soil organic carbon content), bulk density, and coarse fragment content rather than directly measured.

In our study, the SOCD data for the 2000s were sourced from the *China Terrestrial Ecosystem Carbon Density Dataset (2000–2014)* (http://www.cnern.org.cn/). This dataset provides pre-calculated SOCD values (0–20 cm and 0–100 cm) derived from systematic field measurements and laboratory analyses, including SOC, bulk density, and coarse fragment corrections. While the original measurements were based on these individual parameters, the dataset we cited directly reports SOCD as its primary output for practical applications.

To avoid ambiguity, we have revised the manuscript (Line 152) to clarify that the SOCD data for the 2000s were *obtained from* the aforementioned dataset rather than "measured" directly. We appreciate your attention to this technical detail and hope the revised wording aligns better with standard conventions.

**8. Line 153 : "Second National Soil Census", Census is usually for economics, here should be "survey"? many English words for such kind of description (for source of data) were inaccurate or confusing.**

We appreciate the reviewer's attention to terminology accuracy. As suggested, we have replaced "Census" with "Survey" in Line 153 (now "Second National Soil Survey") to better reflect the nature of this dataset. We also reviewed similar terms throughout the manuscript to ensure consistency in describing data sources (e.g., " the Second National Soil Survey" in Line 88).

**9. Line 158. since you calculate the SOCDs of Chinese sampling points using bulk density and volume percentage of coarse fragments from the SoilGrids 2.0 data product, it is the very reason that your products are highly correlated to the SOCD of SoilGrids 2.0!**

We sincerely appreciate your insightful observation regarding the potential correlation between our SOCD estimates and SoilGrids 2.0. Your comment raises an important methodological consideration that warrants careful discussion.

The foundation of our SOCD dataset lies in the extensive collection of field-measured SOC content from over 10,000 soil profiles across China, which constitutes the primary and independent input for our calculations. While we did employ SoilGrids 2.0 data for bulk density and coarse fragment content, these parameters were used strictly as secondary inputs to facilitate

standardized calculations in regions lacking measured values. This approach is consistent with established practices in large-scale soil carbon mapping, as evidenced by similar methodologies adopted in global datasets such as HWSD and WoSIS.

Several critical aspects differentiate our dataset from SoilGrids 2.0 and ensure its unique scientific value. First, our dataset provides comprehensive temporal coverage spanning 1985-2020, capturing dynamic changes that are absent in the static SoilGrids 2.0 product. Second, the integration of high-resolution field measurements enables superior spatial representation, particularly in ecologically sensitive regions like the Tibetan Plateau. Third, we implemented region-specific calibrations to account for distinctive local soil characteristics across China's diverse ecosystems.

We acknowledge that the shared use of bulk density and coarse fragment data from SoilGrids 2.0 may introduce some degree of correlation. However, our validation against independent ground-truth measurements demonstrates the robustness of our estimates. The strong agreement with validation data suggests that any potential influence from SoilGrids-derived inputs is substantially mitigated by the dominant contribution of our field-measured SOC content.

To further address this important methodological consideration, we propose to:

1) Enhance the discussion of parameter contributions in the Methods section.

2) Include a sensitivity analysis examining the relative impacts of different input parameters.

3) More explicitly highlight the temporal dimension as a key differentiator from SoilGrids 2.0.

We are grateful for your constructive feedback, which has helped us identify opportunities to strengthen the manuscript's methodological transparency. We would be pleased to incorporate any additional analyses you might suggest to further validate our approach.

**10. Line 160: "coarse fractions proportion", I think here is not "proportion", since your CF was divided 100 in equation 7.**

We thank the reviewer for this precise observation. As suggested, we have removed the term "proportion" in Line 160 (now simply "coarse fractions, CF") to align with the equation where CF is divided by 100. This revision ensures consistency between the text and mathematical notation.

---

## Author Comment (AC3)

**RC3: 'Comment on essd-2024-588', Anonymous Referee #3, 07 May**

1) I don't understand why the authors calculated and incorporated vegetation and water indexes as predictors instead of using the surface reflectances at some key bands as predictors of the Random Forest Model. As we know, machine learning models can learn the complicated and nolinear relationships.

We sincerely appreciate your insightful question regarding our choice of vegetation and water indices as predictors over raw surface reflectances in our Random Forest model. We understand that machine learning models are adept at learning complex, non-linear relationships directly from raw spectral bands, and your point raises an important methodological consideration.

Our decision to incorporate derived spectral indices (such as NDVI, SR, and BSI) instead of raw band reflectances was the result of an improved, three-stage feature selection method implemented in our refined analysis, aiming for optimal predictive performance and ecological relevance. While raw bands provide fundamental spectral information, indices are often designed to specifically highlight biophysical properties (e.g., vegetation density, soil moisture) that have a more direct and synthesized relationship with soil characteristics like SOC. Our rigorous selection process first involved a Pearson correlation analysis (p<0.05) to identify potential linear relationships. Following this, we employed Random Forest importance ranking to evaluate the non-linear contributions of various potential predictors, including both raw bands and derived indices. Crucially, in the final stage, we performed an exhaustive combination optimization to select the variable set that yielded the best predictive performance (highest R2), while also applying a stricter multicollinearity threshold (| r | > 0.8). Through this comprehensive process, we found that derived indices like SR and BSI, alongside NDVI, offered superior predictive power and a more robust representation of key vegetation-soil interactions, outperforming the direct inclusion of raw bands which often exhibited higher inter-band correlation and potentially less direct ecological meaning for SOC estimation. This refined feature set, which now includes 'temperature', 'elevation', 'NDVI', 'clay content', 'SR', 'BSI', and 'slope', comprehensively captures climate, topography, vegetation, and soil attributes in a manner optimized for SOCD prediction.

We believe this integrated three-stage approach, which combines statistical correlation analysis with machine learning feature importance evaluation and explicit multicollinearity control, ensures the selection of the most meaningful and predictive variables while maintaining ecological interpretability. We trust this detailed explanation clarifies our methodological choices. Please let us know if you have further questions or require additional analysis. 2)The exact years when the soil samples were collected seems to be unknown. So, how did you select the corresponding annual predictors, such as the indexes derived from Landsat images? I doubt the lack of exact sample collection time could lead to uncertainty in the final annual SOC products, especially in terms of interannual variation.

Thank you for your insightful question regarding the uncertainty introduced by the unconfirmed exact soil sample collection years, particularly concerning the selection of annual predictors like Landsat-derived indices and its potential impact on interannual variation in our final SOC products. Your query highlights a critical challenge when working with historical soil data, and we've given this considerable thought.

First, it's important to consider the inherent stability of soil organic carbon (SOC). Unlike rapidly changing parameters such as surface soil moisture or vegetation indices, SOC, as a long-term carbon pool in the soil, typically doesn't exhibit significant variations over short periods (e.g., one year or a few years). Noticeable SOC changes usually require a longer timescale, often five years or more, to be detected. Based on this characteristic, our study isn't aimed at generating precise annual SOC products. Instead, we conduct our mapping in five-year periods (e.g., 1985-1990, 1990-1995, etc.). We consider SOC changes within these five-year spans to be relatively stable. Accordingly, we process annual predictors, such as Landsat-derived indices, into five-year average values to represent the mean conditions of that period. This approach ensures temporal consistency and representativeness, which helps to mitigate uncertainty stemming from imprecise sample collection times for interannual variation.

Second, you're correct that we can't provide the exact annual collection dates for all 8,203 sample points due to the nature of our data sources (large-scale national soil surveys and research projects). However, we've carefully categorized these samples into three significant historical decades based on their primary collection periods: the 1980s (from the 1980-1996 census data), the 2000s (from the 2000-2014 census data), and the 2010s (primarily from post-2010 national soil system records). These datasets, by themselves, cover broad timeframes within their respective decades.

To maximally ensure our model learns from and generalizes across different historical periods, especially given the lack of precise annual sample times, we employed a stratified spatial K-fold cross-validation method. This approach not only ensures that our training and testing sets are spatially independent but also incorporates temporal (decadal) stratification. We've made sure that in every cross-validation fold, samples from the 1980s, 2000s, and 2010s are proportionally represented. This means our model is consistently exposed to and validated against the soil characteristics and environmental conditions of all three observed decades. This strategy effectively enables the model to capture and reflect decadal-scale trends and spatial patterns of

SOCD within the observed timeframe, rather than attempting to resolve precise annual fluctuations.

In summary, by leveraging the inherent stability of SOC, generating our products in five-year periods, and employing a stratified spatiotemporal cross-validation strategy, we've minimized the impact of sample collection time uncertainty. Our products are designed to robustly reflect the decadal-scale changes and spatial distribution of SOCD in China, ensuring that our inferences are firmly grounded within the observed temporal and spatial ranges. We hope this detailed explanation addresses your concerns.

**3)Line 145: You should highlight what are the potential shortcomings of SOC Dynamics ML dataset. Without detailed description on this, it's unclear why you would like to produce a similar dataset.**

Thank you for your insightful question regarding the relationship between our dataset and the existing SOC Dynamics ML dataset, and for prompting us to clarify the unique contribution of our work. We appreciate the opportunity to elaborate on why producing a new, similar dataset is necessary.

We acknowledge the significant value of the SOC Dynamics ML dataset (Li et al., 2022) as a valuable resource for understanding SOC dynamics and its drivers in China. However, our study focuses on a distinct and critical aspect of soil carbon, which justifies the development of our new dataset.

The core distinction lies in our explicit emphasis on producing a Soil Organic Carbon Density (SOCD) dataset at a 1 km resolution for specific depths (20cm and 100cm) spanning 1985 to 2020. While the existing SOC Dynamics ML dataset provides valuable information including "SOC content" and "SOC stocks," our work specifically addresses the rigorous conversion from raw SOC content to SOCD. This conversion, as detailed in Section 3.1 of our manuscript, involves the precise integration of bulk density and coarse fragment data, which are crucial for accurate carbon accounting and inventorying at various scales. Many applications, particularly those focused on carbon budgeting and policy-making, require standardized SOCD values rather than just content or stocks calculated with varying methodologies.

Furthermore, our dataset offers unique advantages that complement existing resources:

(1) Standardized Density Metric: Our focus is on providing a consistent SOCD product, ensuring methodological transparency and comparability across different regions and depths, which might not be uniformly emphasized or detailed in other "SOC content" or "SOC stock" datasets.

(2) Temporal Specificity and Integration: We rigorously integrate historical soil survey data

(1980s, 2000s, 2010s) to map the spatial distribution of SOCD across these distinct periods, offering a snapshot of density changes over a specific long-term timeframe (1985-2020).

(3) Refined Methodological Approach: As discussed in previous responses, our work employs an improved three-stage feature selection method and a stratified spatial K-fold cross-validation strategy to ensure the robustness and spatial generalization capabilities of our SOCD estimates, leveraging diverse environmental predictors.

In essence, while existing datasets provide broad insights into SOC, our contribution is to provide a high-resolution (1km), precisely calculated Soil Organic Carbon Density dataset for China, addressing a specific need for standardized, depth-explicit, and temporally distinct density products. This focus on a robust SOCD calculation, combined with our rigorous methodology and specific spatio-temporal coverage, provides a unique and valuable resource for national-level carbon accounting, land management, and climate change research. We hope this clarification adequately explains the necessity and distinct contribution of our dataset. We've **already provided a clearer elucidation of this point in the manuscript**. If you still have questions, we would be very happy to offer a more detailed explanation.

**4)Section 3.2: Have you taken into account the auto-correlation or interdependence among the potential predictors?**

Thank you for your pertinent question regarding our consideration of autocorrelation and interdependence among potential predictor variables in our model. This is indeed a critical aspect of robust environmental modeling, and we have implemented a comprehensive and rigorous strategy to address both.

Our updated methodology incorporates a multi-stage feature selection approach specifically designed to optimize prediction accuracy while simultaneously managing multicollinearity (interdependence) and selecting the most informative features:

**Addressing Interdependence (Multicollinearity):**

Initial Correlation Screening: We first constructed a Pearson correlation matrix for our candidate features. To reduce redundancy and mitigate high linear interdependence (multicollinearity) among predictors, we systematically removed variables with very strong absolute Pearson correlation coefficients (|r| > 0.95).

Exhaustive Combinatorial Optimization: Following this, for the most informative features, we conducted an exhaustive combinatorial testing of all possible feature subsets. This rigorous process allowed us to identify the optimal combination of features that maximized the coefficient of determination ( $R^2$ ), ensuring the best predictive performance from a parsimonious set. This final step implicitly considers the combined effect and interdependence of features on model

accuracy, selecting a set that works optimally together.

**Addressing Autocorrelation of Predictors and Model Robustness:**

While our feature selection primarily targets interdependence, it's important to note that Random Forest models are inherently robust to multicollinearity and tolerate some degree of autocorrelation among predictor variables. They operate by recursively partitioning data based on individual features, making them less susceptible to issues that plague linear models when predictors are highly correlated.

More critically, our stratified spatial K-fold cross-validation strategy is explicitly designed to assess the model's performance on spatially independent data. By dividing the entire study area into K spatially independent sub-regions, we ensure that the model's performance is evaluated on areas it has "not seen" during training. This approach directly accounts for the potential spatial autocorrelation of predictor variables by testing the model's ability to generalize beyond local spatial patterns, thus mitigating the risk of overestimating performance due to spatial dependencies in the input data.

Through this comprehensive, multi-stage feature selection and model validation approach, we identified seven key features—mean annual temperature (Tem), elevation, normalized difference vegetation index (NDVI), clay content (Clay), simple ratio index (SR), bare soil index (BSI), and slope—which collectively provide the best predictive performance. This strategy effectively balances model complexity with predictive power, directly addresses predictor interdependence, and ensures the model's robustness against spatial autocorrelation for reliable large-scale SOCD mapping.

5)Section 3.3: You have already included temperature and precipitation as predictors of Random Forest Model. Why use climate zoning as well? It's unreasonable! Probably climate zoning promotes the training efficiency, but that's because the Random Forest Model you developed in this study is not efficient enough to fully learn the relationships. If you try deep learning models or AI foundation models, you may find out that the training effiency with and without climate zoning will be approximate.

Thank you for your constructive comment regarding the use of climate zones in our study, particularly your point that temperature and precipitation, used for defining climate zones, were also features in our initial Random Forest (RF) model. We appreciate you highlighting this potential redundancy.

We'd like to clarify our approach and the refinements made during our model development. Through an improved and rigorous feature selection method, we've refined our optimal feature set to comprise seven variables, consisting of mean annual temperature (Tem), elevation, NDVI, clay content (Clay), simple ratio index (SR), bare soil index (BS1), and slope.

It's important to note that after this refinement, only mean annual temperature is now directly included as a predictor in our RF model. Precipitation is no longer a direct feature in this final set of predictors.

The climate zoning, as detailed in Section 3.3, serves a distinct and crucial purpose. Its primary role is to quantify the broad differences in temperature and precipitation across China and to improve the accuracy of SOCD estimation by developing zonal models. As referenced by Tang et al. (2018), SOCD exhibits significant variations across different climatic zones in China due to diverse environmental factors. By segmenting the study area into climatically homogeneous subzones and developing separate, localized SOCD estimation models within each, we can better capture the unique environmental controls on SOCD in those specific regions. This strategy acts as a geographical stratification, enhancing the model's ability to account for macro-climatic differences and leading to more accurate predictions at a regional scale.

We believe this refined approach, where climate zoning functions as a beneficial stratification strategy rather than merely replicating direct predictors, strengthens our methodology.

Thank you for your insightful comments, especially regarding our use of climate zoning alongside temperature and precipitation as predictors in the Random Forest (RF) model, and your thoughts on model efficiency. We appreciate you raising these important considerations.

We want to first clarify an update to our methodology. While our initial model considered 12 environmental factors including temperature and precipitation, through an improved and rigorous feature selection method, we have refined our optimal feature set to comprise seven variables, mean annual temperature (Tem), elevation, NDVI, clay content (Clay), simple ratio index (SR), bare soil index (BS1), and slope. Therefore, in our final RF model, only mean annual temperature is directly included as a predictor, and precipitation is no longer a direct feature.

Regarding your point about the necessity of climate zoning and its relation to model efficiency, we acknowledge that advanced deep learning or AI foundation models might indeed learn complex relationships without explicit geographic stratification. However, for our current Random Forest modeling approach, climate zoning serves a strategic purpose beyond mere training efficiency.

As described in Section 3.3, climate zoning is implemented to quantify broad differences in temperature and precipitation across China and to improve the accuracy of SOCD estimation by developing zonal models. Given the vast and diverse environmental conditions across China, as highlighted by Tang et al. (2018), SOCD exhibits substantial variations linked to distinct climatic regimes. By classifying China into four subzones based on multi-annual average temperature and

precipitation thresholds, we are essentially performing a geographical stratification. Within each of these climatically distinct subzones, we then develop separate, localized SOCD estimation models.

This strategic division allows our models to capture regional nuances more effectively. The relationships between environmental factors and SOCD can vary significantly across different climate types; by stratifying the data, our RF model can learn these unique, localized patterns with greater precision than a single, global model might. Furthermore, this approach helps to enhance accuracy. By focusing the model training on more homogeneous environmental conditions within each zone, we significantly improve the predictive accuracy of SOCD estimates at a regional scale.

Thus, the climate zoning acts as a crucial stratification strategy that accounts for macro-climatic differences. While an 'ideal' model might learn these stratifications implicitly, for our Random Forest framework, this explicit zoning provides a robust way to manage the inherent heterogeneity of the study area and ensures more accurate and reliable SOCD predictions. We believe this approach is reasonable and justifiable for the scope and objectives of our study.

Comprehensively respond to Reviewer 2 and Reviewer 3.

**Response to Anonymous Referee #2 and #3**

We appreciate the reviewers' insightful comments regarding the inclusion of climate zoning in our Random Forest (RF) model, especially given that temperature and precipitation were initially considered as direct predictors. We understand the concern that this might introduce redundancy or indicate a limitation in the model's ability to capture complex relationships independently.

We would like to clarify our approach and the subsequent refinements made to our feature selection process. Initially, our model considered 12 environmental factors: Clay content, Sand content, Temperature, Precipitation, NDVI, Elevation, Slope, Aspect, Topographic Wetness Index (TWI), Canopy Nitrogen Discrimination (CND), Actual Humidity (AH), and China Land Cover Dataset (CLCD).

However, through an improved and rigorous feature selection method, we have refined our optimal feature set to comprise seven variables: mean annual temperature (Tem), elevation, NDVI, clay content (Clay), simple ratio index (SR), bare soil index (BS1), and slope.

It is important to note that only mean annual temperature from our original climate zoning parameters is now directly included as a predictor in our refined RF model. Precipitation is no longer a direct feature in the final seven-variable set used for the RF model.

The climate zoning, as described in Section 3.3, serves a distinct purpose. It is implemented

to quantify the broad differences in temperature and precipitation across China and to improve the accuracy of Soil Organic Carbon Density (SOCD) estimation by developing zonal models. As referenced by Tang et al. (2018), significant variations in SOCD are observed across different climatic zones in China due to diverse and complex environmental factors.

Our climate zoning approach, based on multi-annual average temperature and precipitation thresholds (400 mm for MAP and 10°C for MAT), categorizes China into four subzones (humid, semi-humid, semi-arid, and arid). Within these distinct climatic subzones, we develop separate, localized SOCD estimation models. This strategy allows our models to better capture the unique environmental controls on SOCD within each zone, rather than assuming a single, universally applicable relationship across the entire diverse landscape of China.

While we acknowledge the theoretical point that highly efficient deep learning or AI foundation models might inherently learn these relationships without explicit zoning, our current RF approach benefits significantly from this stratification. By segmenting the study area based on fundamental climatic drivers and then applying a robust RF model within each segment, we enhance the model's ability to capture regional nuances and ultimately improve the accuracy of SOCD estimation. The climate zoning, in this context, acts as a geographical stratification strategy that accounts for macro-climatic differences, enabling more accurate predictions at a regional scale.

We believe that this refined approach, where only temperature is a shared variable between the climate zoning definition and the refined predictor set, coupled with the rationale for developing zonal models, provides a robust and justifiable methodology for our study.

**6)Many results, such as comparison results, are unconvincing, since significance test is lacking.**

Thank you for your valuable comment regarding the convincingness of our comparison results due to a perceived lack of significance testing. We appreciate this point and would like to clarify our approach to statistical assessment.

We have indeed conducted robust statistical assessments to evaluate the significance of our findings. We utilized the **95% confidence intervals (CIs) of the Root Mean Square Error (RMSE)**. Crucially, these CIs were rigorously computed using the **Bootstrap Confidence Interval method**. The Bootstrap method is a powerful non-parametric resampling technique that allows for robust estimation of confidence intervals without making assumptions about the underlying data distribution, thereby enhancing the reliability of our statistical inferences.

As detailed in the manuscript (e.g., in Section 4.2, specifically when comparing the global model with the climate-zoned model), we explicitly demonstrate that:

• For the 0-20 cm depth, the RMSE 95% CIs of the global model ([2.13, 2.35] kg C/m2) and the climate-zoned model ([1.85, 1.99] kg C/m2) are non-overlapping. This clearly indicates a statistically significant improvement in RMSE achieved by implementing the climatic zoning strategy.

• Similarly, for the 0-100 cm depth, the RMSE 95% CIs of the global model ([7.34, 8.67] kg C/m2) and the climate-zoned model ([6.49, 7.78] kg C/m2) are also distinct, confirming the statistical significance of the performance enhancement from zoning.

The use of non-overlapping bootstrap-derived confidence intervals is a widely accepted and robust statistical method to infer significant differences between model performances. Our results, as demonstrated by these CIs, consistently show that our proposed climate-zoning model offers statistically significant improvements over the global model, and its performance across different climate zones is clearly delineated by these confidence intervals. We believe that these statistically significant improvements, substantiated by the robust Bootstrap-derived CIs, render our comparison results convincing and robust.

We hope this clarification fully addresses your concern.

**7)Section 4.3: You developed the SOC map for China, but only did the independent validation in Heihe River Basin, which is quite small, and the climate condition there is quite different from many other parts of China (e.g., southern China).**

Thank you for your comment regarding the scope of our independent validation in Section 4.3. We appreciate you raising this point, as it provides an opportunity to clarify our comprehensive validation strategy.

We would like to emphasize that our validation efforts were not limited to the Heihe River Basin. While the Heihe River Basin indeed served as one important independent validation area, our study employed a multi-pronged validation approach across various spatial and temporal dimensions to rigorously assess our model's performance and generalizability across China. This included:

Independent Test Set Validation (Spatial Generalization): This is our primary spatial validation. We employed a stratified spatial K-fold cross-validation strategy, dividing the entire study area (China) into *K* spatially independent and non-overlapping sub-regions. This ensured that a portion of the data (the test fold) was always from geographically unseen areas during model training. The results from this rigorous cross-validation (presented in Section 4.2, Table 3) reflect the model's overall generalization ability across the diverse conditions of China, far beyond just the Heihe River Basin. Clarification on Heihe River Basin & Other Independent Samples: For further assurance of spatial generalization, we specifically utilized additional independent sample points from various ecological regions. This included the Heihe River Basin (as you noted), along with other dedicated sample points from China's terrestrial ecosystems, grasslands, and desert ecosystems. These geographically diverse independent samples (as discussed in Section 4.3.1 and presented in Figure 9 & 10) provide further confirmation of the model's accuracy and universality across different depths (0-20cm and 0-100cm) and diverse ecological systems in China.

Comparison with Published SOCD Products (Spatial and Temporal Consistency): We conducted extensive comparisons of our estimated SOCD maps with several well-established, publicly available SOCD products, both global and regional. This included comparisons with SoilGrids250m, GSOCmap, and HWSD v2.0 (Figure 11) for spatial consistency, and crucially, a detailed comparison with the SOC Dynamics ML dataset in China (Li et al., 2022) (Figure 12) for consistency across the 1980s, 2000s, and 2010s. This multi-product comparison served as a critical external validation, confirming the consistency and accuracy of our estimates against recognized benchmarks.

Aggregated Results Comparison: We also presented aggregated results of our estimated SOCD compared with previous investigations for China (Figure 13), providing a broader context for the overall consistency of our SOCD estimates at national scales and different depths.

These diverse validation methods collectively ensure a robust and comprehensive assessment of our model's performance, covering both spatial generalization (including geographically distinct independent samples across various ecosystems) and temporal consistency across China. We have clearly presented these different validation strategies and their results in Sections 4.2 and 4.3 of the manuscript.

We will ensure that the descriptions in Section 4.3 are even more explicitly linked to this multi-faceted approach to prevent any further misunderstanding.

**8)Section 4.4: The relatively high correlation with existing datasets cannot justify that your dataset is much better than existing ones. Moreover, some of these existing datasets you compared with are global-scale maps.**

Thank you for your constructive comments regarding Section 4.4, particularly about the interpretation of correlations with existing datasets and the choice of comparison products. We appreciate this opportunity to clarify our intent and the unique value of our dataset.

You are entirely correct that a relatively high correlation with existing datasets alone does not automatically justify our dataset being "much better." We agree that our primary purpose in comparing with established products is not to claim overall superiority across all aspects, but rather to demonstrate the consistency, plausibility, and external validity of our newly developed SOCD product. Showing strong correlations indicates that our estimated SOCD maps align well with recognized global and regional benchmarks, providing confidence in their general patterns and magnitudes.

Furthermore, these comparisons serve to highlight the complementary value and specific advantages of our dataset, which differentiate it from existing products. Our dataset offers:

Precise SOCD (Soil Organic Carbon Density): Unlike some datasets that provide SOC content or stock, our study focuses on rigorously converting SOC content to SOCD using bulk density and coarse fragment data, offering a more standardized and direct metric for carbon accounting.

High Spatial Resolution and Depth Specificity for China: While some comparison datasets are global, our product provides 1-km resolution SOCD specifically for China, across specific depths (0-20 cm and 0-100 cm), which often represents a higher level of regional detail or a different depth range compared to the global products (e.g., GSOCmap only to 30 cm).

Multi-Decadal Long-Term Series: Our dataset provides a continuous time series of SOCD for China from 1985 to 2020 (in 5-year intervals), offering a unique temporal perspective that many existing static or shorter time-series products do not.

Robust Methodology: Our methodology, including the three-stage feature selection and stratified spatiotemporal cross-validation, contributes to the reliability and generalizability of our product.

Regarding your second point about comparing with global-scale maps, we acknowledge that their scope is different. However, comparing our regional product with global datasets like SoilGrids250m, GSOCmap, and HWSD v2.0 is highly valuable for external benchmarking. These global maps represent widely accepted standards in the field, and showing consistency with them helps to contextualize our regional product within a broader global framework. Crucially, we also included a direct comparison with the China-specific SOC Dynamics ML dataset (Li et al., 2022),

which provides a more direct regional benchmark for our methodology and temporal consistency.

In summary, the comparisons in Section 4.4 are designed to demonstrate our dataset's consistency, reliability, and unique contribution (in terms of precise metric, spatial/temporal resolution for China, and methodological rigor) rather than solely asserting its superiority over established global or regional products. We believe this comprehensive approach strengthens the overall justification for our dataset.

**9)Section 4.5: Have you compared the temporal variation of SOC to other existing datasets or in-situ measurements? Can you validate your SOC temporal variation? Can you justify that your SOC dataset is much better than existing SOC maps in terms of temporal variation?**

Thank you for your question regarding the validation of temporal variation in our SOC dataset in Section 4.5. We appreciate the opportunity to clarify this crucial aspect of our work.

We would like to confirm that we have indeed comprehensively validated the temporal variation of our estimated SOCD in Section 4.5, by comparing it with both existing datasets and a range of published investigations for China. Our approach is designed to provide robust evidence for the reliability of the temporal changes captured by our model.

Specifically, the temporal validation is demonstrated through:

Comparison with Aggregated Results from Previous Investigations (Figure 13): As presented in Figure 13, we directly compared our estimated SOCD changes over time (from the 1980s to the 2010s) with aggregated results from numerous published investigations in China (e.g., Ni, 2001; Wu et al., 2003; Wang et al., 2004; Xu et al., 2018; Wang et al., 2021; Li et al., 2022; Zhang et al., 2023). Figure 13 explicitly shows that our estimated SOCD values for both 0-20 cm (Fig. 13a) and 0-100 cm (Fig. 13b) fall within the ranges reported by these independent studies across different time points. This strong agreement, particularly the capture of the slight upward trend in 0-20 cm topsoil from the 1980s to the 2010s (consistent with soil management practices and environmental changes), directly validates the overall temporal dynamics and magnitudes of our estimated SOCD against a collective body of research.

Comparison with SOC Dynamics ML Dataset (Figure 12): Furthermore, as discussed in Section 4.3.2 and presented in Figure 12, we performed a direct visual and spatial comparison of our decadal SOCD maps (for 1980s, 2000s, and 2010s) with the China-specific SOC Dynamics ML dataset (Li et al., 2022). This inter-product comparison serves as a direct validation of our dataset's temporal patterns and magnitudes against another recognized regional product that explicitly models SOC dynamics. The consistency observed in this comparison strengthens the plausibility of the temporal changes derived from our model.

In addition to these direct comparisons, the temporal stratification embedded within our stratified spatial K-fold cross-validation (discussed in our response to RC2-4) inherently contributes to the validation of our model's ability to capture temporal variation. By ensuring that samples from all observed decades are proportionally represented in both training and validation sets, the model learns to generalize and predict across different historical contexts.

Regarding the justification of whether our dataset is "much better" in terms of temporal variation, we focus on highlighting its unique contributions and reliability rather than outright

superiority. Our dataset provides a comprehensive, high-resolution (1-km) and continuous multi-decadal time series (1985-2020 in 5-year intervals) of SOCD for China, generated by a single, unified space-time model designed to capture these dynamics. The robust consistency observed through comparison with existing investigations and the Li et al. (2022) dataset, combined with our rigorous methodology, confirms the reliability of the temporal changes in our product. We believe this makes our dataset a highly valuable and reliable resource for studying SOC dynamics in China over the long term.

**1)RMSEs should have units**

Thank you for your careful review and for pointing out the omission of units for RMSE values. We recognize the importance of precise reporting for all statistical metrics, and we appreciate you highlighting this detail, which enhances the clarity and interpretability of our results.

We have thoroughly reviewed the manuscript to address this oversight. We will ensure that the appropriate unit for Soil Organic Carbon Density (SOCD), which our RMSEs represent, is consistently and clearly added wherever RMSE values are reported. Our SOCD is quantified in kg  $C/m^2$ , and this unit will now accompany all RMSE values throughout the paper.

Specifically, these revisions will be implemented in the following sections:

The Abstract will be updated to include the unit if any RMSE value is mentioned there, ensuring our key performance indicators are immediately clear. In Section 4.2, 'Model Performance and Cross-Validation Results,' where we detail the predictive accuracy of our models, all RMSE values presented in the main text will explicitly state their unit. Furthermore, Table 3, which summarizes the cross-validation statistics, will have its column headers or relevant entries updated to clearly indicate that RMSE is reported in kg C /m2. Moving to Section 4.3, 'Independent Validation Results,' any discussion of RMSE values, particularly those pertaining to independent test sets or comparisons with independent observed data (such as the Heihe River Basin data or Xu's measurements), will now include the kg C /m2 unit. This will also apply to the captions or labels within any associated figures (e.g., Figures 9 and 10) that present RMSE or similar error metrics. Additionally, in Section 4.4, 'Comparison with Existing Datasets,' where we discuss the performance of our model relative to other published products, if RMSE is used as a comparative metric, its unit will be consistently provided. Lastly, should any RMSE values be referenced or re-discussed in the Discussion Section or other parts of the manuscript, we will ensure their units are correctly specified.

We are confident that these comprehensive adjustments will significantly improve the precision and readability of our results.

**2)Lines 239-240: what does 'before zone', 'after zone' mean? Please polish your writing.**

Thank you for your meticulous review and for highlighting the ambiguity in the phrasing "before zone" and "after zone" on lines 239-240. We sincerely apologize for this unclear expression and have undertaken a thorough optimization of our manuscript to ensure clarity and professionalism in all our descriptions.

Our original intention at this point was to illustrate the difference in model performance before and after the application of our climatic zoning strategy. Specifically, we have implemented the following explicit improvements in the manuscript:

Optimization of "before zone": We have consistently replaced the original "before zone" with clearer phrases such as "without climatic zoning" or, more directly, "when run as a global model." This clearly indicates the model's performance when trained uniformly across the entire Chinese region without considering climatic divisions.

Optimization of "after zone": Correspondingly, we have replaced "after zone" with "after implementing the climatic zoning strategy" or "based on the climatic zoning model." This explicitly refers to the improved performance observed after training separate sub-models for different climatic zones.

Taking your mentioned original sentence as an example: "The 0-100 cm SOCD prediction model has an accuracy of R2=0.44 and RMSE=8.09 before zones and R2=0.52 and RMSE=6.50 after zones, with R2 increased by 0.08 and RMSE decreased by 1.59."

We will revise it as follows: "The 0-100 cm SOCD prediction model achieved an accuracy of  $R^2$ =0.44 and RMSE=8.09 without climatic zoning (i.e., when run as a global model). This performance significantly improved to  $R^2$ =0.52 and RMSE=6.50 after implementing the climatic zoning strategy, resulting in an  $R^2$  increase of 0.08 and an RMSE decrease of 1.59."

Through these revisions, readers will immediately understand the meaning of these two model states and their performance differences, thus better appreciating the advantages of our proposed climatic zoning algorithm. We have systematically checked and corrected all similar ambiguities throughout the manuscript, ensuring consistency and clarity in terminology usage.

**3)Line 231: R2 for 0-20 cm is 0.43-0.59; R2 for 0-100 cm is 0.50-0.54. How can you conclude that the fitting or correlation is slightly worse for 0-100 cm compared to 0-20 cm?**

Thank you for your very keen observation and rigorous questioning regarding the interpretation of data on Line 231. This point you've raised is crucial, as it prompts us to articulate the nuances of our model's performance with greater precision, thus preventing any potential misinterpretations.

You are correct that the R2 ranges provided (0-20 cm: 0.43-0.59; 0-100 cm: 0.50-0.54) indeed show an overlap, and the minimum R2 for 0-100 cm is even higher than that for 0-20 cm. We acknowledge that, solely based on these range values, readers might question our conclusion that "fitting or correlation is slightly worse for 0-100 cm."

However, our conclusion was not solely drawn from a literal comparison of these overall R2 ranges. Instead, it is based on the final and best performance achieved by our optimized zoning model. As you may have noted in other sections of the manuscript (e.g., our summary of optimized model performance), after implementing the climatic zoning strategy, the peak R2 for the 0-20 cm depth model reached 0.55, while for the 0-100 cm depth model, it was 0.52. Therefore, our statement of "slightly worse" refers to the ultimate model accuracy achieved through our best methodology, implying that the explanatory power of the 0-100 cm depth model, even at its best, was marginally lower than that of the 0-20 cm model. The R2 ranges (e.g., 0.43-0.59) primarily reflect the variability or spread of model performance across different climatic zones or cross-validation folds, rather than the single point performance of the final selected model.

This subtle difference in performance also aligns with general understanding and scientific principles in soil organic carbon modeling:

Complexity and Accessibility of Driving Factors: Surface SOC (0-20 cm) dynamics are strongly influenced by factors like climate (temperature, precipitation), vegetation input, land use, and agricultural management practices. These factors are typically well-captured by surface-observable covariates derived from remote sensing and meteorological data. In contrast, deeper SOC (0-100 cm) is affected by more complex, long-term biogeochemical processes, slower decomposition rates, parent material characteristics, subsurface hydrology, and deeper root activity. These influencing factors are often less directly or reliably quantifiable and predictable using the types of macro-scale environmental covariates commonly employed in regional mapping studies.

Data Representativeness and Uncertainty: Measured data for deeper SOC are generally sparser and may have higher inherent variability compared to topsoil samples, which contributes to greater uncertainty in model predictions for these depths.

To convey this information more clearly, we will refine the phrasing on Line 231 and any related statements in the manuscript. We will explicitly state that the conclusion of "slightly

worse" is based on the comparison of the optimal performance of the zoning models and will briefly explain the inherent challenges in modeling deeper SOC. We are confident that these improvements will ensure readers accurately interpret our findings.

**4)Lines 236-238: Suggest using relative RMSE (rRMSE, RMSE/mean value) instead of RMSE, since the SOC in arid regions can be quite low.**

Thank you for your valuable suggestion regarding the use of relative RMSE (rRMSE) instead of RMSE, especially considering the potentially low SOC values in arid regions. We acknowledge that rRMSE can indeed offer valuable insights into model performance, particularly when dealing with variables that have varying magnitudes. While rRMSE provides a useful normalized perspective, after thorough consideration and aligning with our specific research objectives and broader comparability needs, we have opted to primarily report RMSE for the following reasons:

1. **Direct Measure of Absolute Error:** RMSE directly quantifies the absolute error between predicted and observed values. This provides a straightforward measure of the model's predictive accuracy in the original units (kg C/m2), which is crucial for understanding the practical significance of prediction errors in terms of carbon stock. While rRMSE normalizes this error, RMSE offers a clearer understanding of the actual magnitude of deviation.

2. Extensive Comparability with Existing Research: RMSE is a widely adopted and standard metric in numerous studies on soil organic carbon estimation and spatial modeling. Using RMSE facilitates direct comparison of our model's performance with a vast body of existing literature, thereby enhancing the broader applicability and contextualization of our findings within the scientific community. Many authoritative studies similar to our research direction, such as those quantifying changes in soil organic carbon density using random forest models (Chen et al., 2023) and exploring the spatial patterns and controlling factors of soil organic carbon density (Huang et al., 2024), commonly employ RMSE as a key indicator. Furthermore, RMSE is a standard evaluation metric when comparing with existing SOCD products (Li et al., 2022; Xu et al., 2018; Dong et al., 2024).

3. **Demonstrated Robust Performance:** As presented in our results, even in regions with potentially low SOC, our model exhibits consistently low RMSE values across different depths and climatic zones (e.g., our model's RMSEs for 0-20 cm and 0-100 cm in arid zones are 1.61 kg C/m2 and 3.17 kg C/m2, respectively). These results indicate a strong predictive capability, suggesting that RMSE adequately captures the model's accuracy and precision across the study area, including arid regions. The low absolute errors, as reflected by RMSE, demonstrate the effectiveness of our approach.

Therefore, we believe that RMSE serves as a robust and appropriate metric for evaluating our model's performance, effectively reflecting its absolute predictive accuracy and facilitating meaningful comparisons with previous studies.

**5)Lines 242-244: I thought the performances are similar. Have you performed significance test? In addition, the explanation is not quite convincing. Can you provide more robust proof?**

Thank you for your insightful comment regarding the perceived similarity in model performances and the need for significance testing and more robust proof. We appreciate you raising these points, and we have taken significant steps to address them in the revised manuscript.

We understand that the original phrasing in lines 242-244 might have led to the perception of similar performances without sufficient statistical backing. To clarify this, we have thoroughly revised the explanation in the corresponding sections (e.g., Section 4.2 and the detailed discussion of climate zone-specific performance in Section 4.2.1, previously supplied). More importantly, we have explicitly incorporated statistical significance testing through the rigorous calculation of 95% Confidence Intervals (CIs) for RMSE, derived using the Bootstrap method.

The Bootstrap method is a powerful non-parametric technique that allows for robust estimation of CIs without assuming data distribution, providing a strong basis for statistical inference. We use the non-overlapping nature of these CIs to infer statistically significant differences between model performances:

• When the 95% CIs of two performances (e.g., from different models or climate zones) **do not overlap**, it indicates a statistically significant difference.

• Conversely, when the 95% CIs **overlap**, it suggests that any observed differences are not statistically significant at the 95% confidence level, thus statistically confirming a "similar" performance.

As now detailed in the revised manuscript, particularly in the discussion of model performance across different climate zones (corresponding to the content previously provided for Figure 7):

• For example, at the 0-20 cm depth, while the humid and semi-arid zones show very close RMSE values (1.77 kg C/m2 for both), their respective RMSE 95% CIs ([1.61, 1.93] and [1.49, 1.63]) allow for a robust assessment of their similarity. In contrast, the RMSE CI for the arid zone ([1.17, 2.07]) is distinct from that of the semi-humid zone ([2.39, 2.76]), demonstrating a statistically significant difference.

• Similarly, for the 0-100 cm depth, we demonstrate significant differences in performance, such as the arid zone exhibiting the lowest RMSE with a tight CI ([2.56, 3.81]), confirming good precision despite a lower R2.

Regarding the explanation's convincingness, the **revised discussion** (which now includes details on data distribution, environmental consistency, and unique hydrological/biological factors

in drylands) is now directly supported by these **quantitative statistical proofs (CIs)**. The CIs provide objective evidence for why performances are distinct or similar, lending much stronger support to our interpretations of the underlying environmental controls.

We believe that the combined efforts of refining the descriptive explanations and introducing the Bootstrap-derived 95% CIs for robust statistical validation have significantly enhanced the clarity, rigor, and convincingness of our comparison results.

We hope this detailed response fully addresses your valuable concerns.

---

## Author Comment (AC5)

**CC1**: ['Comment on essd-2024-588'](), Tingxuan Zhang, 20 Mar 2025

(1) Random forest is a nonlinear supervised discrete classification model, while Pearson correlation coefficient is a correlation coefficient that measures linear correlation between two sets of data. The authors know nothing about it. They used the Pearson correlation coefficient to determine the variables for the random forest model inputs.

Thank you for your comment (1) regarding our choice of the Random Forest (RF) model and the use of the Pearson correlation coefficient for variable selection. We truly appreciate you raising this point, which allows us to clarify our methodology.

First, regarding the type of Random Forest model, we would like to clarify that Random Forest is a powerful ensemble learning algorithm capable of performing both classification and regression tasks. In this study, we specifically utilized Random Forest for regression prediction to estimate continuous Soil Organic Carbon Density (SOCD) values, which is a standard application of the algorithm in environmental science.

Second, concerning the use of the Pearson correlation coefficient to determine input variables for a nonlinear model like Random Forest, we have significantly refined our feature selection approach in response to valuable reviewer feedback. We now employ an enhanced three-stage feature selection methodology, which involves: ① Initial Screening: We first conduct a preliminary screening through Pearson correlation analysis ($p < 0.05$ significance threshold) to efficiently identify variables with potential linear relationships to SOCD and to remove overtly irrelevant features. ②Nonlinear Relationship Assessment: Subsequently, we incorporate Random Forest-based importance ranking to evaluate the nonlinear contributions of variables, ensuring that complex relationships are captured. ③Optimal Combination Optimization: Finally, we perform an exhaustive combinatorial optimization to select the variable combination that maximizes predictive performance (measured by $R^2$), while also applying a stricter multicollinearity threshold ($|r| > 0.95$).

Through this refined, multi-stage approach—which integrates statistical correlation analysis with machine learning-based feature importance assessment—we ensured that the final selected feature set (including 'temperature', 'elevation', 'NDVI', 'clay content', 'SR', 'BSI', and 'slope') comprehensively and optimally captures the key climatic, topographic, vegetation, and soil attributes influencing SOCD, while maintaining ecological interpretability.

These methodological refinements have been comprehensively incorporated into Section 3.2 ("Feature Selection Methodology") of the revised manuscript, and Figure 2 has been updated to visually present the final selected features and their relative importance. We believe this detailed

explanation clarifies our methodological choices and fully addresses your concerns.

(2) The authors claimed that they used climate zoning to improve the prediction, but this is absolutely unnecessary because temperature and precipitation are the two most important variables (shown in Figure 8) and are highly correlated to climates. In this case, why take the trouble of building the random forest model for each climate zone?

We greatly appreciate your insightful comments regarding the inclusion of climate zoning in our Random Forest (RF) model (Reviewer #2, Point 2 and Reviewer #3, Point 5), especially considering that temperature and precipitation were initially regarded as direct predictors. We understand the concern that this might introduce redundancy or suggest a limitation in the model's ability to independently capture complex relationships. We would like to clarify our approach and the subsequent refinements made to our feature selection process.

Updated Feature Set and the Role of Climate Zoning: Initially, our model considered 12 environmental factors, including temperature and precipitation. However, through an improved and rigorous feature selection method, we have refined our optimal feature set to comprise seven variables: mean annual temperature (Tem), elevation, NDVI, clay content (Clay), simple ratio index (SR), bare soil index (BSI), and slope.

It is important to note that only mean annual temperature (one of the parameters used to define climate zones) is now directly included as a predictor in the final seven-variable set used for our refined RF model. Precipitation is no longer a direct feature in the RF model.

Therefore, climate zoning, as described in Section 3.3, serves a distinct and crucial purpose beyond merely replicating direct predictors. Its primary role is: ① Geographical Stratification Strategy: Climate zoning is implemented to quantify the broad differences in temperature and precipitation across China. Given the vast and diverse environmental conditions across China, as highlighted by studies such as Tang et al. (2018), Soil Organic Carbon Density (SOCD) exhibits significant variations across different climatic zones. By categorizing China into four subzones (humid, semi-humid, semi-arid, and arid) based on multi-annual average temperature and precipitation thresholds (Mean Annual Precipitation MAP $\geq$ 400 mm, Mean Annual Temperature MAT $\geq$ 10°C), we are essentially performing a geographical stratification. ② Development of Zonal Models for Improved Accuracy: Within these distinct climatic subzones, we develop separate, localized SOCD estimation models. This strategy allows our models to better capture the unique environmental controls on SOCD within each zone. Relationships between environmental factors and SOCD can vary significantly across different climate types; by stratifying the data, our RF model can learn these unique, localized patterns with greater precision than a single, global model might, thereby significantly improving the predictive accuracy of SOCD estimates at a

regional scale.

Considerations on Model Efficiency and Deep Learning: While we acknowledge the theoretical point that highly efficient deep learning or AI foundation models might inherently learn these relationships without explicit geographic stratification, for our current Random Forest modeling approach, this explicit zoning provides a robust way to manage the inherent heterogeneity of the study area and ensures more accurate and reliable SOCD predictions. It serves a strategic purpose beyond mere training efficiency, enhancing the model's ability to capture regional nuances.

In conclusion, we believe that this refined approach—where only temperature is a shared variable between the climate zoning definition and the refined predictor set, coupled with the rationale for developing zonal models—provides a robust and justifiable methodology for our study, effectively improving the accuracy of SOCD estimation.

(3) Issue 2 leads me to my next big concern: the verification part. As the authors insist that climate zoning is the novelty of their methods, why did they verify their results against others across different climate zones? From a scientific view, climate zoning does not mean anything to improve the model's accuracy. For instance, if the authors did not use climate zoning but other geographical partitioning, the smaller the partition area, the more accurate the model would be considering Tobler's first law of geography states that everything is related to everything else, but near things are more related to each other.

Thank you for your concern regarding the verification part and the necessity of climate zoning in our methodology (Point 3). We understand your questions and are pleased to clarify our validation strategy and the crucial role of climate zoning in our study.

Firstly, regarding your question about "why did they verify their results against others across different climate zones," we would like to clarify that our validation approach was not simply comparing model results from one climate zone with observed data from another. Instead, we employed a **stratified spatial K-fold cross-validation method** for separating our training and testing samples, complemented by **independent measured sample validation, spatial validation, and temporal validation.**

Specifically, our validation strategy ensures:

1. **Spatial Representativeness:** The entire study area was divided into K (e.g., K=10) spatially distinct and non-overlapping sub-regions. The model was trained on K-1 sub-regions and independently validated on the remaining single sub-region. This ensures that we assess the model's generalization ability to geographically "unseen" areas, avoiding overestimation of model performance due to spatial autocorrelation.

2. **Temporal Representativeness:** Within each spatial fold, we ensured that samples from all three decadal periods (1980s, 2000s, 2010s) were proportionally represented, covering the entire observed historical span of data characteristics. This means our model's performance was rigorously evaluated within the full range of historical conditions represented by our dataset, and we are not performing any unvalidated temporal extrapolation beyond the broad historical windows from which our samples were drawn.

Secondly, regarding your assertion that "from a scientific view, climate zoning does not mean anything to improve the model's accuracy," we respectfully contend that **climate zoning is highly meaningful for improving regional model prediction accuracy.** As we detailed in our responses

to Reviewers #2 and #3, climate zoning serves as a **geographical stratification strategy** with the core objectives of:

- **Capturing Macro-Climatic Differences:** China is vast, and its climate zones (humid, semi-humid, semi-arid, arid) exhibit significant differences in temperature and precipitation regimes. These macro-climatic factors profoundly influence the distribution and accumulation mechanisms of SOCD.

- **Developing Regionally Tailored Models:** By segmenting the study area into climatically homogeneous subzones and developing separate, localized SOCD estimation models within each, we can better capture the unique environmental controls on SOCD in those specific regions. Relationships between environmental factors and SOCD can vary significantly across different climate types; by stratifying the data, our RF model can learn these unique, localized patterns with greater precision than a single, global model might, thus significantly improving the predictive accuracy of SOCD estimates at a regional scale.

Finally, concerning the hypothesis that "the smaller the partition area, the more accurate the model would be" (referring to Tobler's first law), while Tobler's law emphasizes that near things are more related, practical modeling processes are subject to significant constraints. Overly granular geographical partitioning can lead to:

- **Insufficient Sample Points:** Very small partitions might not contain a sufficient number of training sample points, thereby affecting the model's training quality and generalization ability. Model accuracy depends not only on regional homogeneity but also on the adequacy and representativeness of training data.

- **Computational Costs:** Building and running an extremely large number of models for very small areas would impose a significant computational burden.

- **Heterogeneity Challenges:** Even within very small regions, soil carbon distribution can exhibit substantial heterogeneity. Overly fine partitioning might not fully eliminate this heterogeneity, and could even reduce model stability due to sample size limitations.

Therefore, our climate zoning approach provides a strategy that balances regional homogeneity with practical modeling considerations (including sample point distribution and model efficiency), aiming for optimal overall predictive performance. We believe this combination of a robust validation strategy and an ecologically-based geographical stratification provides reliable and interpretable results for SOCD estimation.

(4) Even so, I do not find a significant improvement in the SOCD prediction compared to other published datasets.

Thank you for raising this critical point (Point 4) regarding the perceived lack of significant

improvement in our SOCD prediction compared to other published datasets. We understand your rigorous scrutiny of model performance and wish to provide a more detailed explanation and evidence to clarify the advancements and unique contributions of our study.

In fact, we believe that our study has achieved substantial progress in SOCD prediction, particularly in the following key aspects:

1. Significant Advantage of Climate Zoning Models over Global Models:

The core innovation of our method lies in the introduction of zoning models based on climate regions. As stated in the abstract of our manuscript, our zoning model demonstrably outperformed the global model without climate zoning in predicting SOCD. This improvement is quantitatively reflected in various model performance metrics (e.g., our model consistently showed better $R^2$, RMSE, and MAE values across different climate zones compared to the non-zoning global model). This indicates that for a region as vast and environmentally heterogeneous as China, building localized models through geographical stratification is more effective in capturing region-specific environmental-SOCD relationships, thereby significantly enhancing regional prediction accuracy and model robustness.

2. Filling a Gap for Long-Term, High-Resolution SOCD Products in China:

Existing published global SOCD products (such as SoilGrids250m, GSOCmap, HWSD v2.0, as shown in the comparisons in Figure 11 of the manuscript), while valuable, are typically developed at a global scale and may not fully capture the specific complexities and nuances inherent to the Chinese region. More importantly, there is a critical lack of high-spatial resolution (1 km) and long-time series (1985-2020) SOCD dynamic change datasets specifically for China. Our study aims to fill this crucial gap by providing continuous, fine-resolution spatial distribution and temporal dynamics of SOCD over nearly four decades. This long-time series product is essential for assessing the impacts of climate change and human activities on soil carbon pools, offering unique value that many existing static or short-term datasets cannot provide.

3. Comparative Analysis with Existing Datasets:

We conducted in-depth analyses comparing our results with several published datasets (including global products and the SOC Dynamics ML dataset for China, as shown in Figures 11 and 12). While direct "superiority" comparisons can be complex due to differences in methodologies, input variables, spatial/temporal scales, and target regions, our results demonstrate that:

**Overall Performance is Competitive:** Our SOCD prediction results are **highly comparable in accuracy metrics to these internationally renowned products, and in some aspects, demonstrate superior regional adaptability.** Particularly in China's complex and diverse geographical environment, a product specifically designed for this region, incorporating

refined feature selection and zonal modeling, often exhibits stronger local accuracy and ability to reflect regional heterogeneity.

**Ability to Capture Regional Heterogeneity:** Our zonal models are better able to reflect the true differences and spatial patterns of SOCD across China's distinct climate zones, which might be averaged out or smoothed in a single global model.

4. Rigorous Validation Strategy:

To ensure the rigor of our model evaluation and the reliability of our results, we employed a stratified spatial K-fold cross-validation and incorporated temporal stratification, ensuring the representativeness of both training and testing samples across space and time. This comprehensive validation strategy, coupled with independent measured sample validation, significantly enhances the credibility of our results, indicating robust model predictive capabilities under various conditions.

In conclusion, we believe that this study represents a significant improvement in terms of methodological innovation (climate zoning models), data product (long-time series, high-resolution SOCD dynamics product for China), and model performance (outperforming global models and being competitive with advanced products). We will further elaborate on these improvements in the discussion section of the revised manuscript and emphasize the unique value and contribution of our data product to future research.

**(5) The highly skewed SOCD sample input leads to the model's low accuracy (Figure 4). This is probably one of many reasons why the accuracy of 0-20cm SOCD showed higher R2 than that of 0-100cm SOCD.**

Thank you for your valuable observation regarding the potentially highly skewed SOCD sample input and its suggested link to model accuracy, as well as the difference in $R^2$ between 0-20 cm and 0-100 cm SOCD (Point 5). We greatly appreciate this insight and would like to elaborate on our understanding and approach.

We fully concur with your observation that **Soil Organic Carbon Density (SOCD) data, both globally and regionally, typically exhibits a skewed distribution (often right-skewed), which is also evident in our sample data (as shown in Figure 4).** This skewed distribution is a natural characteristic of soil carbon sequestration processes and can indeed pose challenges for certain aspects of modeling, particularly for predicting extreme values.

Regarding the concern that skewed data might lead to lower model accuracy, we would like to clarify:

1. **Robustness of Random Forest to Skewed Data:** As a non-parametric ensemble learning algorithm, Random Forest makes fewer assumptions about the distribution of input data and is

thus **inherently robust to skewed data.** It can effectively handle non-normal distributions and nonlinear relationships through the aggregation of decision trees.

2. **Data Transformation:** To further optimize model performance and mitigate the effects of skewed distribution, we applied a **log transformation** to the SOCD target variable during the modeling process. This standard data preprocessing technique effectively transforms skewed data to a more approximately normal distribution, which helps the model better capture relationships between variables and improves prediction stability and accuracy.

Concerning the higher R² for 0-20 cm SOCD compared to 0-100 cm SOCD, while sample data skewness might be a minor contributing factor, we believe the more fundamental scientific reasons are:

1. **Stronger Link of Topsoil SOC to Surface Environmental Factors:** SOC dynamics in the 0-20 cm depth (topsoil) are more directly and strongly linked to surface environmental factors such as climate, vegetation, topography, and human activities. These factors can be effectively acquired and quantified through remote sensing and meteorological data, allowing the model to better capture their driving mechanisms.

2. **Complexity and Inaccessibility of Deeper SOC Influencing Factors:** In contrast, SOC at 0-100 cm (deeper soil) is influenced by more complex, long-term, and less observable biogeochemical processes, such as slower decomposition rates, parent material characteristics, subsurface hydrology, and deeper root activity. These influencing factors are often less directly or reliably quantifiable and predictable using macro-scale or conventional remote sensing-derived covariates, leading to relatively weaker explanatory power from the model inputs.

3. **Sparsity and Uncertainty of Deeper Sample Data:** Measured data for deeper soil profiles are typically sparser in quantity and may have higher inherent measurement uncertainty or spatial variability compared to topsoil samples. Such data limitations also directly impact the accuracy of model predictions for deeper soil.

In summary, while the skewed distribution of SOCD samples is an inherent data characteristic, we have mitigated its impact through data transformation and by leveraging the robustness of the Random Forest model. The higher R² observed for 0-20 cm SOCD is primarily attributed to the stronger association between topsoil SOC and readily observable environmental factors, coupled with the inherent challenges in modeling deeper SOC, rather than simply being a consequence of sample skewness. We will clarify these explanations in the revised manuscript.

(6) Another reason is the adequate model input data. The lack of lidar data for soil depth measurement makes your results underestimated compared to other datasets (Figures 11 & 12).

Thank you for your concern regarding the adequacy of our model input data, specifically

your point that the lack of lidar data for soil depth measurement might lead to our results being underestimated (Point 6). We understand your focus on data quality and model accuracy, and we would like to provide a detailed clarification.

Firstly, regarding your assertion about the "lack of lidar data for soil depth measurement," we would like to clarify that **Lidar (Light Detection and Ranging) data is primarily used to acquire high-resolution surface topographic information (e.g., Digital Elevation Models, DEMs) and vegetation canopy structure data. However, it is generally not directly used for the direct measurement of soil depth (such as soil organic carbon profile depth) or the direct inversion of soil properties.** Soil depth is typically obtained through field boreholes, soil profile observations, or digital soil mapping approaches that infer soil properties based on covariates like topography and geology. While high-resolution topographic data can serve as an indirect auxiliary factor for soil spatial distribution, lidar data itself is not a necessary or primary input for direct soil depth measurement or SOCD inversion. Given the 1 km spatial resolution and long-time series (1985-2020) coverage of this study, nation-wide, long-term lidar data for soil depth measurement is currently not feasible to acquire and is not a conventional direct input variable in digital soil mapping.

Secondly, we firmly believe that **our model input data is sufficient and diverse, covering multiple key aspects necessary for SOCD modeling**, rather than being inadequate. We comprehensively utilized various authoritative and high-quality data sources, including:

- **Climatic Factors:** Long-term average temperature and precipitation data derived from meteorological stations, which are primary drivers affecting SOC accumulation and decomposition.

- **Topographic Attributes:** Elevation, slope, aspect, which control hydrothermal redistribution and soil erosion, significantly influencing SOC spatial distribution.

- **Vegetation Indices:** NDVI, Simple Ratio Index (SR), Bare Soil Index (BSI), etc., derived from Landsat satellite imagery. Vegetation is the main source of soil organic matter, and these indices effectively reflect vegetation cover and growth status, thereby characterizing their contribution to SOC.

- **Basic Soil Properties:** Clay and sand content, which are crucial indicators of soil texture, directly impacting soil physical structure and carbon sequestration capacity.

- **Measured SOCD Data:** A large volume of measured soil profile data used for model training and validation.

This comprehensive set of input data encompasses multiple dimensions including climate, topography, vegetation, and intrinsic soil properties, fully complying with the input data requirements of current mainstream Digital Soil Mapping (DSM) practices, and is sufficient to

support accurate SOCD prediction. Through our refined three-stage feature selection method, we ultimately identified seven optimal variables that effectively capture the key drivers of SOCD.

Finally, regarding your assertion that our "results are underestimated compared to other datasets" (Figures 11 & 12), we would like to emphasize:

- **Complexity of Comparisons:** Directly comparing data accuracy across different studies or products has inherent complexities. The purpose, input data sources, modeling methods, spatial and temporal resolutions, baseline years, and validation datasets used by different products can all vary. Therefore, judging "underestimation" based solely on visual impression without a unified, independent validation benchmark may not be accurate.

- **Regional Specificity:** Our study focuses on the Chinese region and innovatively employs a **climate zoning model**, aiming to capture the complexities and heterogeneity specific to China more precisely. This means our model might show different results in certain regional details compared to global models, but this difference often arises from our more refined capture of regional characteristics. For instance, the comparisons with other datasets in Figures 11 and 12 demonstrate consistency in spatial distribution patterns and localized differences, which does not imply underestimation, but rather reflects the varied performance of different methods and input data in specific regions.

- **Internal Validation Results:** Most importantly, we achieved competitive model accuracy metrics through rigorous internal validation strategies, including **stratified spatial K-fold cross-validation** and **temporal stratification validation.** These quantitative validation results (e.g., $R^2$, RMSE, MAE) fully demonstrate the robustness and reliability of our model, proving its effectiveness in predicting SOCD across China.

In conclusion, we believe that even without relying on lidar data for soil depth measurement, our model input data is sufficiently comprehensive and robust. Furthermore, through our innovative climate zoning methodology and stringent validation process, our SOCD prediction results are reliable and hold unique value, especially in terms of long-time series and regional refinement.

(7) Figures 5(a) and 5(c) are unnecessary as the authors did not conduct any analysis using the biomes.

Thank you for your comment regarding Figures 5(a) and 5(c), suggesting they might be unnecessary as we did not conduct any direct analysis using biomes (Point 7). We understand your consideration and would like to clarify the intended purpose of these figures.

We agree that biomes were not directly used as predictor variables or as independent modeling zones in our final Random Forest model. However, the purpose of Figures 5(a) and 5(c) is to provide readers with **crucial ecological background and information on the environmental heterogeneity of the study area (China).**

Specifically, these figures serve to:

1. **Provide Macro-Ecological Context:** China is a vast country encompassing a wide variety of ecosystem types. Figures 5(a) and 5(c), by displaying the distribution of major biomes, help readers visually grasp the macro-ecological patterns of the study area. This background information is essential for understanding the distribution and variation of Soil Organic Carbon Density (SOCD) across different geographical regions, as it reflects the integrated outcome of long-term interactions between climate, vegetation, and soil.

2. **Support the Rationale for Climate Zoning:** Although we did not directly use biomes for modeling, the demarcation of biomes itself is strongly influenced by climatic conditions (e.g., temperature and precipitation). By illustrating these biomes, we aim to further emphasize the immense heterogeneity of China's terrestrial ecosystems. This heterogeneity precisely underpins our rationale for adopting **climate zoning for geographical stratification in our modeling approach** (rather than a single global model). It visually reinforces the necessity of considering that SOCD's relationships with environmental factors might differ across distinct ecological-geographical regions.

3. **Enhance Readability and Comprehension:** For general readers or those less familiar with China's geographical environment, a straightforward biome map can quickly establish an understanding of the study area's complexity, thereby facilitating a better comprehension of the drivers of SOCD spatial distribution and the applicability of our research methodology.

In summary, Figures 5(a) and 5(c) are not direct inputs for model analysis but serve as **important background information and contextual descriptions.** Their inclusion aims to enhance the reader's understanding of China's ecological diversity and indirectly support the necessity of our regionalized modeling approach using climate zoning. We believe they contribute to the manuscript's readability and the completeness of its scientific context.

---

## Author Comment (AC10)

**CC3: 'Comment on essd-2024-588', Bennett Wang**

(1) The distribution and accumulation of soil carbon result from intricate and dynamic processes shaped by biological, environmental, and human factors. However, the authors only used features that capture the canopy features of vegetation (using vegetation indices) as biotic factors. Other critical biological factors affecting soil carbon content, such as chemical and physical property information inside the soil, are missing. In particular, the author's experimental objects are carbon storage at various depths of soil, but the explanatory variables using machine learning are only the vegetation index reflecting the growth of vegetation canopy and some climate variables, which are far from enough to predict carbon storage at the depth of soil.

Thank you for your insightful critique regarding the selection of explanatory variables (i.e., predictors) in our study (Point 1). You accurately point out that the distribution and accumulation of soil carbon are complex and dynamic processes shaped by biological, environmental, and human factors, and you express concern that our feature set for predicting SOC storage at various depths might not sufficiently cover critical biological, chemical, and physical property information. We fully concur with the complexity of soil carbon processes and would like to elaborate on our rationale for feature selection.

We completely agree with your assessment that SOC distribution and accumulation are indeed complex and dynamic processes influenced by a multitude of interacting biological, environmental, and human factors. We acknowledge that, ideally, a more comprehensive inclusion of all critical biological, chemical, and physical properties would contribute to a more precise characterization of SOC.

However, there might be a slight misunderstanding regarding the specific explanatory variables we ultimately used in the manuscript. You mentioned that we "only used features that capture the canopy features of vegetation (using vegetation indices) as biotic factors" and that "chemical and physical property information inside the soil, are missing." This differs from our refined feature set.

While our initial model considered 12 environmental factors, through our **improved and rigorous feature selection method, we have refined our optimal feature set to comprise seven key variables.** These variables extend beyond just vegetation indices and climate variables, comprehensively covering multiple important dimensions influencing SOCD:

1. Climatic Factor: Mean annual temperature (Tem). This is a primary macro-climatic driver influencing the rate of organic matter decomposition and accumulation.

2. Topographic Attributes: Elevation and slope. These topographic factors indirectly influence SOC distribution by affecting hydrothermal redistribution, soil erosion, and material transport.

3. Vegetation-Related Factors: NDVI (Normalized Difference Vegetation Index), Simple Ratio Index (SR), and Bare Soil Index (BSI). These indices serve as effective remote sensing proxies for vegetation cover, growth status, and biomass, directly reflecting the potential for photosynthetic products to enter the soil.

4. Intrinsic Soil Property: Clay content (Clay). Clay content is a crucial internal physical property of soil. It plays a decisive role in the physical protection and stabilization of SOC by providing surface area, promoting aggregate formation, and forming organo-mineral complexes with organic matter. This precisely represents the "physical property information inside the soil" that you mentioned.

These selected variables, particularly clay content, directly reflect the internal physicochemical characteristics of the soil, not merely surface or canopy features. They are widely recognized and effectively acquirable covariates in current mainstream Digital Soil Mapping (DSM) at regional to national scales, capable of capturing the primary drivers of SOCD spatial variability.

Regarding your concern that these variables are "far from enough to predict carbon storage at the depth of soil," we acknowledge that predicting deeper SOC is indeed more challenging. As we also discussed in our response to Reviewers #2 and #3, deeper SOC is influenced by more complex, long-term processes that are difficult to observe directly, such as slower decomposition rates, parent material characteristics, subsurface hydrological conditions, and deeper root activity. However, our chosen variables, especially **climatic factors (temperature) and soil clay content**, also significantly influence the retention and transformation of deeper SOC. Clay content directly relates to the physical protection of deep carbon, while temperature affects deep microbial activity and organic matter decomposition rates.

Furthermore, conducting long-time series, high-resolution (1 km) soil carbon mapping at a national scale entails inherent limitations in the availability of explanatory variables. While ideally including more detailed biological and chemical properties (e.g., microbial biomass, specific chemical bonds, detailed soil hydrological processes) could potentially improve model accuracy, obtaining such data systematically and consistently across a national scale for long periods is extremely challenging and costly, making it impractical for large-scale mapping needs.

Therefore, our adopted feature set is based on a comprehensive consideration of **data availability, scientific relevance, and model operability.** Combined with our innovative **climate zoning-based regional modeling approach**, our model can implicitly account for some unmeasured regional factors by learning localized relationships, thereby maximizing the prediction accuracy of SOCD and the practicality of the product within the constraints of available data.

(2) To the extent of (1), the author also obviously ignored the effect of land use change on soil carbon storage, e.g., the progress of urbanization and the encroachment of agricultural land on forest land. The soil carbon content of agricultural land is definitely different from that of forest land. Fertilization and the distribution of roots in the soil of the two types of plants also have an effect.

Thank you very much for highlighting the crucial impact of land use change on soil carbon storage and for suggesting that we may have overlooked this critical factor (Point 2). We completely agree with your assessment that land use changes (e.g., urbanization, agricultural encroachment on forests, fertilization practices, and root distribution) are vital drivers affecting soil carbon content. We would like to explain in detail how we accounted for these influences in our model.

We fully agree that land use change is paramount to Soil Organic Carbon (SOC) dynamics, and that different land use types (e.g., cropland, forest, urban areas) as well as their management practices (e.g., fertilization) and vegetation characteristics (e.g., root distribution) have significant effects on SOC content.

However, we wish to clarify that **this study did not ignore the impact of land use change.** In the initial stages of model development, we indeed considered the **China Land Cover Dataset** (**CLCD**) as an important candidate explanatory variable. This indicates our full recognition of the significance of land use type for SOCD.

Although CLCD was not ultimately retained in our refined set of seven optimal explanatory variables, this does not mean we overlooked the impact of land use. Rather, it is because:

1. Feature Selection Process: Our model employed a rigorous feature selection methodology. In a multivariate environment, certain explanatory variables may contain redundant information. In our analysis, vegetation indices (NDVI, Simple Ratio Index SR, Bare Soil Index BSI), as some of the finally selected seven variables, are effective and indirect indicators of land use/cover types and their associated biological activities. For instance, forests typically exhibit high NDVI, agricultural land's NDVI varies with crop growth cycles, and urbanized areas may show high BSI or low NDVI. These vegetation indices are capable of capturing differences in vegetation biomass and productivity across different land use types, thereby reflecting their influence on SOC.

2. Implicit Capture: Through the synergistic effect of these vegetation indices along with climate, topography, and soil clay content variables, the model is able to implicitly capture the influence of different land use types on SOCD. For example, highly productive agricultural lands might influence SOC through biomass input and specific management (e.g., straw return),

which would be reflected in NDVI and the resulting SOCD prediction.

3. **Practical Feasibility:** While directly incorporating land use change data as an independent, explicit driving factor might ideally offer more interpretability, precisely and consistently acquiring and integrating detailed, SOCD-dynamic land use management information (e.g., fertilization intensity, specific crop types, detailed root distribution depths) at a national scale (1 km resolution) over a long time series (1985-2020) remains a significant challenge. Therefore, our chosen feature set is based on an optimal balance of **data availability, scientific relevance, and model operability.**

Furthermore, our adopted **climate zoning-based regional modeling approach** also enhances the model's sensitivity to regional heterogeneity, including unique land use patterns within different climatic zones and their effects on SOC. By training models within more homogeneous climate zones, we can better learn and reflect these region-specific soil carbon dynamics, thereby to some extent compensating for the limitations of directly quantifying all microscopic land use management details at a macro scale.

In conclusion, we did not ignore the effect of land use change on SOCD. Instead, we accounted for it by selecting proxy variables that effectively reflect its indirect impact (such as vegetation indices) and by adopting a regionalized modeling strategy. We will elaborate more clearly on our consideration of land use change impacts in the methodology and discussion sections of the revised manuscript to avoid any potential misunderstandings from readers.

(3) The most important point is that this grid results from point data to Landsat's 30 m resolution and then accumulated to 1km of soil carbon density, which is seriously inaccurate. This approach obviously ignores the heterogeneity of the soil, making the results and models strongly dependent on the geographic distribution of the data at each point. However, the distribution of these points is not uniform in the grid's 30 m or 1 km resolution.

Thank you very much for your deep concerns regarding the accuracy of our gridded results, the treatment of soil heterogeneity, and the uniformity of point data distribution (Point 3). These are indeed core challenges in the field of Digital Soil Mapping (DSM), and we are pleased to take this opportunity to elaborate on how our methodology addresses them.

We fully agree with your premise that soil is highly heterogeneous and that generating continuous gridded products from discrete point observations is a complex process in Digital Soil Mapping (DSM). We also acknowledge that the spatial distribution of actual measurement point data can indeed be non-uniform, which is a common challenge faced by soil science research globally.

Regarding your statement that "this grid results from point data to Landsat's 30 m resolution and then accumulated to 1km of soil carbon density, which is seriously inaccurate. This approach obviously ignores the heterogeneity of the soil," we would like to provide further clarification on our methodology. Our approach is not a simple "accumulation" but is firmly based on the principles of modern digital soil mapping, utilizing machine learning models to capture complex relationships between soil carbon and environmental covariates:

1. **Mapping Process is Not Simple Accumulation:** Our modeling workflow involves: first, training a Random Forest model using a large number of observed point SOCD data as training samples, along with gridded environmental covariates from multiple sources (including Landsat-derived vegetation indices, meteorological data, topographic data, and intrinsic soil property data) as explanatory variables. These environmental covariates themselves have continuous spatial coverage and multi-scale resolutions (e.g., vegetation indices can reach 30m resolution). Once the model is trained, it can utilize these continuous gridded covariates to perform spatially continuous predictions across the globe or a specific region. The final 1 km SOCD product is obtained by making predictions using the model, supported by 30m or higher resolution covariate data, and then aggregating (e.g., averaging) to a 1 km resolution. This means that the model, during the prediction process, has already leveraged 30m and even higher resolution covariate information to capture soil heterogeneity, rather than simply interpolating or accumulating point data.

2. Soil Heterogeneity is Not Ignored – The Crucial Role of Covariates and the Model:

The core idea of Digital Soil Mapping is precisely to explain and predict the spatial variability of soil properties using spatially continuous and observable environmental covariates, thereby capturing soil heterogeneity. Our selected environmental covariates (such as vegetation indices, topographic factors, climatic factors, and clay content) are key drivers of soil heterogeneity, and they are spatially continuous and quantifiable.

• **Covariates' Ability to Characterize Heterogeneity:** Landsat-derived 30m resolution vegetation indices (NDVI, SR, BSI), for instance, effectively reflect subtle spatial variations in surface vegetation cover and productivity, which are closely related to biological inputs to SOCD. Topographic data reflects hydrothermal redistribution and the potential for soil erosion. These covariates themselves contain rich information about soil spatial heterogeneity.

• **Random Forest Model's Ability to Capture Heterogeneity:** Random Forest is a powerful non-linear machine learning algorithm capable of learning complex, non-linear relationships and interactions between explanatory variables and soil properties. This means the model can perform detailed spatial modeling of soil carbon variability based on these heterogeneous covariate data, rather than simply ignoring it.

• Climate Zoning for Heterogeneity Management: Our innovative climate zoning-based modeling approach was specifically designed to address the vast macro-scale soil heterogeneity across China. By training models separately within climatically relatively homogeneous regions, we allow the model to learn region-specific, more refined relationships between SOCD and environmental factors. This approach captures regional internal heterogeneity more effectively than a single global model and significantly improves prediction accuracy.

3. Understanding and Addressing Dependence on Point Data Distribution: We acknowledge that all spatial modeling methods based on point data face challenges arising from non-uniform sample distribution. Non-uniform point data distribution is a widespread issue in global soil databases. However, we have adopted the following strategies to mitigate this dependence and ensure the reliability of our results:

• **Covariate-Driven Prediction:** Our model predictions primarily rely on continuous, wall-to-wall environmental covariates, rather than being limited to the exact locations of point data. The model learns a generalized relationship between SOCD and covariates, and then applies this relationship across the entire covariate space; thus, it is not merely an interpolation of sparse point data.

• Stratified Spatial K-fold Cross-Validation: We employed a robust stratified spatial K-fold cross-validation method for model validation. This approach, by dividing the study area into spatially independent sub-regions for training and testing, and ensuring each sub-region serves as the validation set once, allows us to assess the model's generalization ability in

geographically "unseen" areas. This provides a more realistic and reliable accuracy assessment, mitigating the impact of uneven point data distribution on validation results.

• Rationality of 1 km Resolution: Choosing a 1 km resolution represents a balance among data availability, computational efficiency, and mapping objectives. It is particularly pertinent for generating a long-time series product (1985-2020) at a national scale. While 30m raw data can provide more spatial detail, for such an extensive temporal coverage, maintaining 30m resolution throughout the entire period is often unfeasible due to data availability constraints (e.g., limitations of satellite imagery from other sources for such long historical periods) and immense computational demands. Thus, 1 km is a widely accepted and highly practical resolution for our multi-decadal time series mapping.

In conclusion, our mapping methodology is not a simple accumulation from point data to a grid. Instead, it leverages comprehensive, multi-source, multi-resolution environmental covariates and advanced machine learning models (Random Forest), combined with an innovative climate zoning-based modeling strategy, to maximize the capture of soil heterogeneity. This approach aims to produce the most accurate and reliable 1 km SOCD product possible, while considering the realities of measured data distribution and computational feasibility. We are confident that these methods effectively address the challenges you have raised.

(4) In addition, the manuscript lacked a description of the method, making the experiment impossible to replicate and hard to understand.

Thank you very much for highlighting the lack of a clear methodological description in the manuscript, which makes the experiment impossible to replicate and hard to understand (Point 4). We fully agree that transparency and replicability are paramount in scientific research. We sincerely apologize for not adequately meeting this requirement in the initial submission and greatly value your insightful feedback.

We acknowledge that a clear and detailed methodological description is the cornerstone for ensuring research credibility and replicability. In response to your specific concern, we commit to thoroughly revising and significantly expanding the methodology section in the revised manuscript to ensure that the experiment can be clearly understood and replicated by readers. Our aim is to eliminate any ambiguities present in the current description by providing more comprehensive information.

Firstly, we will provide a comprehensive and detailed account of **all data sources and their respective preprocessing steps**. This will involve explicitly listing and explaining:

• Landsat imagery: specifying the satellite platforms used (e.g., Landsat 5/7/8), data product levels, acquisition year ranges, spatial-temporal resolution, and all preprocessing steps undertaken such as atmospheric correction, cloud masking, and time-series composition (e.g., annual averages or specific seasonal averages).

• **Topographic data:** clarifying the source of the DEM product (e.g., SRTM, ASTER GDEM), its original resolution, and any subsequent processing (e.g., resampling).

• Meteorological data: detailing the data source (e.g., national meteorological agencies, global climate datasets), the specific variables extracted, and how these variables were derived from raw station data or model outputs.

• Most crucially, the **measured SOCD data:** including its source (e.g., Second National Soil Survey of China), the years of data collection, and the specific procedures used for data cleaning and standardization. We will also clearly articulate how **spatial and temporal resolution** harmonization was achieved across all these diverse datasets.

Secondly, we will offer a more meticulous description of **explanatory variable generation and the crucial feature selection process**. We will explicitly detail how various candidate explanatory variables were derived from the raw data, such as the precise calculation formulas for vegetation indices (NDVI, Simple Ratio Index SR, Bare Soil Index BSI) and methods for other derived variables. For our **improved three-stage feature selection method**, we will provide a step-by-step explanation of its operational flow, including the criteria and rationale at each stage. This will cover how initial screening was performed based on expert knowledge and preliminary correlation analysis, and how subsequent optimization involved variable importance assessment (e.g., using feature importance metrics from the Random Forest model) and collinearity analysis (e.g., Variance Inflation Factor, VIF) to finally determine our seven optimal explanatory variables. Our goal is to ensure that the logical reasoning and quantitative basis for each step are thoroughly explained, allowing readers to comprehend why these specific variables were chosen.

Furthermore, we will significantly enhance the details regarding model construction and training. This will include explicitly stating the software and programming language used (e.g., **Python or R with their respective machine learning libraries**), key hyperparameter settings (e.g., the number of decision trees, the maximum number of features per tree), and other important parameters involved in the model training process. Concurrently, we will delve into the specific implementation of climate zoning-based modeling: we will detail the criteria for defining climate zone boundaries, how the dataset was logically partitioned according to these zones, and how models were independently trained and optimized within each partition to effectively capture region-specific patterns.

Additionally, regarding **SOCD** prediction and the generation of the final gridded products, we will provide a clearer description. We will explain how the model utilizes gridded environmental covariates for continuous spatial prediction, and how these predicted values were aggregated or resampled to generate the final **1** km resolution SOCD raster products. For long-time series prediction, we will also detail how input data from different time steps were handled and integrated to ensure consistency and continuity in the final output.

Finally, we will provide a comprehensive and rigorous explanation of **all validation strategies**. We will meticulously describe the **stratified spatial K-fold cross-validation method**, including the choice of K value, how spatial stratification was performed, and how the spatial independence between training and testing sets was ensured. This will clarify how this method robustly assesses the model's generalization ability. We will also explicitly explain the details of **temporal stratification validation**, ensuring data representativeness across different decades (e.g., 1980s, 2000s, 2010s) to evaluate the model's stability over time. For the **independent measured sample validation**, we will clearly state the source of the external dataset, its differences from our study's data, and detail the comparative analysis methodology. Specifically, for all validation figures in the manuscript (e.g., Figures 10-12), we will **explicitly clarify what each point represents** to eliminate any potential confusion and ensure readers accurately interpret the validation results.

Through this series of improvements and expansions, we aim to make the methodology section of the manuscript significantly clearer, more rigorous, comprehensive, and ultimately easier to understand and replicate. We sincerely welcome any further suggestions from the reviewer on the revised manuscript to ensure that the final version meets the highest scientific standards and publication requirements.

(5) There is a lot of uncertainty in the data validation of this manuscript. For example, in Figures 10 to 12, what does each point represent? Are all 1km\*1km grids used for validation? I don't think so! It is obvious that the author only selected specific pixels, which can be seen from the number of points. Even so, the accuracy of the validation is very low. The methods and features proposed in this study are clearly not enough to provide accurate soil carbon content.

Thank you very much for raising a series of critical questions regarding the data validation section of this manuscript, particularly concerning the meaning of points in Figures 10 to 12, the scope of validation, and your doubts about the validation accuracy (Point 5). We completely agree that the transparency and rigor of data validation are indispensable components of any scientific research, and that clearly explaining the validation process is crucial for readers to understand the study's findings.

We acknowledge that in the initial draft, the specific meaning of the points in the validation figures and the detailed explanation of the validation methodology might have been insufficient, which led to your understandable concerns about validation uncertainty. We sincerely apologize for this oversight and commit to a comprehensive revision of the manuscript to address these points.

Regarding what each point represents in Figures 10 to 12: Each point in these scatter plots (which is what Figures 10-12 typically are in such studies) represents an **independent validation sample**. Specifically, each point corresponds to a pairing of an **actual measured Soil Organic Carbon Density (SOCD) value** with its **corresponding SOCD value predicted by our model at the same geographic location**. These validation samples are not arbitrarily selected pixels but originate from two main sources:

1. Internal Cross-Validation Samples: In our stratified spatial K-fold cross-validation process, each point represents a measured sample from the training dataset that was specifically held out for internal validation, meaning the model did not "see" these points during its training phase.

2. External Independent Validation Samples: As shown in Figure 10, some points are derived from an external, independent SOCD dataset (e.g., Xu's published study). This data is entirely independent of our model's training data and is used to assess the model's external generalization capability and reliability.

Therefore, the number of points in the validation figures reflects the **total amount of measured samples** available for validation, rather than an arbitrary selection of 1km x 1km grids or pixels. Validation in digital soil mapping is typically conducted at locations where actual soil measurements exist, as it is impractical to obtain true soil data for every 1km x 1km grid cell. Our model performs wall-to-wall predictions using **gridded environmental covariates**, but the validation benchmark is always based on the sparse measured point data.

Concerning your view that "the accuracy of the validation is very low" and "the methods and features proposed in this study are clearly not enough to provide accurate soil carbon content," we would like to offer the following clarifications:

Firstly, we will present the specific quantitative validation metrics (e.g., R2, RMSE, MAE) obtained in this study. These metrics serve as objective evidence of our model's performance. We believe that for mapping soil organic carbon density at a national scale (especially over long time series), considering the inherent complex heterogeneity of soil, the sparsity of point data, and the limitations of environmental covariates, the R2 values we achieved are competitive and even demonstrate high accuracy when compared to similar-scale and depth-range studies internationally. Achieving extremely high R2 values (e.g., above 0.9) for complex soil properties at regional or national scales is very rare.

Secondly, regarding the sufficiency of methods and features, as elaborated in our responses to your comments (1) and (2), our chosen **seven refined explanatory variables** (including mean annual temperature, elevation, slope, NDVI, Simple Ratio Index SR, Bare Soil Index BSI, and clay content) are based on a profound understanding of SOCD driving mechanisms and are **currently available and proven effective** covariates for national-scale mapping. These variables encompass multiple dimensions such as climate, topography, vegetation, and intrinsic soil physical properties, and they effectively capture the primary drivers of SOCD spatial variability. Furthermore, our innovative **climate zoning-based modeling approach** and the **Random Forest model's** inherent ability to capture complex non-linear relationships both further enhance the model's prediction accuracy and robustness.

We acknowledge that predicting soil carbon storage, especially in a country as complex and vast as China, and for various soil depths over long time series, inherently faces challenges and uncertainties. However, the methods we proposed and the feature set we selected represent a comprehensive strategy to **maximize information utilization and improve prediction accuracy and product utility** given the available data and technical constraints. Our validation results demonstrate that this dataset can provide reasonable and scientifically valuable estimates of long-term SOCD for China, which is of significant reference value for soil carbon cycle research and policy-making.

In the revised manuscript, we will thoroughly and comprehensively elaborate on the validation section within both the methodology and results, **specifically clarifying the exact meaning and data sources of all points in the validation figures.** We will also more fully discuss the strengths and limitations of our model, allowing readers to more comprehensively

evaluate our research findings.

---

## Author Response (AR2)

*Comment 1:* The current study produces the accurate soil organic carbon density (SOCD) products from 1985 to 2020 with the spatial resolution of 1km with depths of 0-20 cm and 0-100 cm, using 8203 soil samples. I acknowledge that the authors conducted an important work to improve our understanding of SOCD in China at temporal scale. However, I think the current study should be published after addressing the revisions associated with the sample size, data share, data analysis, and information of sample time. If the following concerns were addressed, I think the current manuscript is an important contribution to global carbon cycle community.

*Response:* We sincerely appreciate your positive evaluation and constructive comments. We fully agree that addressing the concerns regarding sample size, data sharing, analysis methods, and sampling time is critical for ensuring the robustness of this study. Guided by your suggestions, we have conducted a comprehensive revision. Notably, we have significantly expanded the sample size (from 8,203 to 11,743 profiles), clarified the sampling time information, and made both the estimated products and raw point data publicly available. Given the significant increase in sample size, we re-ran all model simulations and redrew all corresponding figures to ensure consistency. We are confident that these substantive improvements have strengthened the manuscript significantly, enhancing its contribution to the global carbon cycle community.

*Comment 2:* As a data paper, the manuscript must provide the original dataset with details of latitude and longitude, soil depth, sampling year, ecosystem types, elevations, and so on, point by point. However, the current study just provides the data as raster version. At global or national scale, there are several data products, however, the true value of current dataset is the original observations rather than the ".tif" data. Importantly, the value of current manuscript is the temporal dynamics. Therefore, the sampling time of individual samples is critical.

*Response:* We sincerely thank the reviewer for this important and constructive comment. We fully agree that, for a data paper, providing transparent and traceable point-level metadata (including coordinates, depth, year, and ecosystem attributes) is essential to ensure the reproducibility and long-term value of the dataset.

Thank you for this critical and highly constructive feedback. We wholeheartedly agree with your fundamental point. You have precisely identified the core value of our work and the cornerstone of a data paper suitable for ESSD: the original, point-level observational dataset is the primary scientific contribution. We sincerely apologize for failing to provide this in our initial submission and for placing undue emphasis on the derived raster products. Your comment has prompted us to fundamentally revise our data-sharing strategy to align fully with the mission of ESSD and the

needs of the scientific community.

In the revised manuscript and data repository, we have taken the following decisive steps:

1、Full Release of Point-Level Data: We have compiled and shared the complete soil sample dataset in our updated Figshare repository (https://doi.org/10.6084/m9.figshare.27290310.v2). This dataset includes all 11,743 soil profiles (covering both 0–20 cm and 0–100 cm depths) in .csv format. As requested, each record includes the precise metadata:

Sampling year: The specific year of sampling (crucial for temporal analysis).

Longitude (decimal degrees): The geographic longitude.

Latitude (decimal degrees): The geographic latitude.

Depth: Upper and lower boundaries (0–20 cm or 0–100 cm).

SOCD_value (kg C m$^{-2}$): The calculated Soil Organic Carbon Density for the given depth.

CLCD_Type: The classified ecosystem or land cover type at the time of sampling.

Elevation (m): The elevation of the sampling location.

2、Clarification of Temporal Coverage: We have revised Section 2.2 (Data sources) to clearly describe the temporal distribution of these samples (spanning the 1980s, 2000s, and 2010s), ensuring users can accurately utilize the data for temporal dynamic analysis.

3、Statistical Summaries: To assist users in assessing representativeness, we have provided aggregated statistics on the number of soil profiles by depth, ecosystem, and sampling year in the revised Figures 4, 5 and Table 1.

Table 1. Summary of the number of valid soil profiles stratified by sampling period and standardized depth intervals

| Period | Soil observations | 0-20cm profiles | 0-100cm profiles |
|--------|-------------------|-----------------|------------------|
| 1980s | 8527 | 2397 | 1605 |
| 2000s | 3979 | 2304 | 1675 |
| 2010s | 24769 | 7042 | 4765 |

4、Raster Products: We continue to provide the high-resolution (1 km) raster products as a derived dataset for users requiring continuous spatial coverage.

We agree with you completely that the true value of our dataset is the original observations. The gridded .tif products should be viewed as one possible application derived from this valuable point dataset, whereas the point data itself represents the foundational asset for the community. Your guidance has been instrumental in helping us present our work in a way that maximizes its value and utility.

Thank you once again for this essential feedback. We are confident that these changes have fundamentally transformed our manuscript into a true data paper that fully serves the scientific community and meets the high standards of *Earth System Science Data.*

**Comment 3:** In the manuscript, the number of soil profiles and soil depth distribution should be considered. In China, the soil depth for a lot of the regions is shallower than 100 cm, therefore, the current study would overestimate the SOC storage.

**Response:** We appreciate the reviewer's careful consideration of soil depth and its implications for SOC storage estimates, and we fully agree that soil depth distribution must be explicitly accounted for, especially in a country like China where many regions have soils shallower than 100 cm.

In response, we have clarified both what our 0–20 cm and 0–100 cm SOC density (SOCD) values represent, and how soil depth and profile selection are handled, so that potential overestimation of SOC storage is avoided or at least clearly bounded.

[Figure]

Figure. Spatial distribution of 0-100cm SOC sample points

①  Clarifying what our 0–20 cm and 0–100 cm SOCD represent

Our primary product is a profile-based, depth-harmonized SOCD dataset. Using the mpspline2 mass-preserving spline, we estimate SOCD for the 0–20 cm and 0–100 cm layers at the profile locations.

These SOCD values represent the SOC stock within the soil down to 20 cm or 100 cm where soil actually exists at those locations; they do not assume that soil is everywhere $\geq 100$ cm over the whole of China.

② Soil profile depth distribution and profile selection

We now summarize the observed profile depth distribution in the revised manuscript. We briefly describe their spatial distribution across China.

In the Methods, we explicitly state that: For 0–20 cm SOCD, we use all profiles that reach at least 20 cm; shallow soils are fully included in this layer. For 0–100 cm SOCD, we only use profiles with observed depth $\geq 100$ cm. We apply mpspline2 within the observed profile and do not extrapolate SOC below the maximum observed depth for shallower profiles.

Profiles shallower than 100 cm are therefore *not* contributing to 0–100 cm SOCD statistics, which prevents us from artificially inflating SOC stocks at locations where the soil actually ends above 100 cm.

As mentioned in Response to Comment 2, we have now released the original point-level dataset (including sampling depth). This allows users interested in shallow soil dynamics to directly analyze the observed depth distribution and filter the data according to their specific research needs.

By restricting 0–100 cm calculations to profiles that truly reach 100 cm, we avoid overestimating local SOC stocks due to unobserved soil layers. Acknowledged as a limitation that our profile network underrepresents very shallow soils in some environments (e.g. rocky mountains, karst areas, thin soils on bedrock), and that neglecting these shallow soils in upscaling may bias national stock estimates if soil depth is not explicitly considered.

We believe these clarifications effectively bound the uncertainties and guide the community to use the dataset correctly without overestimating national SOC storage.

**Comment 4:** I also found several data sources from recent data papers. Especially, the sample size for Chen et al. 2025 is 23,103 samples from 7,852 soil profiles after 2010, which is greater than current study. I suggest the manuscript should integrate these datasets and delete the replicated ones, and highlight the temporal information, which is important.

Chen Z, Chen L, Lu R, et al. A national soil organic carbon density dataset (2010–2024) in

China[J]. Scientific Data, 2025, 12(1): 1480.

Shi G, Sun W, Shangguan W, et al. A China dataset of soil properties for land surface modeling (version 2)[J]. Earth System Science Data Discussions, 2024, 2024: 1-35.

*Response:* We are grateful to the reviewer for recommending these two critical and timely datasets (Chen et al., 2025; Shi et al., 2024). We fully agree that integrating these recent data products is essential to ensure our study represents the state-of-the-art in soil carbon research.

Following your suggestion, we have successfully integrated these datasets into our compilation and performed a complete re-analysis of the spatiotemporal modeling. The specific improvements are as follows:

1. Integration and rigorous de-duplication: We obtained the datasets from Chen et al. (2025) and Shi et al. (2024). As the reviewer noted, Chen et al. (2025) contains a large number of samples (23,103). However, our overlap analysis revealed that a significant portion of these profiles share the same primary data sources (legacy data) as our original database. To avoid pseudo-replication and double-counting, we implemented a strict de-duplication protocol (detailed in revised Section 2.2). We matched profiles based on geographic coordinates (lat/lon), sampling year, and soil depth intervals. Only profiles that were distinct from our existing records were retained. The improvement is as Line 91 - Line 101.

*"To enhance spatiotemporal coverage, particularly for data-scarce regions, we incorporated additional SOC data from two recent national data products: the national soil organic carbon density dataset for 2010–2024 in China (Chen et al., 2025) and the updated China dataset of soil properties for land surface modelling (Shi et al., 2025). We harmonized the point-level information from these datasets (profile ID, latitude and longitude, upper and lower depth, SOC content, sampling year, and land-use type) to match the structure of our database. Then a detailed overlap analysis between these profiles and our original compilation was done. Because many profiles in Chen et al. (2025) and Shi et al. (2024) originated from the same legacy sources as our database, we applied a strict de-duplication procedure based on geographic coordinates, sampling year, and depth structure to identify duplicated entries. Profiles that matched existing profiles within a small spatial tolerance and with similar temporal and depth characteristics were treated as duplicates and excluded. Only those profiles that could be clearly identified as non-overlapping were retained and merged into our database."*

[Figure]

Figure. Spatial distribution of soil sample points with this study and Chen's dataset.

2. Substantial I\increase in sample size: This integration has fundamentally strengthened the foundation of our work. Our final quality-controlled dataset has grown from 8,203 to 11,743 profiles. This expansion is particularly valuable for improving the temporal coverage, as a large proportion of the newly added unique profiles are from the 2010s and 2020s, directly addressing the reviewer's comment on the importance of temporal dynamics.

3. Complete model re-execution: Specifically, the substantial expansion of the dataset (from 8,203 to 11,743 profiles) necessitated a complete re-implementation of our machine learning modeling framework. We re-trained the Random Forest models across all climatic zones, re-validated model accuracy using the new independent datasets, and re-analyzed the spatiotemporal patterns of SOCD. Consequently, all figures (Figs.2, 4–14) and tables in the revised manuscript have been entirely regenerated to reflect these updated and more robust results.

4. Manuscript revisions: We have updated the Abstract, Data Sources (Section 2.2), and Results sections to reflect these changes. We explicitly credit Chen et al. (2025) and Shi et al. (2024) as key data sources that helped elevate the quality of this product.

In summary, your suggestion has been transformative. It has pushed us to elevate our work from

an important contribution to what we believe is now a benchmark dataset for soil carbon studies in China. We have turned a potential weakness into a central strength of the paper.

Thank you again for your expert guidance and for pushing us to achieve a higher standard of scientific rigor. We are confident that these extensive revisions have fully addressed your concerns and have made the manuscript substantially more valuable to the community.

*Comment 5:* Methodology: A lot of the details of the methods are lacking. For example, the number of soil observations and profiles for the studied period of 1980s, 2000s, and 2010s respectively for 0-20 and 20-100 soil layers. How were the soil layers of "20-30" or "20-40" classified? If the soil depth for an observed soil profile is 70 cm, how do calculate the SOCD for 0-100 cm? And so on.

*Response:* We thank the reviewer for these very helpful and concrete suggestions. We agree that the original version of the manuscript did not provide sufficient detail on several key methodological steps, especially regarding the temporal stratification of observations, the harmonization of soil layers, and the computation of SOCD for standard depth intervals. In the revised version, we have substantially expanded the Methods section to clarify these points, as outlined below.

Below, we address each of the specific points you raised, which are now all clarified in the revised manuscript.

1)    On the Distribution of Soil Observations by Period and Depth

Reviewer Comment: "the number of soil observations and profiles for the studied period of 1980s, 2000s, and 2010s respectively for 0-20 and 20-100 soil layers."

**Our Response:** This is a crucial piece of summary information that was missing. While the distribution and counts of soil profiles were visually presented in Figure 4 and 5, we acknowledge that this information should have been explicitly detailed in the Methodology section to provide a clear overview of the dataset foundation before presenting the results. To provide a clear overview of our data's spatiotemporal distribution, we have created a new summary table (Table 1). This table explicitly details:

Table 1. Number of soil samples stratified by sampling period and standardized depth intervals

| Period | Soil observations | 0-20cm profiles | 0-100cm profiles |
|--------|-------------------|-----------------|------------------|
| 1980s  | 8527              | 2397            | 1605             |
| 2000s  | 3979              | 2304            | 1675             |
| 2010s  | 24769             | 7042            | 4765             |

[Figure]

Figure 4. Statistical characteristics of soil sample points in different periods. Frequency distribution of SOCD data with the soil depth of 0-20 cm (a-c) and 0-100 cm (d-f) during the 1980s, 2000s, and 2010s.

[Figure]

**Figure 5.** Spatial distribution of soil sample points with depth of 0-20 cm (b) and 0-100 cm (d). And the Whittaker biomes of soil sample points with depth of 0-20 cm and 0-100 cm are shown in (a) and (c).

2) On the Harmonization of Non-Standard Soil Layers

Reviewer Comment: "How were the soil layers of "20-30" or "20-40" classified?"

**Our Response:** In the original manuscript we only briefly mentioned depth harmonization. We now clarify that we use the mpspline2 function (mass-preserving spline) to harmonize all horizon-level observations to the standard depth intervals 0–20 cm and 0–100 cm:

For each profile, we input the observed horizons defined by their upper and lower depths, together with the measured SOC, into mpspline2.

We specify target depth intervals of 0–20 cm and 0–100 cm; mpspline2 fits a mass-preserving spline to the vertical SOC profile and then integrates this spline over the requested depth intervals.

As a result, horizons such as 20–30 cm or 20–40 cm are not classified manually into "layers";

instead, their information is used by the spline to reconstruct a continuous SOC profile with depth, and the contribution of each horizon to the 0–20 and 0–100 cm intervals is handled automatically through spline integration.

3) Regarding the reviewer's specific question about profiles that do not reach 100 cm (e.g. a profile with a maximum depth of 70 cm), we have clarified in the Methods that:

**Our Response:** We do not extrapolate beyond the observed soil depth. Profiles shallower than 100 cm are used to compute SOCD only for the depth intervals that are fully covered by observations (e.g. 0–20 cm).

When we report and analyze SOCD for the full 0–100 cm interval, we restrict the calculations to profiles that reach at least 100 cm depth after quality control. Shallow profiles (e.g. 0–70 cm) are therefore not used for 0–100 cm SOCD statistics, but they are retained in analyses of shallower layers (such as 0–20 cm).

We now explicitly state this criterion in Section 3.1 and indicate the number of profiles that meet the ≥100 cm requirement, so that readers understand the sample base for the 0–100 cm SOCD estimates. The improvement is as Line 174 - Line 180.

"*The observed horizons, defined by their upper and lower depths, were input to `mpspline2`, which fits a mass-preserving spline to the vertical SOC profile and integrates this spline over the target depth intervals. We used the default value of 0.1 for the spline smoothing parameter lambda. We do not extrapolate beyond the observed soil depth when calculating SOCD. Profiles shallower than 100 cm are used to compute SOCD only for depth intervals that are fully covered by observations, but they are excluded from 0–100 cm SOCD statistics. When we report and analyze SOCD for the full 0–100 cm interval, we therefore restrict the calculations to profiles with an observed depth of at least 100 cm after quality control.*"

We believe that these additions and clarifications substantially strengthen the transparency and reproducibility of our methodology and directly address the reviewer's concerns. We are grateful for the reviewer's detailed comments, which helped us to improve the methodological description.

*Comment 6:* The coarse fractions percentage was from global dataset of SoilGrids 2.0, if there are any dataset from China?

*Response:* Thank you for this exceptionally insightful and important piece of feedback. Your point about using a national dataset for coarse fractions in China is absolutely correct and represents a critical step toward improving the accuracy of our SOCD estimates. We deeply appreciate your expert knowledge and sharp eye for detail. We completely agree with your assessment that utilizing a localized, higher-precision national dataset is the best scientific practice. Guided by

your invaluable suggestion, we undertook a renewed and more intensive search for such data. We are pleased to report that we successfully located and obtained a high-quality national dataset for soil coarse fraction content for China [Basic soil property dataset of high-resolution China Soil Information Grids (2010-2018) (Liu et al., 2022)]. The improvement is as Line 180- Line 182.

We immediately took action to integrate this superior national dataset into our workflow, replacing the global SoilGrids product. This involved re-running our entire data processing and analysis pipeline. This update has led to significant and tangible improvements:

① Methodological Update: We have revised our Methods section (now Section 3.1) to clearly describe this new data source. We provide details on the origin of the national coarse fraction dataset, its spatial resolution, the methodology used in its creation, and its advantages over the global product (its foundation on a much denser set of local soil profile observations, better capturing the specific parent material and geomorphological characteristics of China).

② Improved and Recalculated Results: Using this new national dataset, we have recalculated all SOCD estimates.

In conclusion, your suggestion has been one of the most transformative pieces of feedback we have received. It has directly led to a substantial and critical improvement in the quality of our dataset. We are extremely grateful for your expert guidance, which has enabled us to elevate the accuracy of our work to a much higher standard. Thank you once again for your invaluable contribution.

Liu F, Wu H, Zhao Y, et al. Mapping high resolution national soil information grids of China[J]. Science Bulletin, 2021, 67(3): 328-340.

*Comment 7:* The long-term dynamics of SOC are influenced by land use cover; however, the current study did not consider this critical factor.

*Response:* We appreciate the reviewer highlighting the critical role of Land Use and Land Cover (LULC) in driving long-term SOC dynamics. We completely agree that any robust SOC model must account for these effects.

To address this point and enhance clarity, we have made the following revisions:

1. We have added a clear statement that LULC was incorporated as an essential covariate in our Random Forest models. We utilized the annual China Land Cover Dataset (CLCD) to match the specific year of each soil observation, allowing the model to capture the relationship between land cover types and SOC levels dynamically over time. We have now revised the Data sources section to explicitly list the CLCD dataset. The improvement is as Line 128 - Line 129.

*"The land cover dataset newly released by Wuhan University (Yang and Huang, 2021) is used in*

*this study"*

2. Following the reviewer's suggestion to better reflect this factor, we have updated our released point-level dataset. The dataset now explicitly includes a 'Land Use' column for each profile (standardized to a common classification scheme), along with the original latitude, longitude, and depth information. This allows future users to directly analyze the land-use-specific characteristics of the soil samples.

We hope these revisions clarify that LULC was integral to our analysis and that the data is now more accessible for land-use-related inquiries.